# Enhancing aortic valve drug delivery with PAR2-targeting magnetic nano-cargoes for calcification alleviation

Jinyong Chen [1,2,3,6], Tanchen Ren [1,2,3,6] ✉, Lan Xie[1,2,3,6], Haochang Hu[1,2,3], Xu Li[4], Miribani Maitusong[1,2,3], Xuhao Zhou[1,2,3], Wangxing Hu[1,2,3], Dilin Xu[1,2,3], Yi Qian[1,2,3], Si Cheng[1,2,3], Kaixiang Yu[1,2,3], Jian`an Wang [1,2,3,5] ✉ & Xianbao Liu [1,2,3] ✉

Calcific aortic valve disease is a prevalent cardiovascular disease with no available drugs capable of effectively preventing its progression. Hence, an efficient drug delivery system could serve as a valuable tool in drug screening and potentially enhance therapeutic efficacy. However, due to the rapid blood flow rate associated with aortic valve stenosis and the lack of specific markers, achieving targeted drug delivery for calcific aortic valve disease has proved to be challenging. Here we find that protease-activated-receptor 2 (PAR2) expression is up-regulated on the plasma membrane of osteogenically differentiated valvular interstitial cells. Accordingly, we develop a magnetic nanocarrier functionalized with PAR2-targeting hexapeptide for dual-active targeting drug delivery. We show that the nanocarriers effectively deliver XCT790—an anti-calcification drug—to the calcified aortic valve under extra magnetic field navigation. We demonstrate that the nano-cargoes consequently inhibit the osteogenic differentiation of valvular interstitial cells, and alleviate aortic valve calcification and stenosis in a high-fat diet-fed low-density lipoprotein receptor-deficient ($Ldlr^{-/-}$) mouse model. This work combining PAR2- and magnetic-targeting presents an effective targeted drug delivery system for treating calcific aortic valve disease in a murine model, promising future clinical translation.

Calcific aortic valve disease (CAVD) is a common cardiovascular disease with high incidence and poor prognosis[1]. The development of transcatheter aortic valve replacement (TAVR) markedly improved the longevity and quality of life of patients with severe CAVD. Yet, no drugs have been approved for treating aortic valve calcification[2,3]. Hence, exploring effective strategies to prevent valve calcification may provide patients with additional therapeutic options.

As a chronic progressive disease, the efficacy and safety of drugs, as well as the patient's compliance and acceptance, are significantly influenced by the dosage and frequency of drug administration. The efficacy of conventional systematic drugs can be limited by liver metabolism, kidney excretion, and nonspecific distribution, leading to a relatively small amount of drugs reaching the target site[4]. Therefore, strategies capable of improving the drug targeting ability and reducing

[1]Department of Cardiology, The Second Affiliated Hospital, Zhejiang University School of Medicine, 310009 Hangzhou, P.R. China. [2]State Key Laboratory of Transvascular Implantation Devices, 310009 Hangzhou, P.R. China. [3]Cardiovascular Key Laboratory of Zhejiang Province, 310009 Hangzhou, P.R. China. [4]Department of Vascular Surgery, Zhongshan Hospital, Fudan University, 200030 Shanghai, P.R. China. [5]Research Center for Life Science and Human Health, Binjiang Institute of Zhejiang University, Hangzhou 310053, P.R. China. [6]These authors contributed equally: Jinyong Chen, Tanchen Ren, Lan Xie. ✉e-mail: rentanchen120@zju.edu.cn; wangjianan111@zju.edu.cn; liuxb@zju.edu.cn

adverse side effects have broad application prospects, prompting the development of targeted drug delivery systems (TDDSs)[5,6]. Indeed, a multi-bioactive TDDS has proven effective as a promising therapeutic strategy for abdominal aortic aneurysms[7]. In addition, biomimetic liposomes with excellent biocompatibility and homing ability have been applied for the targeted treatment of myocardial infarction and atherosclerosis[8,9]. However, similar progress is yet to be made in CAVD. The development of TDDSs for CAVD has faced three primary obstacles. First, the blood flow velocity around the aortic valves exceeds 4 m s$^{-1}$ after stenosis, impeding the accumulation of drugs or nanoparticles within the leaflets[10]. Second, aortic valves do not contain a capillary plexus, restricting the entry and retention of nanomedicines in the leaflets (i.e., passive targeting)[11,12]. Third, no specific targeting markers have been reported for CAVD, hindering the effective delivery of drugs via active targeting.

Magnetic targeting is a promising strategy for improving drug delivery efficiency. Due to the responsiveness to exogenous magnetic field (EMF), which is non-invasive and tissue-penetrating, magnetic nanoparticles (MNPs)-based drug delivery systems are highly anticipated, especially for targeting deep tissues. To date, MNPs have primarily been applied for the targeted treatment of tumors. Guided by EMF, MNPs increase the localization of antineoplastic drugs and serve as the source of magneto-thermodynamic therapy[13]. Moreover, the combination of magnetic targeting and hyperthermia has been applied in anti-infection therapy within deep tissues (e.g., bacterial osteomyelitis in bone marrow), which are not readily accessible via systemic drug administration[14]. Meanwhile, MNPs have been increasingly used to transport therapeutic agents to other pathological tissues, including the ischemic brain and heart[15,16]. Additionally, mesenchymal stem cell-derived exosomes incorporated with MNPs have been directed to brain ischemic lesions, ultimately reducing the infarct volume and promoting motor function recovery[17]. MNPs have also been conjugated with anti-CD63 and anti-myosin light chain (MLC) antibodies to increase heart function after myocardial infarction. More specifically, the MNPs capture endogenous circulating exosomes via the anti-CD63 antibody, causing accumulation in the ischemic heart under EMF navigation; subsequently, the anti-MLC antibodies bind to the damaged cardiomyocytes, releasing exosomes and increasing heart function[18]. Although MNPs can be enriched around the diseased tissue under EMF guidance, the drug anchoring capacity must be enhanced to counteract the violent blood flow of the stenotic aortic valves. Moreover, targeting accuracy must be improved to enable nanoparticles to act on disease-causing valvular interstitial cells (VICs).

A cell membrane marker specifically upregulated and exposed in diseased tissues can be a target for drug anchoring. During CAVD progression, the lesion site features endothelium damage and the osteogenic differentiation of VICs. Hence, cell membrane markers highly expressed in osteogenically differentiated VICs could represent a potential TDDS targeting site for treating CAVD. One such example is protease-activated receptor-2 (PAR2), a membrane-bound G-protein coupled receptor; its hexapeptide ligand has high affinity and selectivity in binding to the extracellular segment of PAR2[19]. Once bound to its ligand, PAR2 is internalized and routed to lysosomes[20,21]. Moreover, PAR2 has been implicated in atherosclerosis, which shares similar initial pathological features with CAVD[22,23]. Hence, if the protein expression of PAR2 is upregulated in CAVD, it may represent a potential marker of CAVD for a TDDS.

When constructing a TDDS for CAVD, effective therapeutic drugs must also be incorporated. XCT790 is a specific antagonist of estrogen-related receptor alpha (ERRα) that was recently found to inhibit the osteogenic differentiation of human VICs (hVICs)[24]. In fact, of the 1595 drugs that were screened in a previous study, XCT790 was the most effective in correcting transcriptome disorder related to osteogenic differentiation of induced pluripotent stem cell (iPSC)-derived valvular endothelial cells (VECs). Moreover, it effectively

inhibited valve calcification in *Notch1*$^{+/-}$/*mTR*$^{G2}$ mice[25]. However, the mechanism by which XCT790 inhibits osteogenic differentiation of VICs has not been fully characterized. Additionally, systematically inhibiting ERRα may induce nonalcoholic steatohepatitis (NASH), behavioral disturbances, or other adverse events, hindering its clinical application[26,27]. Meanwhile, ERRα is localized within the nucleus, and XCT790 must enter the cell to induce the therapeutic pathways[28]. Thus, an effective TDDS can increase the therapeutic effect and reduce the side effects of XCT790, which may accelerate the clinical translation of CAVD drugs.

Herein, we sought to develop a highly efficient drug delivery platform for CAVD with dual active targeting capabilities to inhibit VICs osteogenic differentiation, thereby attenuating aortic valve calcification and reducing adverse side effects. Poly (lactic-co-glycolic acid) (PLGA) was applied to construct the nano-sized cargo due to its ability to carry the fat-soluble drug XCT790 and its good biocompatibility and biodegradability[29]. To enhance drug retention in the aortic valve further, we inlaid MNPs in the core of PLGA nanoparticles to achieve local magnetic attraction during circulation. Additionally, the nanocarrier was coated with polyethylene glycol (PEG) to enhance the biocompatibility and evade macrophage phagocytosis[5,30]. Most importantly, the nanoparticles were functionalized with PAR2 hexapeptide ligands to ensure CAVD anchoring. Functionally, once injected intravenously, the nano-cargoes became enriched on the calcified leaflets via magnetic field navigation. Mediated by the binding of the hexapeptide to PAR2, the nanoparticles adhered to and were endocytosed by osteogenically differentiated VICs. Finally, XCT790 was released into the cytoplasm, regulating the metabolic reprogramming of VICs, inhibiting the osteogenic differentiation of the VICs, and suppressing calcium deposition in the leaflets (Fig. 1).

Here, we identified a cell surface anchoring site for calcified VICs and devised a dual active TDDS based on surface anchoring and magnetic navigation to achieve pathological aortic valve targeting. This platform represents a powerful platform to potentially provide solutions and ideas for the unmet challenge regarding the therapeutic efficacy of CAVD drugs.

## Results

### PAR2 expression is increased in osteogenically differentiated hVICs and calcified aortic valves

Embrane proteins of VICs which were up-regulated after osteogenic differentiation were screened as targets of the CAVD TDDSs. To this end, we analyzed the gene expression profiles of quiescent hVICs and osteogenically differentiated hVICs (cellular characterization shown in Supplementary Fig. 1) and identified 3468 up-regulated genes (filtered by |fold change (FC)| ≥ 2 and $p < 0.01$; Supplementary Fig. 2a, b). Gene Ontology cellular component analysis was conducted, and genes whose corresponding proteins localized to the cell membrane were screened. After excluding those proteins without specific ligands, the top 20 differentially expressed genes were selected as candidate markers (Fig. 2a). Quantitative polymerase chain reaction (qPCR) verified that the expression of *F2RL1*, *F2RL2*, *NPR1*, *FPR3*, and *CSF2R* were prominently up-regulated after osteogenic induction. In contrast, *NPR1*, *FPR3*, and *CSF2R* were expressed at very low levels in hVICs (Fig. 2b, Supplementary Fig. 2c). Notably, PAR3 (i.e., *F2RL2*-encoded protein) does not respond to synthetic peptides that mimic the putative tethered ligand[31]. Western blot further confirmed that, with prolonged osteogenic induction time, the protein expression of PAR2 (i.e., *F2RL1*-encoded protein) increased in a time-dependent manner and was positively correlated with osteogenic differentiation markers (Fig. 2c).

In addition, considering that the myofibrogenic differentiation of hVICs is another main cause of aortic valve fibrosis and calcification, hVICs were stimulated with 10 ng mL$^{-1}$ transforming growth factor-β (TGF-β) to induce myofibrogenesis[32]. Western blot results showed that

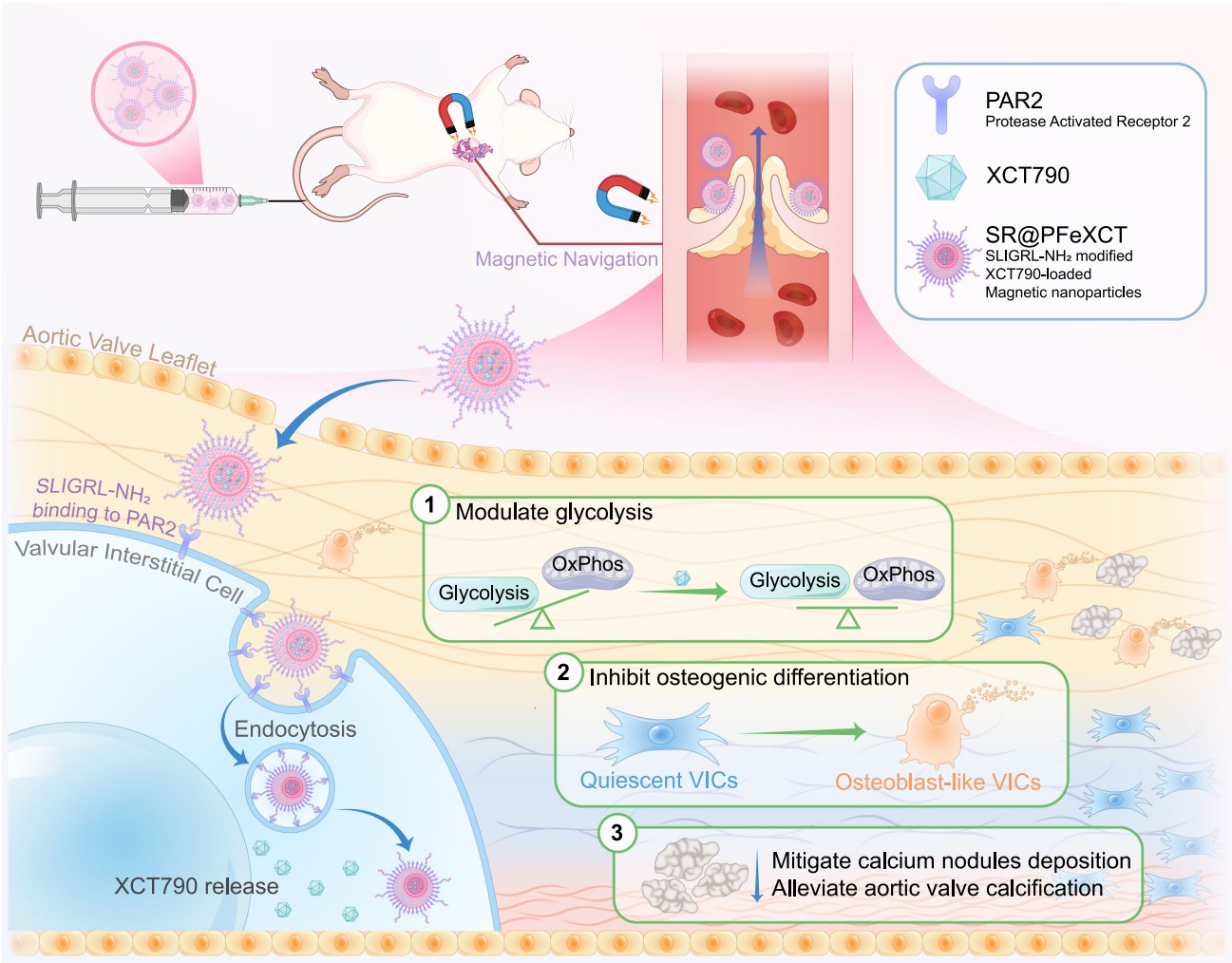

**Fig. 1 | Protease-activated receptor-2 (PAR2)-targeted XCT790-loaded magnetic nanoparticles for the targeted treatment of CAVD.** Targeted nano-cargoes are enriched in the calcified leaflets under the navigation of exogenous magnetic field, adhere to and are endocytosed by osteogenically differentiated valvular interstitial cells (VICs) with the mediation of PAR2, Afterwards, regulate the metabolic reprogramming and inhibit the osteogenic differentiation of VICs, and eventually suppress calcium deposition in the leaflets.

PAR2 protein expression was increased in response to TGF-β stimulation (Fig. 2d). Considering the differences in the pathological process of CAVD between female and male patients, including a higher fibrotic burden in female patients relative to their male counterparts, PAR2 protein expression were examined in female patient-derived hVICs[33]. Immunoblot confirmed the elevation of PAR2 expression in response to osteogenic differentiation and myofibrogenic differentiation in hVICs derived from female patients (Supplementary Fig. 3a, b).

To further confirm that PAR2 expression is upregulated in osteogenically differentiated VICs as well as in calcified valve tissue in vivo, the protein expression pattern of PAR2 was examined in non-calcified and calcified aortic valves (leaflet morphology shown in Fig. 2e, patient information shown in Supplementary Table 1). Western blot analysis revealed that the protein expression of PAR2 was significantly higher in the calcified valve tissue than in the control valve tissue (Fig. 2f). When male and female valve specimens were analyzed separately, we found that increased PAR2 expression in calcified valves was not affected by sex differences (Supplementary Fig. 3c). To assess the robustness of the findings, propensity-score matching (PSM) was employed to minimize the influence of confounding variables. Baseline parameters, such as age, BMI, sex, smoking, comorbidity, and serum lipid profiles, were utilized as matching criteria. Following the

matching process, no disparities were observed in these baseline characteristics, while differences in PAR2 protein expression remained (Supplementary Table 2). Immunohistochemical staining was performed to assess the distribution of PAR2 in the tri-layer structure of leaflets, including the collagen-rich fibrosa on the aortic side, the proteoglycan-rich spongiosa, and the elastin-rich ventricularis. Compared with those in non-calcified leaflets, PAR2-positive cells were more abundant in all three layers of calcified leaflets, particularly in the peri-calcified and fibrotic areas of the fibrosa layer, as evidenced by Alizarin Red staining and Masson's trichrome staining in serial slices (Fig. 2g, h). Immunofluorescence further revealed that PAR2 was not colocalized with CD31—an endothelial cell marker—but colocalized with vimentin (i.e., a marker of VICs) (Fig. 2i, j)[34]. The number of PAR2-positive cells in the calcified aortic valve markedly was increased compared with normal leaflets (Fig. 2j). Moreover, we isolated hVICs from non-calcified and calcified aortic valves, and confirmed that PAR2 protein expression was increased in hVICs derived from calcified leaflets (Fig. 2k).

To investigate whether PAR2 is up-regulated in the calcified aortic valves of mice, two aortic valve calcification models were established: (1) $Ldlr^{-/-}$ mice fed with high-fat diet (HFD) for 24 weeks to induce valve sclerosis and calcification; (2) wild-type mice subjected to transcarotid

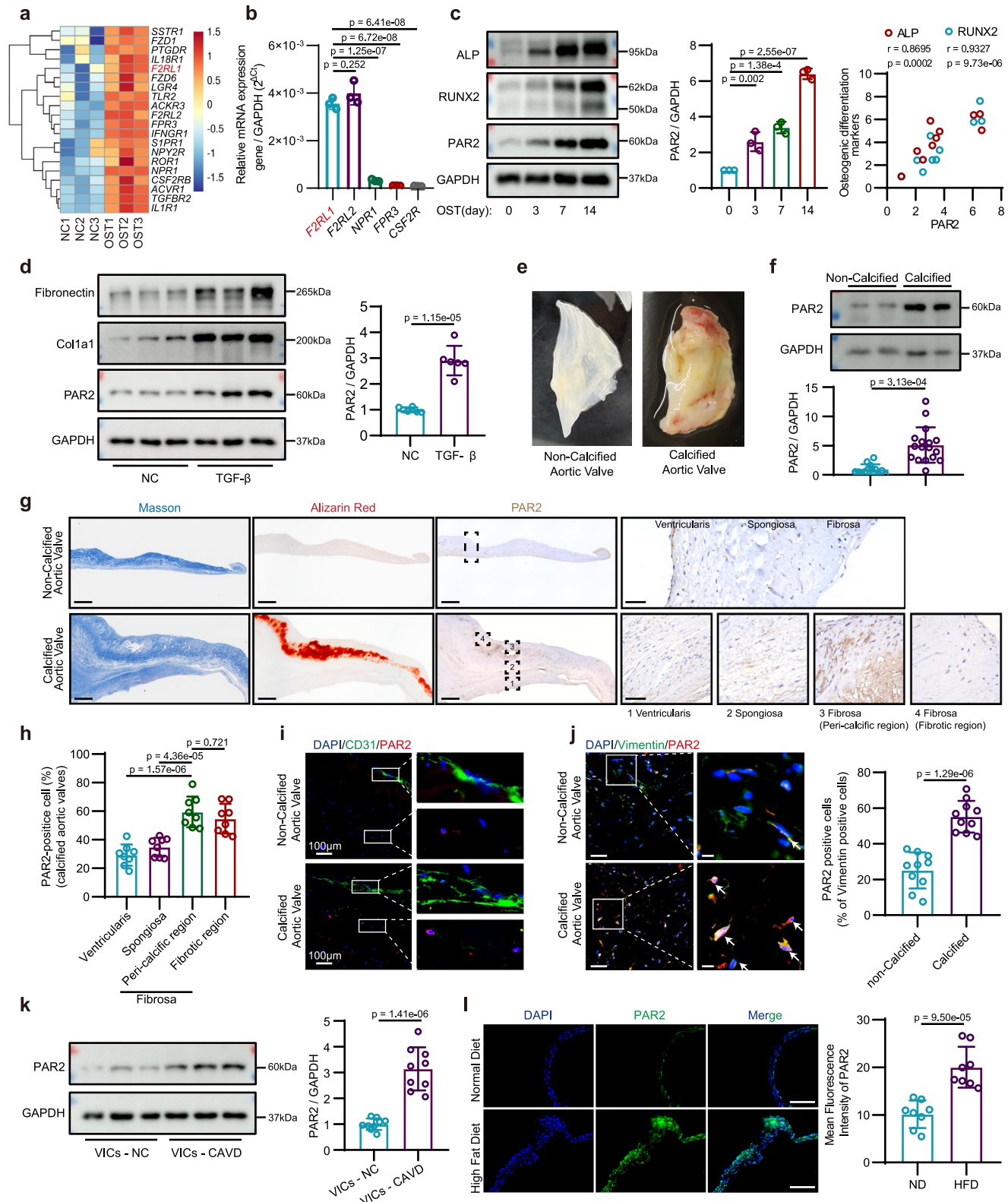

artery wire injury of the aortic valve to induce cell proliferation and valve thickening. Immunofluorescence staining revealed that PAR2 expression was significantly elevated in the aortic valves of *Ldlr*[-/-] mice fed with HFD compared with those fed with normal diet (ND; Fig. 2l). Moreover, PAR2 expression was higher on the aortic side of the leaflet than on the ventricular side (Supplementary Fig. 4). Eight weeks after guidewire injury, elevated PAR2 expression was observed on the battered valves (Supplementary Fig. 5).

Taken together, these data suggest that PAR2, as a membrane protein upregulated in calcified leaflets and osteogenically differentiated hVICs, might be an effective target for a CAVD TDDS.

## PAR2-positive cells contribute to valve calcification

To clarify the physiological functions performed by PAR2-positive cells within the aortic valves, a single-cell RNA sequencing dataset for human aortic valve tissues (PRJNA562645) was analyzed, and cells were

**Fig. 2 | PAR2 expression is increased in osteogenically differentiated hVICs and calcified aortic valves. a** Heatmap of the top 20 differentially expressed genes whose corresponding proteins bind to specific ligands and are localized to the cell membrane. **b** Quantitative analysis of *F2RL1, F2RL2, NPR1, FPR3*, and *CSF2R* mRNA levels in osteogenically differentiated hVICs; $n = 3$ biologically independent samples. **c** Immunoblot and quantification of PAR2, RUNX2, and ALP in hVICs cultured with osteogenic induction medium for different times (0, 3, 7, 14 days); $n = 3$ biologically independent samples. **d** Protein expression of PAR2 in hVICs undergoing myofibrogenic induction for 48 h; $n = 6$ biologically independent samples. **e** Non-calcified and calcified human aortic valve leaflet specimens collected from surgery. **f** PAR2 protein expression in human leaflets; non-calcified group/calcified group: $n = 12/15$ biologically independent samples. **g** Alizarin Red staining, Masson's Tri-chrome staining, and immunohistochemical staining of PAR2 in leaflets (full size, scale bar = 500 μm; enlarged, scale bar = 50 μm). **h** Quantitative analysis of PAR2 in the ventricularis, spongiosa, and fibrosa layers of calcified leaflets; $n = 8$ biologically independent samples. **i** Immunofluorescence staining of CD31 and PAR2 in leaflets. Scale bar = 100 μm; experiment was repeated independently in 3 leaflets. **j** Immunofluorescence photographs and quantitative analysis of vimentin and PAR2 in leaflets (left; scale bar = 50 μm) and in enlarged images (right; scale bar = 10 μm). White arrow indicates localization of PAR2 in hVICs; $n = 10$ biologically independent samples. **k** PAR2 protein levels in hVICs isolated from non-calcified and calcified leaflets; $n = 9$ biologically independent samples. **l** Immunofluorescence staining and quantification of PAR2 in aortic valves of *Ldlr*$^{-/-}$ mice fed a normal or high-fat diet. Scale bar = 200 μm, $n = 8$ biologically independent samples. Values are presented as mean ± SD. Statistical significance of (**b**), (**c**), and (**h**) determined by one-way ANOVA and Tukey's multiple comparison test; statistical significance of (**d**), (**f**), (**j**), (**k**), and (**l**) determined by two-tailed unpaired Student's *t* test. Source data are provided as a Source Data file.

classified according to previously described cell marker genes[35]. Four main cell clusters were identified, namely, valve-derived stromal cells (VDSCs; *LUM, HLA-B, INHBA, SRM, RPL17*, and *EIF3J*), VICs (*FOS, COL1A1*, and *COL3A1*), VECs (*SELE, SERPINE1*, and *PI3*), and macrophages (*CXCL3, CCL3L1*, and *CXCL5*). VDSCs represent a subset of stromal cells primarily residing within calcified valves, are differentiated from VICs or VECs, and can lead to valve fibrosis and calcification[35]. Subsequent analysis revealed that *F2RL1*-positive cells were primarily VICs and VDSCs rather than VECs or macrophages (Supplementary Fig. 6a). In addition, calcified valve-derived *F2RL1*-positive cells had higher *INHBA* (encodes inhibin β-A, a member of the TGF-β superfamily) expression, representing TGF-β pathway activation, as well as higher *CXCL1* (encodes growth-regulated alpha protein) expression, indicating an increased ability to recruit inflammatory cells (Supplementary Fig. 6b). Uniform manifold approximation and projection (UMAP) dimension reduction was performed to reveal the cluster structure of *F2RL1*-positive cells, and gene set enrichment analysis (GSEA) was conducted to predict the cell phenotypes with *LUM, HLA-B, INHBA, SRM, RPL17, EIF3J*, and *CXCL1* selected as the VDSCs gene list[36]. Most *F2RL1*-positive cells belonged to the same cluster (Supplementary Fig. 6c). GSEA revealed that *F2RL1*-positive cells with higher VDSCs-related gene expression primarily originated from CAVD patients (Supplementary Fig. 6d). Hence, using the gene–cell matrix for the *F2RL1*-positive cells, we reconstructed the lineage trajectory between healthy and calcified conditions. Results identified two cell populations from two origins in the healthy condition that differentiated into the same cell population in the CAVD condition (Supplementary Fig. 6e). Moreover, *CXCL1* and *INHBA* expression increased gradually along the *F2RL1*-positive cell lineage trajectory (Supplementary Fig. 6f), indicating that targeted interventions against *F2RL1*-positive cells may inhibit the evolution of this cell cluster and, consequently, CAVD progression.

These results suggest that *F2RL1*-positive cells are disease-causing cells that can cause valve fibrosis and calcification, further supporting the design of targeted interventions against valve cells with high PAR2 expression.

## PAR2-ligand functionalization enhances nanoparticle binding to osteogenically differentiated hVICs

To verify the feasibility of PAR2 as a binding site of calcified valves for nanocarriers, FDA-approved materials with strong clinical translational potential are more appropriate. PLGA is a promising biodegradable and biocompatible polymer with surface modification potential, and can well load hydrophobic drugs and nanoparticles[37,38]. More specifically, an oil-in-water (O/W) emulsion solvent evaporation method was used to synthesize the nanoparticle core comprising PLGA, XCT790, and 10-nm MNPs, with bovine serum albumin (BSA, an amphiphile) dissolved in water as a stabilizer. The BSA molecules on the particle surface provided amino groups that became covalently conjugated to OHC−PEG−CHO via condensation of aldehyde and amino groups. OHC−PEG−CHO also functioned as the crosslinker of PAR2-targeting

peptides (SLIGKV-NH$_2$ or SLIGRL-NH$_2$) (Fig. 3a). First, to optimize the ratio of ligands and nanoparticles, BSA was labeled with Cy7, and SLIGKV-NH$_2$ peptides were labeled with Rhodamine B; relative quantification was conducted according to the fluorescence intensity ratio (Rhodamine B/Cy7). When the ligand concentration increased from 3.91 μg mL$^{-1}$ to 0.125 mg mL$^{-1}$, the fluorescence intensity ratio gradually increased, indicating an increase in the number of peptides on the nanoparticle surface. However, when the ligand concentration exceeded 0.125 mg mL$^{-1}$, the increase in fluorescence intensity slowed significantly, indicating that the nanoparticle surface had become saturated with peptides (Supplementary Fig. 7). In this study, we used a concentration of 0.2 mg/mL to ensure a high peptide grafting ratio. Additionally, the size distribution and zeta potential of the nano-carriers were measured. The obtained PLGA nanoparticles had an average size of 147.4 ± 6.3 nm, which augmented to 186.8 ± 6.9 nm following modification without impacting the size distribution (Fig. 3b). The zeta potential was increased from −26.93 ± 0.57 mV to −18.50 ± 0.79 mV, potentially due to the negatively charged BSA being covered by more neural PEG (Fig. 3c). Scanning electron microscopy (SEM) confirmed the spherical structure of SLIGKV-modified, MNPs and XCT790-loaded nanoparticles (SK@PFeXCT); SK@PFeXCT was 159.4 ± 6.7 nm, which agreed with the DLS results (Fig. 3d).

To assess the targeting ability of nanoparticles for osteogenically differentiated hVICs, Cy5-labeled nanoparticles (PFeCy5 or SK@PFeCy5) were synthesized and incubated with osteogenically differentiated hVICs. Nanoparticle binding was observed by fluorescence microscope and quantified by flow cytometry. The SLIGKV-NH$_2$ modification significantly enhanced the affinity of nanoparticles for hVICs, as evidenced by the increased colocalization of SK@PFeCy5 with hVICs compared with PFeCy5 (Fig. 3e, f). Moreover, osteogenically differentiated hVICs captured 5-fold more SK@PFeCy5 than quiescent hVICs, indicating that modification of nanoparticles with SLIGKV-NH$_2$ conferred the ability to target osteogenically differentiated hVICs (Fig. 3e, f). To confirm that upregulated PAR2 mediated the increased nanoparticle internalization, osteogenically differentiated hVICs were transfected with PAR2 siRNA and incubated with nanoparticles; PAR2-knockdown efficiency was verified by western blotting (Supplementary Fig. 8). A reduced number of Cy5-positive hVICs and decreased mean fluorescent intensity (MFI) were observed, confirming the importance of PAR2 for the targeting ability of SK@PFeCy5 (Fig. 3e, f). To further verify the specific targeting ability (i.e., the specific binding of SLIGKV-NH$_2$ with PAR2), nanoparticle cores were conjugated with a mismatch peptide (SLKGIV-NH$_2$) and incubated with hVICs. The mean fluorescence of hVICs cultured with nanoparticles conjugated with the mismatch peptide was significantly decreased compared with those cultured with SK@PFe (Supplementary Fig. 9). Given that PAR2 expression is elevated in hVICs following myofibrogenic differentiation induced by TGF-β, we stimulated myofibrogenically differentiated hVICs with SK@PFeCy5. Increased binding of SK@PFeCy5 was also observed in myofibrogenically

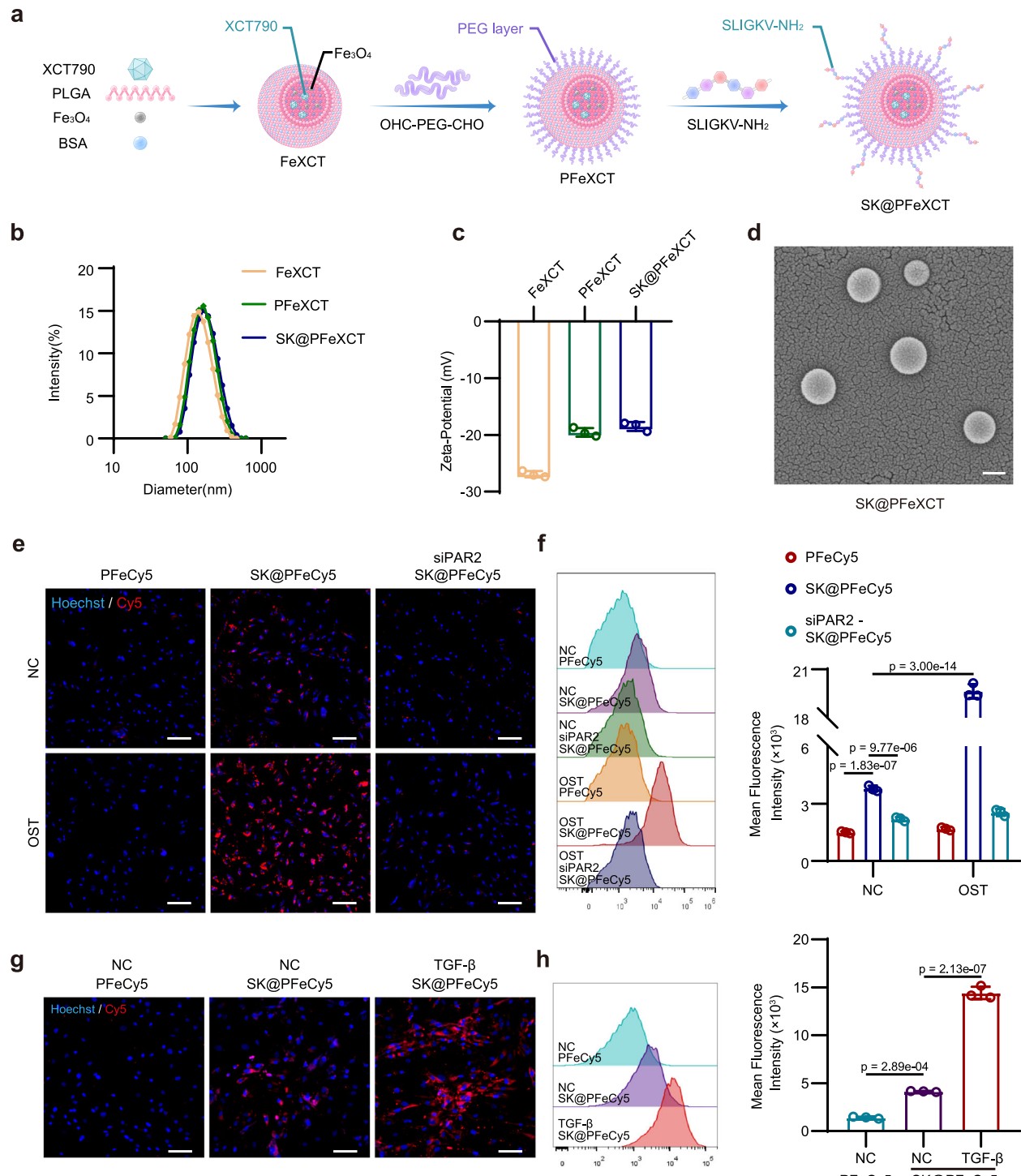

**Fig. 3 | Characterization of targeting nanoparticles and their targeting ability toward osteodifferentiated hVICs in vitro. a** Schematic diagram of SK@PFeXCT preparation. **b** Size distribution curves and (**c**) Zeta-potential of FeXCT, PFeXCT, and SK@PFeXCT; $n = 3$ independent experiments; one-way ANOVA and Tukey's multiple comparison test. **d** Scanning electron microscopy (SEM) images of SK@PFeXCT; Scale bar = 100 nm; experiment was repeated independently for 3 times. **e** Fluorescence images of hVICs following 14 days of osteogenic induction, 24 h of siRNA-mediated PAR2 knockdown, and 8 h of 20 μg mL$^{-1}$ PFeCy5 or SK@PFeCy5 incubation; Scale bar = 200 μm. **f** Flow cytometry histogram and quantification of the PFeCy5 or SK@PFeCy5 uptake by hVICs; $n = 3$ biologically independent samples; two-way ANOVA and Tukey's multiple comparison test. **g** Fluorescence images of hVICs undergoing 2 days of TGF-β-induced myofibrogenic differentiation followed by 8 h of PFeCy5 or SK@PFeCy5 incubation; Scale bar = 100 μm. **h** Flow cytometry histogram and quantification of internalized PFeCy5 or SK@PFeCy5 in hVICs; $n = 3$ biologically independent samples; one-way ANOVA and Tukey's multiple comparison test. Values are presented as mean ± SD. Source data of (**b**), (**c**), (**f**), and (**h**) are provided as a Source Data file.

differentiated hVICs (Fig. 3g, h). Additionally, the targeting ability of SK@PFeCy5 was further confirmed in female-derived osteogenically differentiated and myofibrogenically differentiated hVICs (Supplementary Fig. 3d, e).

Taken together, these results demonstrate that PAR2-ligand functionalization enhances the affinity of nanoparticles for osteogenically and myofibrogenically differentiated hVICs.

## Magnetic field-guided nanoparticle localization in hVICs

Within transmission electron microscopy (TEM) images (Fig. 4a), noticeable dark regions were visible in SK@PFeXCT nanoparticles compared with naked PLGA. The dark regions were attributed to the presence of MNPs, confirmed by annular dark-field scanning TEM (ADF-STEM) and element mapping (Supplementary Fig. 10). The magnetic responsive property was verified via magnetic hysteresis loop characterization (Fig. 4b).

To evaluate the magnetic field-guided cell-nanoparticle interactions, osteogenically differentiated hVICs were incubated with SK@PFeCy5 while magnets were placed below the Petri dish without covering the whole bottom. After 4 h of incubation, SK@PFeCy5 was colocalized with hVICs in a magnetic field-dependent manner; that is, the regions exposed to the magnet exhibited higher fluoresce from the SK@PFeCy5 (Fig. 4c, d). Herein, the addition of EMF promoted the sedimentation and accumulation of the nano-cargoes, facilitating the

contact between the cargo and hVICs seeded at the bottom of the Petri dish and promoting their internalization.

Furthermore, the magnetic targeting ability of SK@PFeCy5 for hVICs in a fluid environment was investigated. To this end, a parallel-plate flow chamber was designed to mimic the shape of aortic valves; the flow rate was set to a shear stress of 5 dyn cm$^{-2}$. The thickness of the plate was 3 mm, equivalent to the distance from the aortic root to the body surface in mouse (Fig. 4e). After 4 h of flow, a large number of SK@PFeCy5 were captured by the hVICs seeded in the simulated aortic sinus in the presence of a magnetic field, while negligible Cy5 fluorescence was observed in hVICs in the absence of EMF navigation (Fig. 4f). These results confirmed the magnetic navigating ability of the SK@PFe nanoparticles.

## Nanotherapy decreases hVIC osteodifferentiation

The curative effect of SK@PFeXCT was further evaluated. First, the drug-loading capacity of SK@PFeXCT was assessed and optimized. Different concentrations of XCT790 were added to the oil phase during the synthesis process, and the loading of XCT790 in the nanoparticles was examined by high-performance liquid chromatography (HPLC). As the amount of XCT790 increased from 5 mg mL$^{-1}$ (concentration of XCT790 in the organic phase) to 15 mg mL$^{-1}$, the XCT790 loading content in nanoparticles increased significantly from $20.80 \pm 0.3274\%$ to $36.33 \pm 1.544\%$, while the encapsulation efficiency

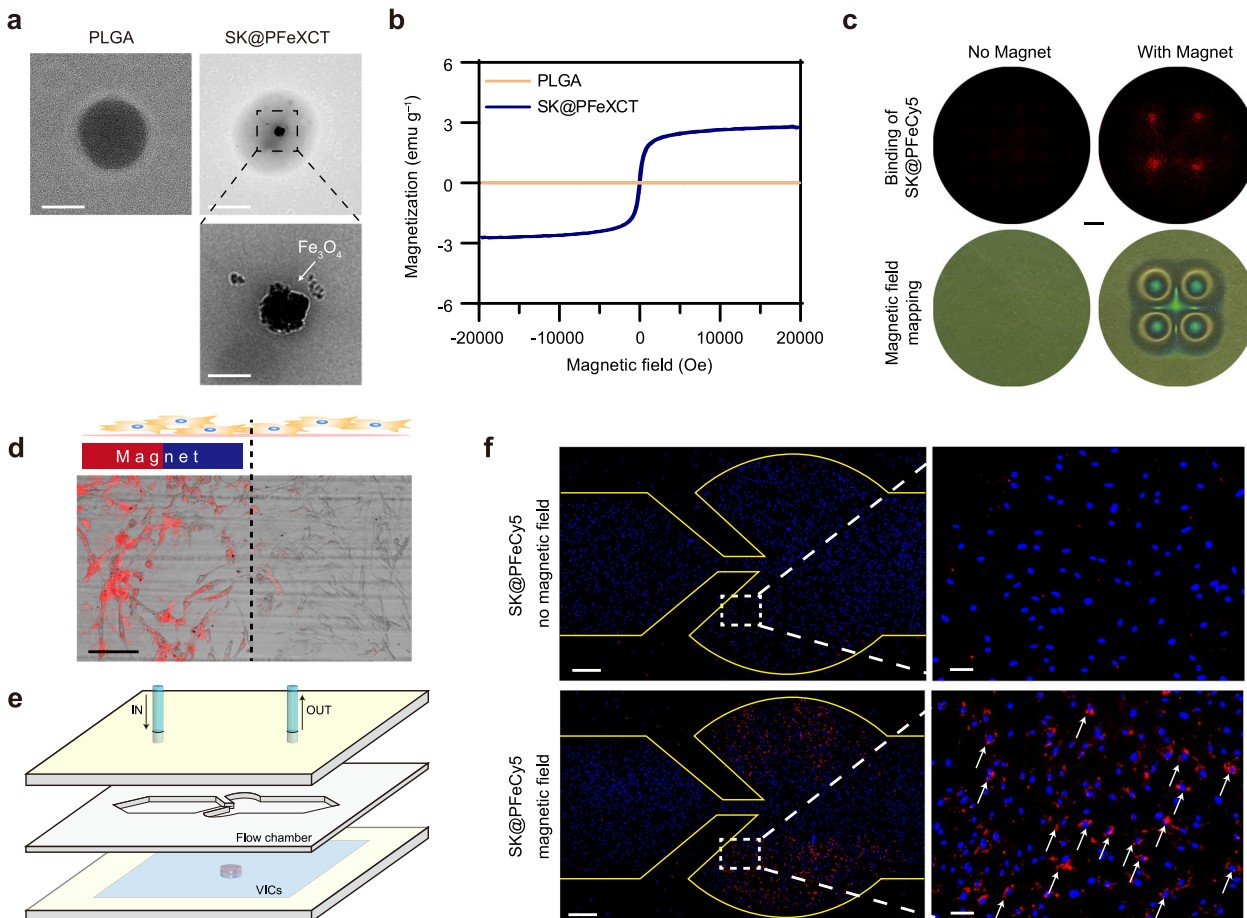

**Fig. 4 | Magnetic targeting ability of SK@PFe in vitro. a** TEM images of PLGA and SK@PFeXCT (upper; scale bar = 100 nm), and enlarged images of SK@PFeXCT (lower; scale bar = 25 nm); experiment was repeated independently for 3 times. **b** Magnetic hysteresis loop for PLGA and SK@PFeXCT at room temperature. **c** Cellular internalization of SK@PFeCy5 with or without an external magnetic field; Scale bar = 2 mm; experiment was repeated independently for 3 times.

**d** Fluorescent and bright field images of SK@PFeCy5 cellular uptake within or near the magnetic field; Scale bar = 200 μm; experiment was repeated independently for 3 times. **e** Schematic of the aortic valve mold in a microfluidic device. **f** Fluorescent images of SK@PFeCy5 cellular uptake under flow in the aortic valve mold with or without a magnetic field in full size (left; scale bar = 1 mm) and enlarged images (right; scale bar = 100 μm); experiment was repeated independently for 3 times.

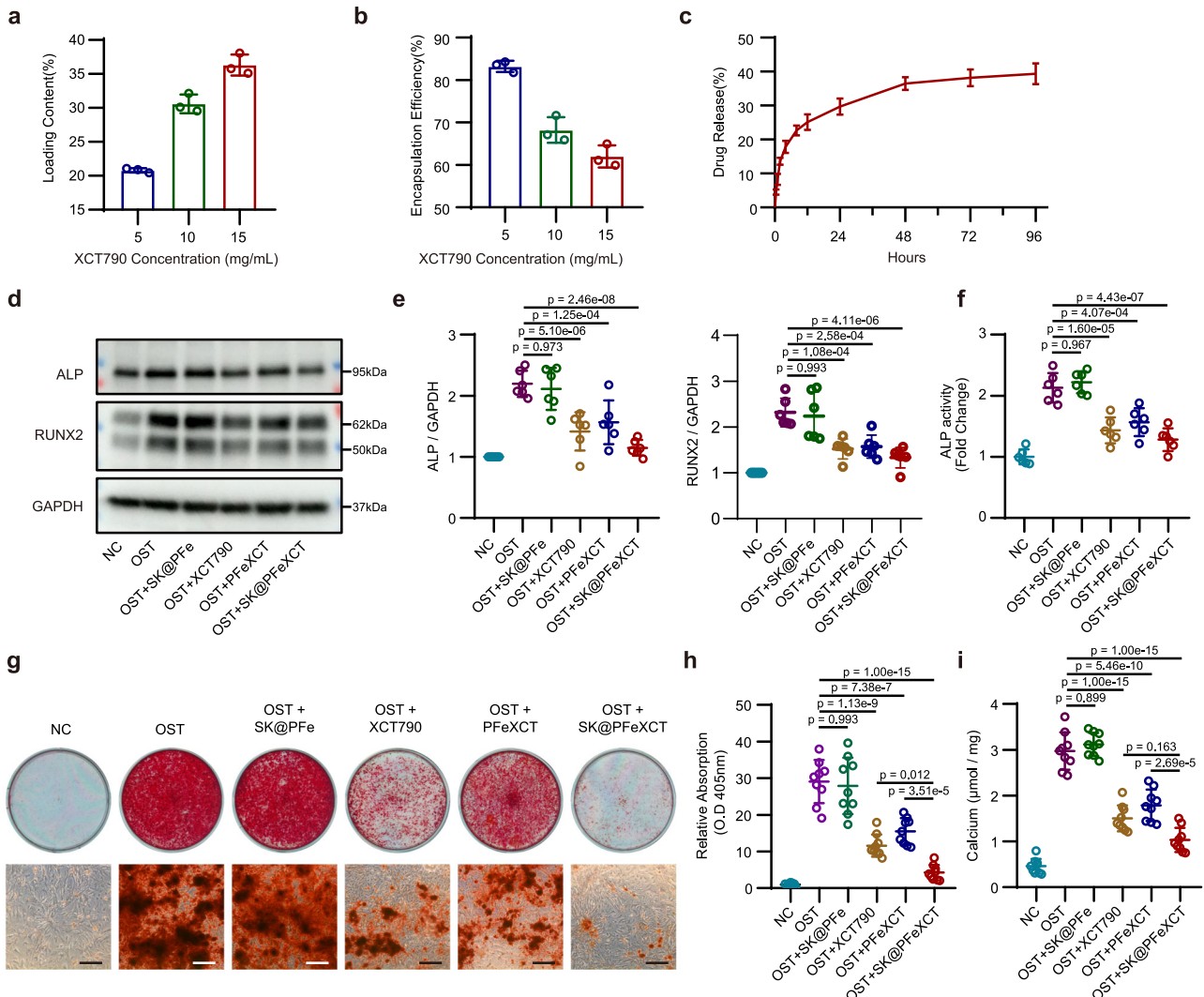

**Fig. 5 | SK@PFe functions as an effective cargo for XCT790 and enhances the suppressing effect of XCT790 on hVIC osteogenic differentiation.** a Loading contents (LC%) and (b) Encapsulation efficiencies (EE%) of XCT790 in SK@PFeXCT; $n = 3$ independent experiments. c Drug release profile of XCT790 in SK@PFeXCT under 37 °C; $n = 3$ independent experiments. d Representative western blot analysis and (e) quantification of osteogenic differentiation-related proteins (ALP, RUNX2) in hVICs; $n = 6$ biologically independent samples; one-way ANOVA and Tukey's multiple comparison test. f ALP activity of hVICs subjected to different treatments, normalized to the total protein content of hVICs, $n = 6$ biologically independent samples; one-way ANOVA and Tukey's multiple comparison test. g Representative Alizarin Red staining of hVICs treated with different formulations for 28 days. Upper panel, digital photos; lower panel, microscopic images of hVICs; Scale bar = 200 μm. h Quantification of Alizarin Red staining; $n = 9$ biologically independent samples; one-way ANOVA and Tukey's multiple comparison test. i Calcium deposition quantification in hVICs cultured with different formulations for 14 days; $n = 9$ biologically independent samples; one-way ANOVA and Tukey's multiple comparison test. Values are presented as mean ± SD. Source data of (a), (b), (c), (e), (f), (h), and (i) are provided as a Source Data file.

decreased from 82.31 ± 1.313% to 62.00 ± 2.634% (Fig. 5a, b). Hence, an XCT790 concentration of 15 mg mL$^{-1}$ was selected for subsequent experiments. Moreover, the release profiles of XCT790 revealed a burst within the first 10 h, which may be due to the release of XCT790 attached to the surface or encapsulated in the superficial layer of the nanoparticles. In the second phase, drug release decreased significantly, likely because the drug release in this stage is regulated primarily by the concentration gradient of XCT790 and nanoparticle degradation[39]. Approximately 40% of XCT790 was released throughout the experimental observation period (Fig. 5c).

The osteogenic differentiation medium-cultured hVICs were further treated with SK@PFe (20 μg mL$^{-1}$), XCT790 (6 μg mL$^{-1}$), or SK@PFeXCT (20 μg mL$^{-1}$). The effective XCT790 concentration was chosen according to our previous research and the concentration of nanoparticles was determined by balancing the XCT790 dosage among groups. The protein expression of the osteogenic markers, alkaline phosphatase (ALP) and runt-related transcription factor 2

(RUNX2), was significantly decreased in the presence of XCT790 or SK@PFeXCT; however, it was impacted by SK@PFe treatment. Although PFeXCT reduced the protein expression of ALP and RUNX2, the effect was significantly weaker than that of XCT790 or SK@PFeXCT, which might be due to the poor internalization and slow drug release of PFeXCT, resulting in lower intracellular XCT790 concentrations in osteoblast-like differentiated hVICs (Fig. 5d, e). Similar results were observed in the ALP activity assay (Fig. 5f).

Given that the primary pathological feature of CAVD is the formation of calcium nodules, we further evaluated the effect of SK@PFeXCT on reducing the extracellular calcium deposition of hVICs. To this end, hVICs were treated with an osteogenic differentiation medium supplemented with SK@PFe, XCT790, or SK@PFeXCT for 3 weeks, during which the medium was refreshed every 4 days. Alizarin Red staining showed that massive calcium nodules were deposited in the OST group, whereas SK@PFe did not affect calcium deposition. Meanwhile, XCT790 significantly

reduced the mineralization of hVICs, while PFeXCT had a weaker effect, which agreed with the western blot results. Notably, SK@PFeXCT exhibited the strongest inhibitory effect on calcium deposition (Fig. 5g, h), possibly due to cell internalization after binding to PAR2 and the sustained intracellular release of XCT790, leading to higher intracellular XCT790 concentration. The calcium deposit results were confirmed by absolute calcium quantification (Fig. 5i). These data indicate that SK@PFeXCT exhibits a superior ability to alleviate the osteogenic differentiation of hVICs.

**Nanoparticles are targeted to calcified aortic valves in mice**

A mouse aortic valve calcification model was employed to evaluate the ability of the nanoparticle platform to target the aortic valve. Due to the difference in PAR2 sequences between human and mice, the PAR2 peptide sequence was altered: SLIGRL-NH$_2$ for humans and SLIGKV-NH$_2$ for mice[19]. The targeting ability of SLIGRL-NH2 functionalized nanoparticles (SR@PFeCy5) and SLIGKV-NH2 functionalized nanoparticles (SK@PFeCy5) were verified by incubating with hVICs and the mouse vascular smooth muscle cell (VSMC) line MOVAS-1, respectively. As expected, MOVAS-1 cells exhibited a stronger affinity for SR@PFeCy5 than SK@PFeCy5, as evidenced by a higher accumulation of Cy5 fluorescence signal in MOVAS-1 cells incubated with SR@PFeCy5 (Supplementary Fig. 11).

To detect the targeting capabilities and systematically evaluate the in vivo distribution of the nanoplatform, IR780-labeled nanoparticles (PFeIR and SR@PFeIR) were synthesized, and a mouse CAVD model was established in HFD-fed $Ldlr^{-/-}$ mice. PFeIR and SR@PFeIR were injected intravenously, and a magnet with a diameter of 5 mm was placed on the mouse's chest above the aortic root for 15 min immediately after injection. Mice were euthanized 6 h after injection, and the major organs were harvested for fluorescence intensity detection (Fig. 6a). The distribution of the nanoplatform in the heart, aorta, liver, kidneys, lungs, and spleen was observed (Supplementary Fig. 12), with a particular focus on the heart and aorta (Fig. 6b). In the aortic root isolated from the ND-fed $Ldlr^{-/-}$ mice, a slightly enhanced fluorescence signal was observed in the SR@PFeIR EMF group compared with the PFeIR without EMF group. In the HFD-fed mice, grafting of PAR2 ligand and adding EMF enhanced the targeting ability of bare PLGA; this dual functionalization synergistically enhanced the targeting ability. (Fig. 6b, c). These results demonstrated the targeting capability of SR@PFeIR under EMF in a CAVD mouse model.

To specifically examine the ability of the SR@Pfe nanoplatform in targeting lesioned aortic valves at different disease stages, $Ldlr^{-/-}$ mice fed with HFD for various time durations were administered with Cy5-labeled SR@PFe (SR@PFeCy5); hearts were subsequently collected and sectioned for analysis. In the aortic valve of ND-fed mice, regardless of the type of nanoparticles injected, relatively no fluorescence

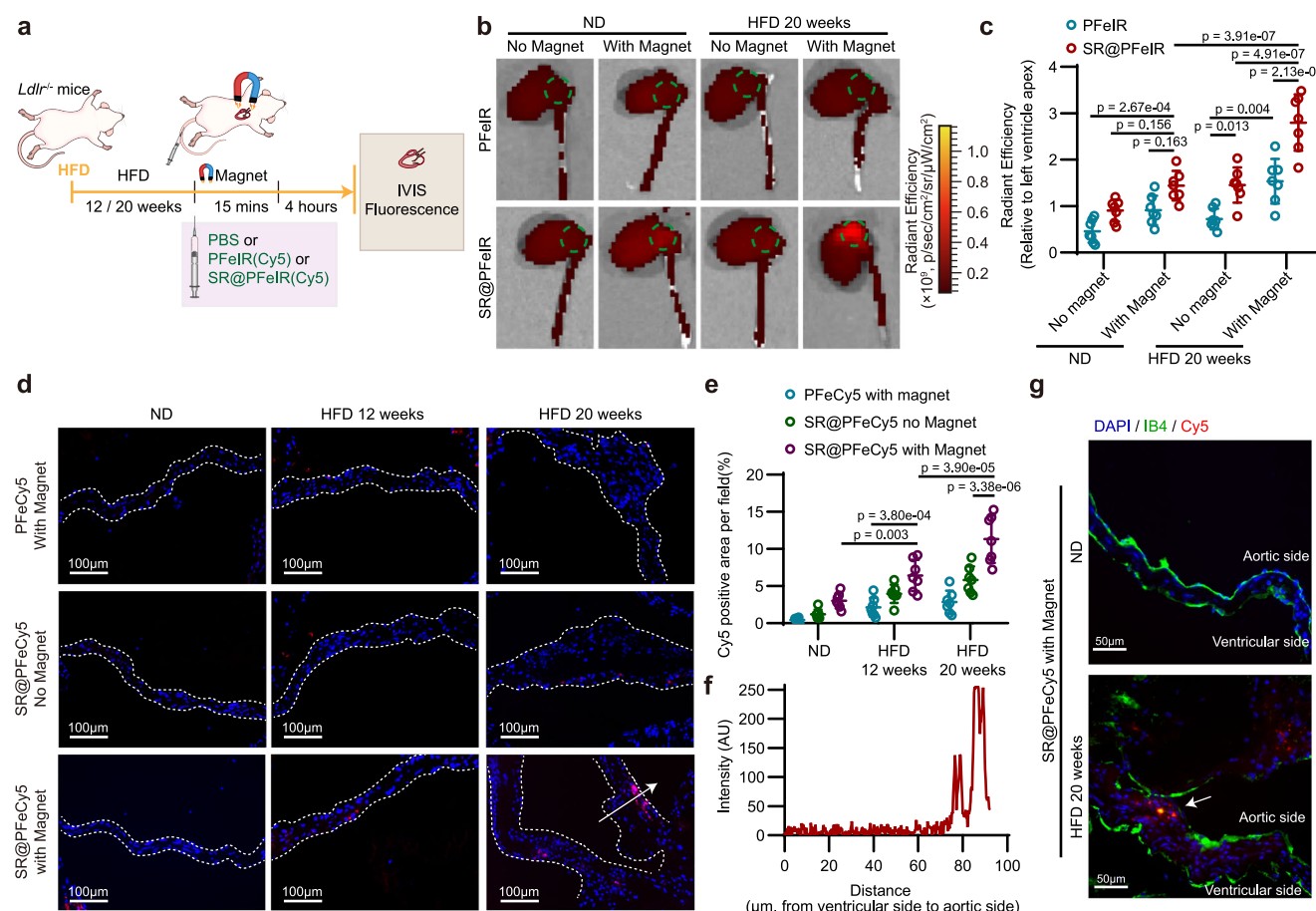

**Fig. 6 | Targeting ability of SR@ PFe in CAVD mice. a** Experimental outline. **b** Ex vivo IVIS images of $Ldlr^{-/-}$ mouse heart and aorta 4 h after intravenous injection of nanoparticles. Green circles indicate aortic roots. **c** Quantification of fluorescence intensity in the aortic root area (green circles) relative to the left ventricle apex of the heart; $n = 7$ biologically independent animals; two-way ANOVA and Tukey's multiple comparison test. (**d**) Fluorescence images of aortic valves of HFD fed $Ldlr^{-/-}$ mice 4 h after injected with nanoparticles. **e** Quantification of Cy5 mean

fluorescence in the aortic valves; $n = 7$ biologically independent animals; two-way ANOVA and Tukey's multiple comparison test. **f** Fluorescence intensity distribution in the aortic valve (white arrow in the image on the left). **g** Co-labeled isolectin B4 and Cy5; Scale bar = 50 μm; experiment was repeated independently for 3 times. Values are presented as mean ± SD. Source data of (**c**), (**e**), and (**f**) are provided as a Source Data file.

was observed, indicating that SR@PFe were not enriched in non-diseased leaflets (Fig. 6d, e). Under EMF navigation, few SR@PFe were enriched in the lesion leaflets of mice fed with HFD for 3 months, while the mice fed with HFD for 5 months had more nanoparticles on the lesion leaflets (Fig. 6d, e). The fluorescence intensity of SR@PFe in the EMF group was higher than that of PAR2-targeting alone or EMF alone, indicating that the dual-targeting worked synergistically (Fig. 6d, e). Notably, most of the SR@PFeCy5 was accumulated in the aortic side of the aortic valve leaflets, while a negligible amount was observed in the ventricular side (Fig. 6f), possibly due to a vortex area with lower flow velocity on the aortic side and the hierarchical expression of PAR2. Additionally, considering that endothelial denudation occurs with valve calcification, we speculate that SR@PFe may enter the sub-endothelial layer through gaps between the endothelium. Immuno-fluorescence confirmed that isolectin B4 was not continuous in the calcified leaflets, and SR@PFeCy5 aggregated in the areas of endo-thelial defects (Fig. 6g). Moreover, fluorescence was detected in the atherosclerotic aortic sinus, which agreed with the higher PAR2 expression in this region[23]. As expected, copious amounts of fluores-cence were detected in the aortic sinus of the HFD-fed $Ldlr^{-/-}$ mice, and more SR@PFeCy5 nanoparticles were detected in the aortic root under magnetic navigation when compared with the single-targeting control groups (Supplementary Fig. 13).

The targeting capacity of the SR@PFe nanoplatform was further confirmed in a wire injury model. Significantly increased fluorescence intensity was observed on the thickened aortic valve of the SR@PFeCy5 group compared with the PFeCy5 group (Supplementary Fig. 14).

These results demonstrate that magnetic-responded nano-particles functionalized with SLIGRL-NH$_2$ have an active targeting capacity to calcified aortic valves in the mouse CAVD model.

### Nanotherapy alleviates aortic valve calcification in mice

Next, the curative effect of SR@PFeXCT was assessed in the CAVD mouse model. After 20 weeks of HFD feeding, initial plaques appeared on the aortic valve of $Ldlr^{-/-}$ mice[40]. $Ldlr^{-/-}$ mice were injected with SR@PFe or SR@PFeXCT intravenously once per week (600 μg/mouse) or with XCT790 intraperitoneally (200 μg/mouse) twice per week (Fig. 7a). Following 8 weeks of treatment, echocardiograms were obtained to assess the cardiac function and aortic stenosis. Motion mode showed no significant differences in cardiac function between the groups (Supplementary Fig. 15). Meanwhile, pulsed-wave Doppler showed that compared with ND-fed mice, the peak transvalvular jet velocity and peak transvalvular pressure gradient were significantly higher in the HFD group, indicating that an aortic valve stenosis model was successfully established (Fig. 7b–d). Twice-weekly intraperitoneal XCT790 injections significantly alleviated aortic stenosis, and the weekly injections of SR@PFeXCT significantly delayed valve stenosis progression, while SR@PFe treatment did not alter these parameters (Fig. 7b–d).

Transformation in the valve leaflet morphology was assessed by H&E staining. After 28 weeks of HFD feeding, the aortic valves of $Ldlr^{-/-}$ mice were markedly thickened, while SR@PFeXCT injections visibly inhibited this thickening (Fig. 7e, f). Given that leaflet thickening was associated with tissue hyperplasia and fibrosis remodeling, Masson's trichrome staining was carried out to assess the collagen content in the valve leaflets[41]. SR@PFeXCT and XCT790 reduced the relative collagen area in the leaflets of HFD-fed mice (Fig. 7g, h). Sirius red staining further confirmed that the type I collagen content in the leaflets was increased in response to HFD, while SR@PFeXCT and XCT790 partially reduced the type I collagen content (Fig. 7i, j). Calcium staining with von Kossa revealed noticeable calcium nodules in the thickened leaf-lets of HFD-fed $Ldlr^{-/-}$ mice, while SR@PFeXCT negated the HFD-induced calcium deposition (Fig. 7k, l). Notably, SR@PFeXCT exhibited a better therapeutic outcome in these parameters compared with

XCT790 alone, even with lower administration frequency, while SR@PFe did not elicit significant effects on valve thickening or calci-fication (Fig. 7e–l). Immunofluorescence staining of osteogenic dif-ferentiation markers was conducted to confirm further the capacity of SR@PFeXCT to inhibit VICs osteogenic reprogramming in vivo. Con-sistent with the calcium deposition results, the expression of RUNX2, osterix (OSX), and bone morphogenetic protein 2 (BMP2) were enhanced by HFD feeding and diminished by SR@PFeXCT or XCT790 treatment (Supplementary Fig. 16).

Metabolic factors, including blood glucose and lipid levels, can impact aortic valve calcification and VICs osteogenic differentiation[22,42]. Therefore, to rule out the influence of systemic metabolic factors on valve thickening and calcification, the body weights of mice were recorded, and metabolic parameters, including blood glucose, total triglycerides, total cholesterol, and low-density lipoprotein (LDL), were measured. Body weight gain occurred faster in HFD-fed $Ldlr^{-/-}$ mice than in ND-fed mice. No significant difference was observed in body weights between HFD mice administered nano-particles or PBS (Supplementary Fig. 17a). At 28 weeks of modeling, the blood glucose, total cholesterol, triglycerides, and LDL levels in the HFD group were significantly increased, while SR@PFeXCT application did not alter these metabolic parameters (Supplementary Fig. 17b–e). This suggests that SR@PFeXCT inhibits aortic valve calcification in a manner that is independent of systemic metabolic regulation.

Overall, these results demonstrate that the PAR2-targeted XCT790-loaded magnetic nanoparticles significantly inhibit the osteogenic differentiation of VICs, preventing the progression of aor-tic calcification in mice.

### Metabolic reprogramming mediates the anti-calcification effects of nanotherapies

The drug selected in this study, XCT790, is a specific inhibitor of ERRα and is widely considered a strong modulator of mitochondrial bio-genesis and glycolysis[43]. Moreover, glycolysis is enhanced in vascular calcification, and the metabolic intermediates may participate in the osteogenic phenotype reprogramming[44]. To further explore the potential mechanism of XCT790-loaded nanoparticles on CAVD alle-viation, their role in hVICs metabolic reprogramming was assessed. Osteogenically differentiated hVICs took up more glucose and pro-duced less ATP than normal hVICs; SK@PFeXCT790 abolished this alteration (Fig. 8a, b). Regarding metabolic intermediate contents, hVICs cultured with osteogenic medium produced and excreted more lactic acid, whereas SK@PFeXCT inhibited this effect (Fig. 8c, d). The extracellular acidification rate (ECAR) assay further confirmed that the basal glycolytic and glycolytic capacity was enhanced during osteo-genic differentiation, while SK@PFeXCT suppressed the metabolic reprogramming (Fig. 8e, f).

RNA-seq was performed on hVICs stimulated with SK@PFeXCT and osteogenic differentiation medium, and the genes involved in glycolysis were analyzed. In agreement with the ECAR results, the expression of *HIF1A*, *PDK4*, and *PDK3* was upregulated after osteogenic differentiation (Supplementary Fig. 18), while only *PDK4* was inhibited once stimulated with SK@PFeXCT (Fig. 8g). This result was subse-quently validated by qPCR. Following osteogenic differentiate induc-tion, the expression of *PDK4* mRNA increased by ~7-fold, while XCT790 significantly inhibited this effect (Fig. 8h). PDK4—a crucial mitochondrial matrix enzyme—participates in metabolic reprogram-ming during the osteogenic differentiation of VSMCs[44]. Meanwhile, ERRα is associated with the transcriptional activation of *PDK4*[45]. Wes-tern blot analysis confirmed that the protein expression of PDK4 was significantly increased following osteogenic induction and abolished by SK@PFeXCT treatment (Fig. 8i). Moreover, SR@PFeXCT injection significantly reduced PDK4 expression in the aortic valve of CAVD mice, as demonstrated by immunofluorescence staining (Fig. 8j). These results suggest that PDK4-mediated inhibition of metabolic

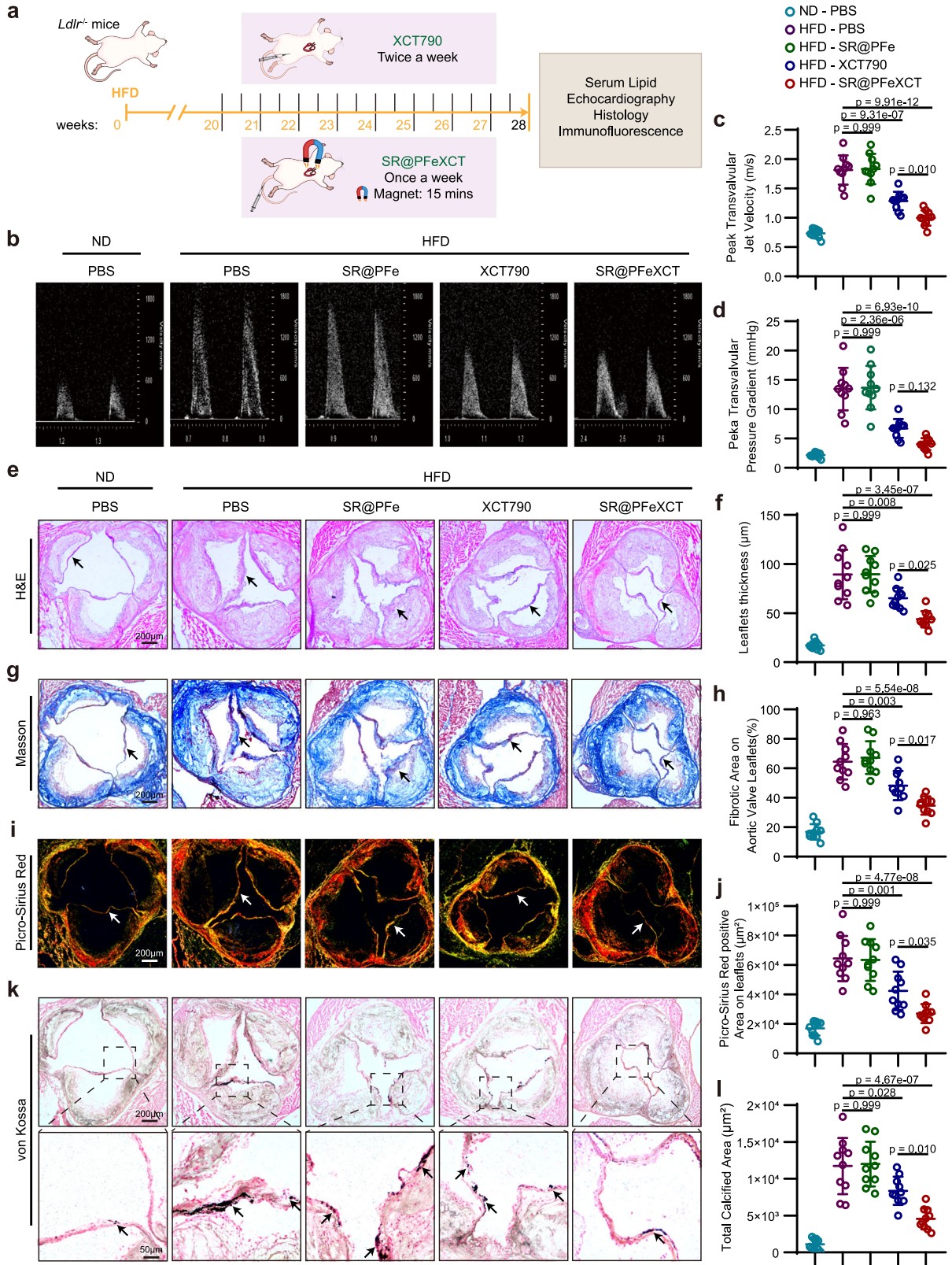

reprogramming is responsible for the curative effect of SR@PFeXCT in aortic valve calcification.

## Biosafety of SK@PFeXCT in vitro and SR@PFeXCT in vivo

The cytotoxicity induced by the nanoparticles was assessed in vitro using a CCK-8 assay. When the concentration of SK@PFeXCT was <50 µg mL$^{-1}$, no significant concentration-dependent cytotoxicity was observed in hVICs, while cell viability decreased at a concentration of 100 µg mL$^{-1}$, with a more pronounced decrease at 200 µg mL$^{-1}$, equal to 100 µM XCT790. Cell activity in hVICs incubated with 200 µg mL$^{-1}$ SK@PFe nanoparticles did not change significantly, indicating that the cytotoxicity of the nanocarrier at concentrations <200 µg mL$^{-1}$ was negligible, and that the decrease of cell viability in hVICs treated by 200 µg mL$^{-1}$ SK@PFeXCT (~100 µM of XCT790) was likely caused by

**Fig. 7 | SR@PFeXCT alleviates aortic valve calcification in mice. a** Experimental outline. **b** Representative echocardiographic images of *Ldlr*[-/-] mice. **c** Peak trans-valvular jet velocity of *Ldlr*[-/-] mice, n = 10 biologically independent animals. **d** Peak transvalvular pressure gradient of *Ldlr*[-/-] mice; n = 10 biologically independent animals. **e** Representative H&E staining of aortic valves. **f** Aortic valve leaflet thickness quantification; n = 10 biologically independent animals. **g** Aortic valves from *Ldlr*[-/-] mice stained with Masson's trichrome. **h** Fibrosis ratio of aortic valve leaflets quantified from Masson's staining; n = 10 biologically independent animals. **i** Picro-Sirius Red staining of aortic valves from *Ldlr*[-/-] mice, visualized under

polarized light. **j** Collagen deposition of the leaflets quantified from Picro-Sirius Red staining; n = 10 biologically independent animals. **k** Von Kossa staining of aortic valves at low magnification (upper rows) and high magnification (bottom rows). Black arrows indicate calcium nodules. (**l**) Calcium deposit area of the aortic valve quantified from von Kossa staining; n = 10 biologically independent animals. Values are presented as mean ± SD. Significance of differences determined by one-way ANOVA and Tukey's multiple comparison test. Source data of (**c**), (**d**), (**f**), (**h**), (**j**), and (**l**) are provided as a Source Data file.

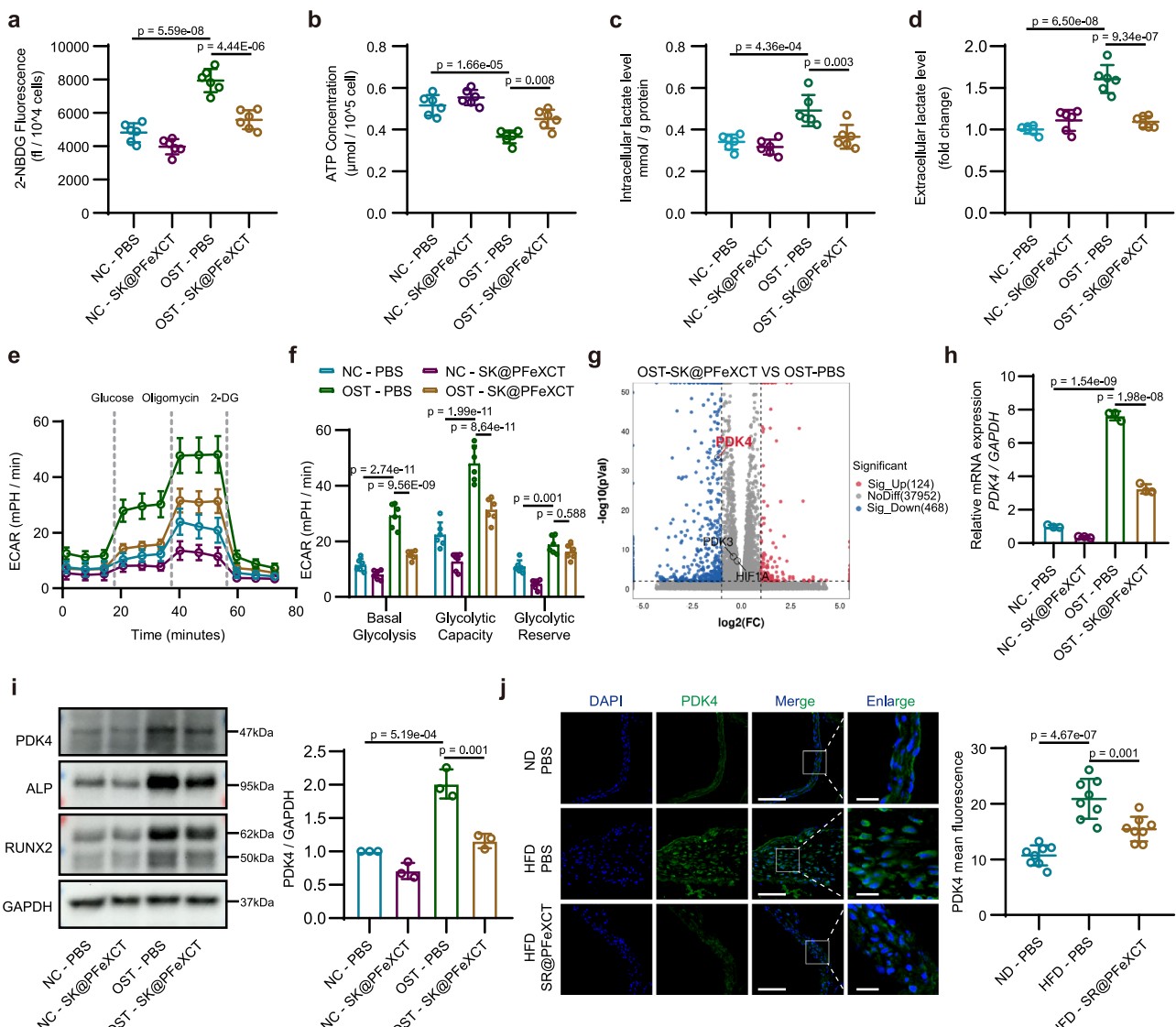

**Fig. 8 | SK@PFeXCT inhibits VICs metabolic reprogramming. a** Glucose uptake, (**b**) ATP generation, (**c**) intracellular lactate concentration, and (**d**) extracellular lactate concentration in hVICs following treatment; n = 6 biologically independent samples. **e** Real-time changes in the extracellular acidification rate (ECAR) in SK@PFeXCT-treated hVICs and (**f**) basal glycolysis, glycolytic capacity, and glyco-lytic reserves; n = 6 biologically independent samples. **g** Volcano plot of differential gene expression between osteogenically differentiated hVICs stimulated with or without SK@PFeXCT. *PDK4*, *PDK3*, and *HIF1A*. Red: up-regulated genes, blue: down-regulated genes, gray: not significant. Adjusted *p* value < 0.01, log2 Fold Change < −1 or > 1, n = 4 biologically independent samples. **h** mRNA expression of PDK4 in

quiescent hVICs and osteogenically differentiated hVICs stimulated with or without SK@PFeXCT, n = 3 biologically independent samples. **i** Protein expression of PDK4, ALP, and RUNX2 in hVICs following treatment; n = 3 biologically independent samples. **j** Immunofluorescence images and quantification of PDK4 in aortic valves from different groups of *Ldlr*[-/-] mice. Scale bar = 100 μm in the left and 20 μm in enlarged images; n = 8 biologically independent animals. Values are presented as mean ± SD. Significance of differences determined by one-way ANOVA and Tukey's multiple comparison test. Source data of (**a**), (**b**), (**c**), (**d**), (**e**), (**f**), (**h**), (**i**), and (**j**) are provided as a Source Data file.

XCT790 toxicity (Supplementary Fig. 19a). Furthermore, time-dependent cytotoxicity of SK@PFeXCT was examined in hVICs by Calcein-AM/PI staining. Under the stimulation of 20 µg mL$^{-1}$ SK@PFeXCT, no apparent cell death was observed on days 1, 3, or 7 (Supplementary Fig. 19b, c). Additionally, the cytotoxicity of SK@PFe in the magnetic field was assessed; no difference was observed in the number of PI-positive cells in the area with or without magnetic fields (Supplementary Fig. 19d). These results confirm the absence of cytotoxicity induced by the nano-cargoes.

Long-term high-dose iron stimulation produces intracellular oxidative stress, which can cause ferroptosis and facilitate the osteogenic differentiation of VICs[46]. Thus, intracellular reactive oxygen species (ROS) production and ferroptosis were evaluated in SK@PFe-treated hVICs. Flow cytometry results demonstrated that SK@PFe treatment did not induce ROS generation in hVICs (Supplementary Fig. 20a). DCFH staining confirmed that the intracellular ROS production caused by magnetic field-guided nanoparticle enrichment was negligible (Supplementary Fig. 20b). Additionally, the abundance of glutathione peroxidase 4 (GPX4)—an important regulator of ferroptosis—was not down-regulated by SK@PFe treatment (Supplementary Fig. 20c). Similarly, the content of malondialdehyde (MDA)—an indicator of lipid peroxidation and an end product of ferroptosis—was not affected by SK@PFe stimulation (Supplementary Fig. 20d). Therefore, the nano-cargo did not elicit apparent intracellular ROS accumulation or ferroptosis.

A hemolysis assay was conducted to determine whether in vivo application of SR@PFeXCT causes erythrocyte rupture. Different concentrations of SR@PFeXCT were incubated with a 2% erythrocyte suspension at 37 °C for 1 h; no significant hemoglobin dissolution was observed when compared with PBS-treated cells (Supplementary Fig. 21a). Subsequently, healthy mice were intravenously injected with PBS or SR@PFeXCT once per week for 2 weeks; the body weights were recorded every 3 days, blood samples were collected on days 3 and 15, and major organs were harvested for H&E staining on day 15. The body weight of mice treated with SR@PFeXCT did not differ from that of those injected with PBS (Supplementary Fig. 21b). Moreover, the serum tumor necrosis factor-α (TNF-α) and interleukin-6 (IL-6) concentrations were not elevated following nanoparticle injection (Supplementary Fig. 21c, d). Similarly, the blood biochemical parameters, including blood urea nitrogen (BUN), creatinine (Cre), aspartate transaminase (AST), and alanine transaminase (ALT), did not differ between the SR@PFeXCT and PBS groups (Supplementary Fig. 21e–h). Major organs, including the heart, liver, lungs, spleen, and kidneys, showed no notable pathological changes (Supplementary Fig. 21i). Collectively, these results demonstrate the biosafety of SR@PFeXCT for treating CAVD in vivo.

## Discussion

To design a suitable TDDS for the treatment of CAVD, we determined that PAR2 is highly expressed in osteogenically differentiated VICs, and synthesized a PAR2-ligand functionalized magnetic response nano-carrier to deliver XCT790 to the calcified aortic valves. Targets for TDDS must be selectively and highly expressed at intercellular and subcellular levels[5]. PAR2 is highly expressed in hVICs within the fibrosa layer of calcified leaflets, and is expressed at low levels by VECs. As a GPCR, PAR2 is primarily located within the plasma membrane, with a small amount distributed in the Golgi apparatus during the synthesis and sorting process and endosomes during endocytosis, showing extremely high subcellular localization specificity[31]. Moreover, targets shed from the cell membrane may competitively inhibit the binding of TDDS to diseased target cells (e.g., soluble CD30)[47]. However, PAR2 is not shed from cells and, thus, will not impact the targeting ability of the TDDS (Supplementary Fig. 22). Furthermore, PAR2 is internalized after binding its ligands and routed to lysosomes, which may promote the internalization and drug release of PAR2-based TDDS[20]. Finally,

once PAR2 is degraded in lysosomes after being endocytosed, it is stored in the Golgi apparatus; following de novo synthesis, it is rapidly mobilized to the plasma membrane for repopulation[48].

Various animal models have been used to study the occurrence of CAVD, including a transgenic aging model, hyperlipidemia model, guidewire injury model, and the recently published hyperlipidemia-based wire injury model, etc[40,41,49,50]. Ldlr$^{-/-}$ mouse fed with HFD for 6 months is a classic mouse model of aortic valve calcification, which simulates the process of valve calcification caused by non-genetic factors, including lipid deposition, inflammatory cell invasion, plaque formation, and thickening and calcification[51]. The wire injury (WI) model was generated by inserting and rotation a guidewire to cause mechanical damages on the endothelium. The damages further induced infiltration of blood cells, inflammation, and valve thickening[52]. The WI model was applied on wild type mice, which have normal blood lipid level, so the pathological process was different from the Ldlr$^{-/-}$ mouse fed with HFD. Moreover, the damage degree can be controlled by operation during wire injury, which lead to higher repeatability of the modeling. In this study, we used two mouse models with different pathological mechanisms to study the expression of PAR2. The results from both of the models verified the reliability of PAR2 elevation in CAVD.

Network-based integration methods are applied in the analysis of CAVD-derived multi-omics and have become an important tool for better understanding the molecular regulatory networks of CAVD initiation, pathogenesis, and treatment[53,54]. By analyzing the gene regulatory networks via RNA-seq, NOTCH1 haploinsufficiency was identified as a potential risk factor for BAV and CAVD[55]. Moreover, network analysis of the sexually dimorphic transcriptome of porcine VICs identified two X-chromosome inactivation escape genes (BMX nonreceptor tyrosine kinase (BMX) and steroid sulfatase (STS)), which may explain the effect of sex on myofibroblast activation in VICs[33]. Furthermore, combining spatiotemporal transcriptomics with protein–protein interaction networks revealed that fibronectin-1 and protease inhibitor alpha-2-macroglobulin are associated with CAVD progression[56]. Meanwhile, the development of single-cell RNA sequencing (scRNA-seq) has enabled the study of VICs heterogeneity and the identification of three VICs subtypes[35]. Combining scRNA-seq and network analysis led to the isolation of a disease-driver VICs population with procalcific potential from human CAVD tissue, while temporal proteomic profiling identified monoamine oxidase-A (MAOA) and collagen triple helix repeat containing-1 (CTHRC1) as potential therapeutic targets[57]. Extracellular vesicles serve as major messengers in intercellular communication, greatly influencing disease progression[58]. The miRNA–mRNA target network facilitates the delivery of miRNAs by extracellular vesicles in valve tissue to targets in recipient cells. Integrated analysis has been employed to predict the cumulative effects of extracellular vesicle cargo. Ultimately, 62 altered miRNAs were screened and 1813 target genes were predicted. KEGG pathway enrichment and siRNA-mediated knockdown experiments verified that WNT5A (Wnt family member 5A), APP (amyloid beta precursor protein), and APC (adenomatous polyposis coli WNT signaling regulator) are regulators that promote CAVD progression[59]. Machine learning strategies combined with targeted RNA-seq have been used to map gene networks disrupted in human NOTCH1 haploinsufficient iPSC-derived endothelial cells; XCT790 was screened to correct the impaired gene network signature and prevent CAVD initiation and progression in Notch1/mTR$^G$2 mice[25]. With the emergence of novel sequencing technologies, such as single-molecule protein sequencing and single-cell metabolomics, network-guide tactics will become a more powerful tool to accelerate the study of CAVD pathogenesis and the discovery of novel drugs[60].

This study has certain limitations. First, other cell surface proteins may also be appropriate markers for nanoparticle targeting. The current findings are inferences and validation based on transcriptome results

from osteogenic differentiated VICs and then identified in pathologic valve tissues. If novel omics methods are applied to analyze the proteome or transcriptome of cells directly in valve tissues, some other membrane proteins may become candidate targets. Furthermore, in addition to membrane proteins, extracellular matrix proteins, as the main protein components of calcified thickened valves, may also be an effective target for CAVD TDDSs[61]. By combining multi-omics analysis with and network-guide tactics, more targeting anchors may be able to identified and compared with higher throughput and efficiency. However, as an exploratory study, our results sufficiently demonstrate that PAR2 is elevated in aortic valve calcification and represents an effective anchor for targeting drug delivery. Second, the optimal application doses of the synthesized nanoparticles were not confirmed via rigorous pharmacological and metabolic testing. Although no significant systemic toxicity was observed in vitro or in vivo (Supplementary Figs. 19–21), further studies are essential to optimize drug dosage. Third, a preclinical drug was used in this study as the model drug for CAVD therapy as no clinical drugs have been reported. Hence, the systemic toxicology and adverse effects on other organs require further assessment. Here, we preliminarily assessed the side effects of XCT790 in HFD-induced liver lesion, as it has been previously reported that liver-specific ERRα-deficient affects the development of liver steatosis and fibrosis, which may lead to non-alcoholic fatty liver disease (NAFLD)[26]. Oil red O staining showed that, when compared to the HFD group, steatosis is slightly increased in response to XCT790, while there was no significant increase in the SR@PFeXCT group (Supplementary Fig. 23a, b). QPCR confirmed that the adipogenic-related genes, ATP-citrate lyase (ACLY) and fatty acid synthase (FASN), were up-regulated in the XCT790 group (Supplementary Fig. 23c). However, no difference was found in collagen deposition, as indicated by Picro-Sirius Red staining, or profibrotic gene transcription, as evidenced by qPCR analysis (Supplementary Fig. 23d, e). Nevertheless, more in-depth studies on XCT790 metabolism and side effects are needed before XCT790 enters clinical trials. With the assistance of our CAVD-targeting drug delivery system, the dosage and frequency of drug administration can be significantly reduced, which may decrease the potential systemic toxicity without impacting the curative effect.

In this study, we identified a cell surface anchoring site for calcified VICs (i.e., PAR2) and devised a targeting nanocarrier with a PAR2-binding peptide shell as the anchor and a MNPs core in PLGA as the navigator. This system achieved enhanced intracellular drug delivery within calcified aortic valves. Hence, this study proposes the design of a drug delivery system with dual targeting capabilities, which may provide insights for the design and translation of precision therapies for CAVD.

## Methods

### Study approvals
All experiments involving humans were conducted in accordance with the Declaration of Helsinki and were approved by the Human Research Ethics Committee of the Second Affiliated Hospital, Zhejiang University (No. IRB-2014-061). All animal experiments were performed in accordance with the Guide for the Care and Use of Laboratory Animals published by the NIH and approved by the Animal Use Committee of The Second Affiliated Hospital, Zhejiang University (No. AIRB-2022-0109).

### Human valve leaflets collection
Calcified aortic valve leaflets were obtained from patients with severe aortic stenosis who underwent aortic valve replacement surgery, while non-calcified aortic valves were obtained from heart transplant recipients, patients with Stanford type A acute aortic dissection or aortic valve regurgitation. Valves from patients with severe aortic regurgitation, rheumatic disease, infective endocarditis, congenital valvular disease, or diabetes were excluded (Supplementary Fig. 24). Written

informed consent was provided by each participant. Leaflets were divided into four pieces immediately after acquisition: (1) digested with collagenase type II for valvular interstitial cell isolation and culture. (2) homogenized in radioimmunoprecipitation assay (RIPA) buffer (cat #P0013B, Beyotime) for protein extraction and immunoblot; (3) embedded in paraffin for immunohistochemistry staining, Masson staining, and Alizarin Red staining; (4) embedded in optimal cutting temperature compound (O.C.T. compound, cat #4583, Sakura Finetek, Japan) for immunofluorescence staining, H&E staining, and Alizarin Red staining.

### Isolation, culture, osteogenic differentiation of hVICs
Isolation and culture of hVICs were performed as previously described[62]. Briefly, non-calcified aortic valve leaflets were washed with PBS twice, minced into 1 mm$^3$ pieces, and digested with collagenase type II (2 mg mL$^{-1}$, in DMEM, cat #17101015, Gibco) for 4 h at 37 °C. The tissue solution was further dissociated by pipetting, and precipitated by centrifugation at $200 \times g$ for 5 min. Isolated hVICs were resuspended and cultured in high-glucose DMEM supplemented with 10% FBS and 1% penicillin/streptomycin. For the isolation of hVICs derived from calcified leaflets, calcium nodules in the leaflets were removed with forceps, and valves were washed twice with penicillin/streptomycin-containing PBS, minced, digested, and used for subsequent cultures. All hVICs used in in vitro experiments were from passages 2–5.

For osteogenic differentiation of hVICs, cells were cultured with osteogenic induction medium (complete medium supplemented with 10 mM β-glycerophosphate, 0.25 mM ascorbic acid, and 1 mM dexamethasone) for various durations. The abundances of RUNX2, ALP, PAR2, and PDK4 proteins were examined by western blotting after 4 days of osteogenic induction. Alkaline phosphatase activity was detected using the ALP Assay Kit (cat #P0321M, Beyotime) on day 4 post-osteogenic induction according to the manufacturer's instructions. Twenty-one days post-osteogenic induction, extracellular calcium deposition was quantified using the Calcium Colorimetric Assay Kit (cat #C004-2-1, Jiancheng, Nanjing, China) and Alizarin Red staining (cat #A5533, Sigma, USA) according to the manufacturer's instructions.

### Western blot analysis
Aortic valve leaflets were collected and stored at −80 °C. For protein extraction, after removing large calcified nodules, leaflets were minced and homogenized in cold RIPA buffer containing PMSF using a tissue grinder (Tissuelyser, Jingxin, Shanghai). Protein concentration was determined with the BCA protein assay (cat #A55860, Thermo Fisher). For western blotting, equal quantities of proteins were separated using 10% sodium dodecyl sulfate-polyacrylamide gel electrophoresis (SDS-PAGE) and transferred to PVDF membranes. After blocking with skim milk, proteins were detected with specific primary antibodies followed by HRP-linked secondary antibodies. Finally, the blots were developed using the enhanced chemiluminescence reagent (cat #E411-04, Vazyme, Nanjing, China) with ImageQuant 800 (Amersham, GE healthcare); the bands were quantified by ImageQuant TL (Version 8.2, Amersham). The following primary antibodies were used at 1:1000 dilution otherwise specified: RUNX2 (cat #ET1612-47, Huabio, Hangzhou, China); ALP (cat #MAB1448, R&D); PAR2 (cat #ab180953, Abcam); PDK4 (cat #12949-1-AP, Proteintech); COL1A1 (cat #PA5-86949, Invitrogen); Fibronectin (cat #ab268020, Abcam); GPX4 (cat #ab125066, Abcam); GAPDH (cat #ET1702-66, Huabio, 1:5000); β-Actin (cat #ET1702-67, Huabio, 1:5000) and β-Tubulin (cat #ET1702-68, Huabio, 1:5000).

### Immunofluorescence staining
Immunofluorescence staining was performed as previously reported[63]. In brief, tissues were cryopreserved in 30% sucrose overnight, frozen in O.C.T. compound, and sectioned at 6 μm thickness. After being dried

at room temperature for 15 min, sections were fixed in 4% paraformaldehyde for 10 min, permeabilized with 0.2% Triton X-100 for 10 min, blocked with 3% BSA for 30 min, and incubated with specific primary antibodies overnight at 4 °C. They were subsequently incubated with fluorescently conjugated secondary antibody for 1 h at room temperature. Nuclei were stained with a mounting medium containing DAPI (cat #H-1200, Vector). The same concentration of rabbit isotype control was used as a negative control. Analyses were performed with the Image Pro Plus software (Version 6.0). The following primary antibodies were used: PAR2 (cat #35-2300, Invitrogen, 1:200) and PDK4 (cat #12949-1-AP, Proteintech, 1:100); α-smooth muscle actin (cat #ab7817, Abcam, 1:200); VE-Cadherin (cat #2500 T, CST, 1:400); Vimentin (cat #ab11256, Abcam, 1:200); CD31 (cat #11265-1-AP, Proteintech, 1:200); Osterix (cat #ab209484, Abcam, 1:300) and BMP2 (cat #ab284387, Abcam, 1:100).

## Immunohistochemistry staining
Immunohistochemistry staining was applied to detect PAR2 expression in aortic valve leaflets. In brief, tissues were fixed in paraformaldehyde, dehydrated in graded ethanol, embedded in paraffin, and sectioned at a 3.5 μm thickness. After dewaxing, rehydration, antigen retrieval, and blocking, sections were incubated with a primary antibody against PAR2 (Invitrogen, 1:200) overnight at 4 °C. They were then incubated with an HRP-conjugated secondary antibody (DAB, cat #PV-9000, zsbio, Beijing, China) and visualized with 3,3-diaminobenzidine (DAB, cat #ZLI-9017, zsbio, Beijing, China). Images were captured by a DM3000 microscope (Leica, Germany) with LAS X software (Version 3.0, Leica).

## Masson's trichrome staining
Masson's trichrome staining was performed with a Masson staining kit (cat #G1340, Solarbio) according to the manufacturer's instructions. After 10-min fixation in acetone-methanol, the sections were dipped in Weigert iron hematoxylin solution for 5 min. Subsequently, sections were rinsed in tap water for 5 min and incubated in a solution containing 0.04% acidic fuchsia and 0.02% azophloxine for 20 min before rinsing with a 1% acetic acid solution. Next, the sections were placed in a 2.5% phosphotungstic acid solution for 10 min and rinsed again with a 1% acetic acid solution. Finally, sections were dehydrated in ethanol and xylene, placed in a toluene solution, and mounted with a quick-hardening mounting medium (cat #03989, Sigma-Aldrich). The fractional areas of the collagen components (blue) in the mouse aortic valve region were obtained using a DM3000 microscope; ten randomly selected microscopic fields per specimen were analyzed.

## Alizarin Red staining
The human aortic valve leaflet was dewaxed and rinsed in 1 × PBS; sections were stained with 1% Alizarin Red stain (Sigma) for 1 h, washed with water, mounted using xylene-based mounting medium, and imaged.

Staining of calcium nodules generated by osteogenically differentiated hVICs was performed by fixing cells in PFA and staining with 1% Alizarin Red stain for 10 min, washing in water, and imaging with a microscope. To quantify the cells, they were washed with 10% acetate at room temperature for 30 min, and the absorbance was determined at 405 nm.

## RNA interference
Cells were seeded into plates and grown to 60–70% confluence. Small interfering RNA targeting human PAR2 (siPAR2) and scramble siRNA (siScr) were transfected into hVICs with RNAiMAX reagent (cat #13778, Thermo Fisher, CA, USA). Knock-down efficiency was assessed via western blotting. The siRNA sequences are listed in Supplementary Table 3.

## Quantitative polymerase chain reaction
To analyze mRNA expression in hVICs, real-time q-PCR was conducted. Briefly, total cellular mRNA was extracted with TRIzol (cat #15596026, Invitrogen, Carlsbad, CA) and quantified using a Nanodrop 2000 (Thermo Scientific). Reverse transcription was performed with PrimeScript RT Master Mix kit (cat #RR036, Takara, Japan) via incubation at 37 °C for 15 min followed by 85 °C for 15 s, and 4 °C for ∞. Real-time q-PCR was performed with TB Green Premix Ex Taq (cat #RR420A, Takara) using the LightCycler 480II (Roche, Switzerland) under the following program: pre-denaturation at 95 °C for 30 s, 40 cycles of 20 s of denaturation at 95 °C, 20 s annealing at 55 °C, 20 s of extension at 72 °C. Relative mRNA expression was calculated using the $2^{-\Delta\Delta Ct}$ method, and *GAPDH* served as the endogenous control. The primer sequences used for RT−qPCR are listed in Supplementary Table 4.

## Sequencing data collection and processing
Single-cell RNA-sequencing profile data (PRJNA562645) were downloaded from the GEO database of the NCBI. Data were integrated and normalized; the Seurat pipeline was used to classify cell clusters, while SingleR was used to identify cell types. The individual gene expression value was visualized using the DoHeatmap function of Seurat. Gene Set Enrichment Analyses were performed using the GSEA software, and the DiffusionMap function from Destiny was used to evaluate cell differentiation trajectory.

For bulk RNA-seq, total RNA from hVICs was sequenced on Illumina XTen by KaiTai biotechnology (Hangzhou, China); differential analysis was conducted using DESeq2, and significant genes were defined as FDR < 0.05. Heatmaps were generated using the OmicStudio tool at https://www.omicstudio.cn/tool; volcano plots were conducted using the ImageGP software.

## Materials for nano-carrier synthesize
Carboxyl-terminated Poly(lactic-co-glycolic acid) (PLGA, 75:25, MW: 40,000–60,000, cat #DG-DLGH75) was purchased from Daigang Biomaterials (Jinan, China); α,ω-Diformyl poly(ethylene glycol) (OHC-PEG-CHO, MW: 2,000, cat #YS-P12201) was purchased from Yusi Pharma (Chongqing, China); $Fe_3O_4$ nanoparticle (diameter: 8–10 nm, cat #103756) was purchased from XFNANO Materials (Nanjing, China); 3-(Trimethoxysilyl)propyl methacrylate (TPM, cat #S111153) was provided by Aladdin Biochemical Tech (Shanghai, China). BSA and IR-780 iodide were purchased from Merck, while Cyanine5 carboxylic acid (Cy5-COOH, cat #D10123) was provided by Duofluor (Wuhan, China). XCT790 was obtained from GlpBio (cat #GC10789, CA, USA). The PAR2 ligand peptide (SLIGKV-NH₂ and SLIGRL-NH₂) was synthesized by Synpeptides (Nanjing, China) while Rhodamine B labeled SLIGKV-NH₂ was synthesized by Allpeptide (Hangzhou, China), and Cy7-labeled BSA was synthesized by Qiyuebio (Xi'an, China).

## Preparation of SK@PFeXCT nanoparticles
PLGA nanoparticles loaded with MNPs and XCT790 (FeXCT) were prepared using an emulsion/solvent evaporation method as previously reported[39]. In brief, 40 mg of PLGA and 5 mg of XCT790 were dissolved in 2 mL of Dichloromethane (DCM, cat #D116149, Aladdin), followed by the addition of 2 mg of MNPs nanoparticles to formulate a dispersed system. This suspension was added to 8 mL of BSA (30 mg mL⁻¹, in deionized water) and sonicated for 3 min with a power of 40 W. The resulting oil-in-water emulsion was added dropwise into 40 mL of deionized water and stirred for 4 h. FeXCT nanoparticles were collected via centrifugation at 10,000 × g for 10 min and washed with ultra-pure water three times. The loading of Cy5-COOH or IR780 iodide was realized using the same procedures. For the preparation of PLGA nanoparticles without magnetic properties, the same methods were performed without MNPs.

OHC−PEG−CHO was utilized to hydrate FeXCT nanoparticles and introduce an aldehyde group onto the surface of FeXCT nanoparticles

to facilitate subsequent peptide modification. Briefly, the FeXCT nanoparticles were pipetted gently into 30 mL of OHC–PEG–CHO (1 mg mL$^{-1}$, in deionized water) and stirred for 4 h at room temperature. PEGylated-FeXCT nanoparticles (PFeXCT) were collected via centrifugation at 10,000 × $g$ for 10 min and washed with ultra-pure water to remove unbound OHC–PEG–CHO.

The conjugation of SLIGKV-NH$_2$ to PFeXCT nanoparticles was achieved via condensation between amines and aldehydes. In brief, PFeXCT nanoparticles were pipetted into 5 mL of SLIGKV-NH$_2$ (0.2 mg mL$^{-1}$, in ultra-pure water) and stirred for 2 h at room temperature. SLIGKV-NH$_2$-functionalized PFeXCT nanoparticles (SK@PFeXCT) were collected via centrifugation at 10,000 × $g$ for 10 min and washed with water thrice to remove unbound peptides. To synthesize SR@PFeXCT nanoparticles, SLIGRL-NH$_2$ was used to modify the PFeXCT nanoparticles using the same methods.

## Characterization of SK@PFeXCT nanoparticles
The size distribution and zeta potential of nanoparticles were measured thrice using a Zetasizer Nano ZS90 (Malvern, Worcestershire, UK) at 25 °C. Nanoparticle morphologies were observed via TEM (JEM-F200, JEOL, Japan) and SEM (Sigma 300, Zeiss, Germany). Magnetic properties were assessed using a vibrating sample magnetometer (VSM) 7404 System (Lake Shore, USA).

The relative content of the ligands on the nanoparticles was determined by fluorescence detection. In brief, different concentration of Rhodamine B-labeled SLIGKV-NH$_2$ peptides was used to modify the PEGylated PLGA core which was synthesized with Cy7-labeled BSA, and the relative quantification was conducted according to the ratio of fluorescence intensity (Rhodamine B / Cy7) with Spark® multimode microplate reader (Tecan, Switzerland).

The XCT790 encapsulation efficiency (EE) and loading content (LC) were quantified using HPLC (UltiMate 3000, ThermoFisher). Briefly, 5 mg of nanoparticles were dissolved in 0.5 mL of DCM and diluted with 4.5 mL of methanol. Separations were performed using a C18 3 μm 4.6 × 50 mm column at an isocratic flow rate of 1 mL min$^{-1}$ with the following mobile phase: 0.25% phosphate buffer (in HPLC water, pH = 3.0, adjusted with triethylamine) and 30% (v v$^{-1}$) acetonitrile. XCT790 concentrations were detected at a wavelength of 280 nm.

The release kinetics of XCT790 were studied via dialysis. Briefly, nanoparticles were dispersed in PBS supplemented with 0.01% tween-80 at a concentration of 1 mg mL$^{-1}$ (5 mL) and added to a dialysis bag (molecular weight cutoff: 3500 Da), which was placed in a beaker containing 45 mL of phosphate buffer and incubated at 37 °C with shaking at 100 rpm. One milliliter of the liquid was collected at intervals and quantified with HPLC; the lost liquid was replenished in the beaker.

## Cellular internalization of nanoparticles in vitro
Mouse vascular smooth muscle cell line MOVAS-1 (Sunncell, cat #SNL-521) and hVICs were cultured with SK@PFeCy5 (0.1 mg mL$^{-1}$, in complete medium) for 6 h, and PFeCy5 was used as a negative control. Nuclei were visualized via Hoechst staining (1:1,000 dilution, cat #H3570, Thermo Fisher). The uptake of nanoparticles by cells was observed using an inverted fluorescence microscope (DMI8, Leica, Germany). For quantitative analysis, after culturing with nanoparticles, cells were digested with trypsin and washed with HBSS; single-cell fluorescence intensity was measured via flow cytometry (LSRFortessa, BD, USA). Mean fluorescence intensities for a total of 10,000 viable cells were analyzed.

## Animal studies
Wild-type C57BL/6J mice were purchased from Vital River Laboratories (Beijing, China), and low-density lipoprotein receptor (Ldlr)$^{-/-}$ mice in the C57BL/6J background were purchased from GemPharmatech

(Nanjing, China). Mice were housed in a specific pathogen-free environment under standard 12-h light/12-h dark conditions with free access to water and food. The room temperature was maintained at 20 - 26 °C, and relative humidity was around 40% ~ 70%.

## Mouse model of aortic valve calcification and stenosis
Two mouse models were established: (1) aortic valve calcification induced by hyperlipidemia; and (2) aortic valve stenosis induced by wire injury.

For the hyperlipidemia model, 8-week-old male Ldlr$^{-/-}$ mice in the C57BL/6J background were fed an HFD (cat #TD 88137, SYSE Bio-tech, Changzhou, China) for 24 weeks, while littermate Ldlr$^{-/-}$ mice were fed an ND (control group). A total of one hundred and sixty-nine Ldlr$^{-/-}$ mice were used in three experiments.

The wire injury model was created as previously described[49]. In brief, 8-week-old male C57BL/6J mice were anesthetized with pentobarbital. The right common carotid artery was exposed via blunt separation, and the blood flow was blocked with an arterial clip. Subsequently, a guidewire with a 15° angled tip (ASAHI INTECC MIRACLEbros 6) was sent through a small sheared incision in the artery across the aortic valve into the left ventricle under ultrasound guidance. The aortic valve injury was achieved by pushing and pulling the guidewire 20 times and rotating the guidewire 200 times, following which the guidewire was slowly removed, and the carotid artery was tightly ligated to prevent postoperative hemorrhage. Sham mice were modeled similarly, except that the guidewire was inserted only into the common carotid artery and did not cross the aortic valve into the left ventricle. A total of twenty-four C57BL/6J mice were used in the wire injury experiment.

## Echocardiography
Cardiac function and aortic valve function were measured via echocardiography using a Vevo® 2100 ultrasound system (VisualSonic, FUJIFILM, Canada). Mice were anesthetized with 1.5–2% isoflurane in 95% O$_2$. Serial B-mode and M-mode images were obtained on the left parasternal long-axis and parasternal short-axis view to assess the cardiac function, pulsed-wave doppler was performed along the parasternal long-axis to measure the peak aortic jet velocity, and transvalvular gradient pressure.

## Histological analysis
After euthanasia via an overdose phenobarbital injection (100 mg kg$^{-1}$), the mouse heart was dissected, embedded in OCT, and cryosectioned at a 7-μm thickness. H&E staining was performed, and the leaflet thickness was quantified as previously described[64]. Briefly, five sections from each mouse were used to calculate the mean leaflet thickness. For each slice, the average width at the thickest point of the three leaflets was defined as the leaflet thickness.

For Picro-Sirius Red staining, cryosections were fixed with 4% paraformaldehyde, stained with Picro-Sirius Red solution (cat #BP-DL030, Senbeijia, Nanjing, China) for 1 h, rinsed with acetate solution, and dehydrated in 100% ethanol. The staining results were visualized under a polarized light microscope, and the collagen content was quantified with Image Pro Plus software.

Aortic valve leaflet calcification was assessed with von Kossa staining. In brief, cryosections were fixed with 4% paraformaldehyde, incubated in silver nitrate solution (5%), and exposed to ultraviolet light for 1 h, followed by 5% sodium thiosulfate incubation for 2 min, and counterstained with a nuclear fast red solution. The mineral extracellular deposition was observed at 5× and 20× magnification with a Leica DM 3000 microscope and quantified with Image Pro Plus software. To exclude the interference of brown melanin deposits in the aortic leaflets, H&E staining at the same section level was performed. Areas that appeared black on H&E staining were not counted as calcium deposits.

### Aortic valve targeting analysis in the mouse model

To assess the aortic valve targeting ability of SR@PFe, fifty-six HFD-fed *Ldlr*$^{-/-}$ mice were intravenously injected with 200 μL of IR780-labeled nanoparticles (PIR, PFeIR or SR@PFeIR, 2 mg mL$^{-1}$, in PBS). Subsequently, a magnet was placed adjacent to the base of the heart for 15 min. Three hours after administration, major organs were collected for ex vivo imaging with an in vivo imaging system (IVIS, PerkinElmer, Waltham, MA).

To further explore the distribution of SR@PFe in the aortic valve leaflets, sixty-three aortic valve stenosis mice were injected with the same dose of Cy5-labeled nanoparticles; the hearts were collected at 3 h post-injection. After embedding in OCT and cutting into 7-μm thick sections, the Cy5 fluorescence at the aortic valve was observed via fluorescence microscopy.

### Administration of SR@PFeXCT to CAVD mouse model

To evaluate the therapeutic effect of nanoparticle administration, fifty 8-week-old male *Ldlr*$^{-/-}$ mice were randomly allocated to five groups: (1) fed with ND; (2) fed with HFD; (3) fed with HFD and intravenously injected weekly with SR@PFe, 15 mg kg$^{-1}$; (4) fed with HFD and intraperitoneally injected weekly with XCT790, 5 mg kg$^{-1}$, twice a week; (5) fed with HFD and injected intravenously weekly with SR@PFeXCT 15 mg kg$^{-1}$. HFD feed was continued for 24 weeks, and all treatments were initiated from HFD feed to week 16 and continued for 8 weeks until echocardiography and euthanasia for tissue harvest.

### Cellular toxicity

The cytotoxicity of nanoparticles was quantified using the cell counting kit-8 (CCK-8, cat #K1018, ApexBio) assay and Calcein-AM/PI staining (cat #K2081, ApexBio). Briefly, hVICs were seeded in 96-well plates and cultured with different concentrations of nanoparticles for 6 h. The medium was replaced after three washes with PBS, and 20 μL of CCK-8 reagent was pipetted into each well. Absorbance at 450 nm was determined after 4 h incubation. The absorbance of wells containing fresh medium served as background, and the absorbance of wells containing hVICs without nanoparticles treatment served as the negative control. Cell viability was calculated as follows: Cell Viability = (OD450 sample − OD450 background)/(OD450 negative − OD450 background) × 100%.

For Calcein-AM/PI staining, hVICs seeded in 96-well plates were incubated with various concentrations of nanoparticles for different time. After three washes with PBS, the medium was replaced by HBSS containing 2 μM of Calcein-AM and 4.5 μM of PI, followed by observation using an inverted fluorescence microscope.

### Reactive oxygen species measurement

Cells were digested with trypsin and washed with HBSS, followed by incubation with 10 μM DCFH-DA (cat #S0033S, Beyotime) at 37 °C for 20 min. Single-cell fluorescence intensity was measured via flow cytometry (LSRFortessa, BD, USA). Mean fluorescence intensities for a total of 10,000 viable cells were analyzed.

### Hemolysis assay

A 2-% erythrocyte suspension was prepared from fresh mouse blood and mixed with nanoparticles for 1 h at 37 °C. Unbroken erythrocytes were removed via centrifugation at 100 × $g$ for 5 min, followed by centrifugation at 10,000 × $g$ for 10 min to remove nanoparticles. The supernatants were collected and hemoglobin concentration was determined based on absorbance at 570 nm. Ultra-pure water was used as a positive control, and PBS as a negative control. The hemolysis rate was calculated as follows: Hemolysis Rate = (OD570 sample − OD570 negative)/(OD570 positive − OD570 negative) × 100%.

### In vivo toxicity

Nine 8-week-olf C57BL/6 J mice were injected with various nanoparticles (200 μL, 2 mg mL$^{-1}$ in PBS) via the tail vein. Whole blood was collected 3- and 15-days post-injection to test liver and kidney function indicators. Major organs were dissected and embedded in paraffin 15 days post-injection, and morphologies were observed via H&E staining.

### Cellular glucose metabolism testing

The glucose uptake of hVICs was assessed with the fluorescent glucose analog 2-NBDG. Briefly, hVICs were cultured in an osteogenic medium for 4 days supplemented with SK@PFeXCT NPs or not, and seeded into 96-well plates (5,000 cells/well). After overnight incubation, hVICs were incubated with glucose-free DMEM for 1 h, followed by 20 min incubation in (2-(N-(7-Nitrobenz-2-oxa-1,3-diazol-4-yl)Amino)-2-Deoxyglucose) (2-NBDG, 100 μM, cat #GC10289, GlpBio) at 37 °C. After three washes with PBS, fluorescence was quantified at 485/535 nm excitation/emission on a Spark® multimode microplate reader (Tecan).

The intracellular pyruvate level was measured with a pyruvate assay kit (cat #A081-1-1, Jiancheng Bioengineering Institute, China) according to the manufacturer's instructions. In brief, hVICs were harvested and lysed in RIPA buffer. After centrifugation at 10,000 × $g$ for 15 min, the supernatants were collected and incubated with the chromogenic substrate at 37 °C for 10 min. Stop solution was added and incubated for 5 min at room temperature. The absorbance value was quantified at a wavelength of 505 nm on a Spark® multimode microplate reader (Tecan). Pyruvate concentrations were corrected for total protein concentration using a BCA assay.

Lactic acid production was quantified by measuring the intracellular lactate level and the amount of lactate released into the medium with a lactic acid assay kit (cat #A019-2-1, Jiancheng Bioengineering Institute, China) according to the manufacturer's instructions. In brief, calcified hVICs were cultured in fresh medium for 24 h, the medium was collected, and cells were lysed with RIPA buffer. Cell lysates or medium was incubated with enzyme working solution and chromogenic substrate at 37 °C for 10 min. A stop solution was added and incubated for 5 min at room temperature. The absorbance value was quantified at a wavelength of 530 nm using a Spark® multimode microplate reader. Ultra-pure water was used as a negative control. Lactate concentrations were corrected for total protein concentration using a BCA assay.

ATP production was measured with a ATP Content Assay Kit (cat #BC0300, Solarbio) according to the manufacturer's instructions. Briefly, cells were collected in extract buffer and lysed via ultrasonic disruption. After centrifugation at 10,000 × $g$ for 10 min, the supernatants were collected and mixed with chloroform followed by centrifugation at 10,000 × $g$ for 3 min. The supernatants were mixed with a working solution in a UV-permeable plate, and the absorbance value A1 at a wavelength of 340 nm was immediately quantified. The mixture was incubated at 37 °C for 3 min, and the absorbance value A2 at a wavelength of 340 nm was quantified. Meanwhile, a standard curve was generated, and the ATP concentrations were determined.

The extracellular acidification rate was measured to estimate the glycolytic capacity using the Seahorse XF96 Extracellular Flux Analyzer (Agilent Technologies, USA) with a glycolysis stress test kit (cat #103020-100, Agilent). In brief, hVICs were cultured in an osteogenic medium for 4 days, either supplemented with SK@PFeXCT NPs or not, and seeded into an XF96 plate (5,000 cells/well). Cells were incubated with XF media for 1 h in a CO$_2$-free incubator at 37 °C before loading into the XF96 apparatus. After the baseline measurements were performed, cells were treated sequentially with glucose (10 mM), oligomycin (1 mM), and 2-deoxy glucose (2-DG, 50 mM). Acidification of the medium was recorded throughout the assay.

## Statistical analysis

Data were presented as mean ± standard deviation (SD) and all experiments were performed with at least three independent replicates. The normal distribution of variables was assessed using the Shapiro–Wilk normality test. The two-tailed, unpaired Student's $t$-test was applied for comparisons involving two groups, and one-way ANOVA or two-way ANOVA followed by Tukey's test was used for comparisons involving ≥3 groups. For skewed variables, the Kruskal–Wallis test was applied followed by Dunn's multiple comparison test. Statistical analyses and graphing were performed using GraphPad Prism v8.0.1. A $p$-value ≤ 0.05 was considered statistically significant.

## Reporting summary

Further information on research design is available in the Nature Portfolio Reporting Summary linked to this article.

## Data availability

The data of bulk RNA-seq reported in this paper have been deposited in the GEO under the accession number "GSE252268". The publicly available scRNA-seq data used in this study are available in the Bio-Project database under accession code "PRJNA562645". Any additional requests for information can be directed to, and will be fulfilled by, the corresponding authors. Source data are provided with this paper.

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

## Acknowledgements

This study was supported by the National Natural Science Foundation of China (82271606 for X.Liu, 32000971 for T.R., 82200407 for S.C., U22A20267 and 82030014 for J.W.), the National Key R&D Program of China (2019YFA0110400 for J.W.), and the Key Program of Major Science and Technology Projects in Zhejiang Province (2021C03097 for J.W. and 2022C03063 for X.Liu).

## Author contributions

Conceptualization: T.R., J.C., and X.Liu; Investigation: J.C., T.R., L.X., H.H., and X. Li; Methodology: T.R., J.C., M.M., X.Z., W.H., D.X., Y.Q., S.C., and K.Y.; Writing & Visualization: J.C.; Writing—Review and editing: T.R. and X. Liu; Funding acquisition: J.W., X. Liu, T.R., and S.C.

## Competing interests

The authors declare no competing interests.
