## [Peer review file · Nature Communications]

REVIEWER COMMENTS

Reviewer #1 (Remarks to the Author):

Here, Chen and Ren et al. found protease-activated-receptor-2 (PAR2) expression to be upregulated in human calcific aortic valve disease (CAVD) tissues and valvular interstitial cells (VICs) isolated from either human CAVD or aortic valve (AV) tissues and exposed to pro-osteogenic/pro-fibrotic (TGF- β stimulation) conditions, with similar results obtained in two independent mouse models of CAVD. By harnessing previously generated single-cell RNA-seq data (Nat Commun 13, 5461 (2022)), the authors perform a descriptive analysis showing that F2RL1 (ie, PAR2-encoding gene)-positive cells may represent a group of cells contributing to CAVD pathogenesis. Next, a magnetic nanocarrier was developed, with intravenous injection of the latter showing high valvular enrichment if PAR2-targeted peptides were used. Then, XCT90, a potentially anti-calcific drug (Science. 2021 Feb 12; 371(6530): eabd0724), was loaded onto these nanocarrier leading to reduced AV calcification/stenosis in vivo, collectively implying that the herein presented data supports the efficacy of a novel target drug delivery system for the prevention and potentially also treatment of CAVD. This is a very innovative and interesting study that certainly warrants publication. However, this reviewer identified some major and minor issues that need to be convincingly addressed by the authors to make the submitted article a candidate for publication in Nature Communications.

MAJOR:

1. It remains elusive why the authors solely focus on PAR2. Indeed, a variety of transcriptional changes occur when calcification sets in (Circulation. 2018 Jul 24;138(4):377-393). Further experimental evidence coupled with convincing data on the molecular/biochemical properties of PAR2 should be provided.
2. The avascular (!) AV is composed of three layers, with the endothelium lining the AV being mainly exposed to intravenously injected drugs. While the authors provide some evidence that F2RL1 expression is mainly confined to VICs rather than valvular endothelial cells (VECs), data on the main mechanism of action of the herein presented drug delivery system (does it migrate across the transendothelial barrier?) should be provided.
3. Along a similar line of thinking, previous work has shown that gene/protein expression is completely distinct depending on the AV layer studied. CAVD occurs preferentially on the aortic side, but the herein presented evidence does not differentiate between the different AV layers. Hence, the authors should experimentally address this issues, with a particular focus on layer-specific PAR2 expression and how drug uptake differs depending on the layer studied.
4. While each human control group has its individual pitfalls (eg, different patient baseline characteristics depending on the pathology studied, tissue preservation techniques may differ), this reviewer would strongly suggest to (i) show the baseline characteristics of the patients the AV tissues were retrieved from, (ii) apply tactics to mitigate any selection bias and/or residual confounding (eg, propensity-score matching), and (iii) provide more details on the tissue preservation protocol.
5. Given that CAVD evolves over years/decades with TAVI procedures now representing the therapeutic cornerstone of end-stage disease, medical therapy may find its place to prevent and/or delay the disease. Thus, this reviewer would strongly encourage the authors to test the herein presented drug delivery system at different disease stages.
6. The fibrotic burden is much higher in female CAVD patients relative to their male counterparts. Hence, any evolving drug delivery system should be tested separately in women and men. This must find consideration in any revision.
7. In the discussion section, network-guided tactics to interpret CAVD-derived omics data should be more comprehensively discussed.
8. The graphical abstract should be redrawn to concisely convey the main message(s) of the submitted article.
9. What was the rationale of using two distinct strategies (Tukey vs Dunn's) to mitigate any inflate in type-I errors due to multiple testing?

MINOR:

1. A variety of minor issues in the abstract have been identified that should be revised:
 - please avoid general statements such as "serious threat to the health of older adults"
 - based on current evidence, we cannot say whether so far tested medical therapies (statins, RANKL inhibition, among others) were ineffective due to inappropriate drug delivery systems or the fact that these targets do not significantly contribute to CAVD pathogenesis in vivo. Thus, the statement 'A proper drug delivery system can be a potential solution to CAVD drug therapy' should be rewritten.
 - the statement 'osteogenically differentiated VICs ... after valve calcification' is a pleonasm. Please rephrase.
 - the specific mouse models used should be stated
 - the sentence 'in induced animal model' should be rewritten
2. It is Tukey and not Turkey's multiple comparison test. Please revise.
3. A flow-chart of the patients included in the present study should be provided (could be Supplement).
4. Is the expression of the reference protein (GAPDH) influenced by pro-calcific/pro-fibrotic conditions?
5. Individual data points should be shown throughout (eg, in Figure 2B they are missing)
6. In the wire injury mouse model, was CAVD induced without the presence of hyperlipidaemia?
7. The advantages/disadvantages of the two mouse models used should be more comprehensively discussed.

Reviewer #2 (Remarks to the Author):

This is an interesting study on a drug delivery system for calcific aortic valve disease. There are a number of issues to be addressed:

The choice of delivery system should be better rationalized, since the platform itself (i.e. drug loaded PLGA NP) has been well known for many years.

Explain the regulatory status of XCT790.

Need to demonstrate avoidance of the XCT790 side effects in a model where the authors show that they do occur with the free drug.

The ratio of ligand to nanoparticle should be determined experimentally.

Controls of nanoparticle with mismatch peptide are missing. The differences in results could simply be that the 'targeted' NP is different. This should be added throughout.

Control of drug delivery without the magnetic field should also be added.

What is the nature of the IONPs used? Are they oleic acid coated? This detail is missing.

The results in Fig 5B/C are slightly curious since there is higher signal everywhere in the targeted NP image. The quantification analysis should be redone normalizing to the signal in the heart or other bystander organ.

Reviewer #3 (Remarks to the Author):

In this study, the authors utilized magnetic nanoparticles loaded with XCT790, designed to target

PAR2 receptor for the treatment of CAVD. The notable results of the manuscript are:

- PAR2 expression is elevated in calcific in both calcified aortic valves and osteogenic differentiated VICs
- the construction of magnetic nanoparticles functionalized with PAR2 ligands and carrying XCT790 that accumulate within calcified aortic valves and osteodifferentiated VICs
- magnetic nanoparticles
- this internalization leads to the inhibition of metabolic reprogramming in VICs by regulating the expression of PDK4.

This research stands out as one of the few studies employing magnetic nanoparticles for the purpose of targeting particular molecules involved in CAVD, with the majority of existing literature primarily concentrating on targeting cancer-related aspects. Lately, magnetic nanoparticles gathered attention due to their potential for targeting specific tissues thorough magnetically guided delivery. However, this technique requires further refinement to achieve optimal precision in delivering to specific tissues. The authors used suitable and vast methodologies, presented in detail. The statistical analysis was clearly explained and the quality of the data is adequate. The specific results are clearly stated and some implications in the field are explained. However, some improvements should be made. My comments to the authors are:

1. The Introduction section should explain in an additional paragraph how SPIO NPs were used previously in targeting other diseases.
2. Diabetic patients (1 in control and 2 in CAVD) should be excluded, as diabetes is known to accelerate CAVD and some results from these patients may interfere with overall statistics.
3. How was the aortic valve area (in cm², presented in Table S1) calculated? Is it the aortic valve opening? Why was this preferred instead of calcific area?
4. Throughout the article, it must made clear what results are from human patients, isolated VICs or mouse model. This is hard to follow sometimes. When appropriate, the title of the result and the title of the figure should also specify the experimental model.
5. In Section 2.4, the phrase stating that targeted NPs suspended in the cell culture media are not internalized into the cells because the distance is too high should be replaced. It is known from literature that based on ligand-receptor affinity, other NPs do not need magnetic field to bind to cellular membrane. This is also confirmed by the authors in 2.5, where they showed SK@PFeXCT binds to PAR2 receptor.
6. The results in Section 2.6 and Figure 5 were compared to NPs not under magnetic field control?
7. The data in Section 2.9 does not present cytotoxicity of nanoparticles in magnetic field.
8. Knowing that iron-based nanoparticles induce oxidative stress, do the authors have any data on this matter?
9. Taking into account that this is an exploratory study, as the authors claim themselves, the conclusion that the designed NPs could solve the clinical problem of CAVD is overestimated. This study and other using magnetic NPs need further validation in order to interpret the results in such a manner.
10. The English writing should be revised throughout the article, regarding grammar or phrase construction. For example, the authors should use "targeted nanoparticles" instead of "targeting NPs" or "protein expression" vs "abundance".
11. Some paragraphs in the Introduction Section are not clear. Some examples are:
 - "Magnetic navigation is a promising strategy to increase the drug accumulation. With the help of magnetic field, anti-CD63 antibodies-conjugated magnetic nanoparticles dragged captured circulating exosomes to accumulated around the infarcted myocardium."
 - "A disease specific cell membrane marker that exposure during proctological process can be a potential bollard for drug anchoring."

Response to reviewer #1

Reviewer #1 (Remarks to the Author):

Here, Chen and Ren et al. found protease-activated-receptor-2 (PAR2) expression to be upregulated in human calcific aortic valve disease (CAVD) tissues and valvular interstitial cells (VICs) isolated from either human CAVD or aortic valve (AV) tissues and exposed to pro-osteogenic/pro-fibrotic (TGF- β stimulation) conditions, with similar results obtained in two independent mouse models of CAVD. By harnessing previously generated single-cell RNA-seq data (Nat Commun 13, 5461 (2022)), the authors perform a descriptive analysis showing that F2RL1 (ie, PAR2-encoding gene)-positive cells may represent a group of cells contributing to CAVD pathogenesis. Next, a magnetic nanocarrier was developed, with intravenous injection of the latter showing high high valvular enrichment if PAR2-targeted peptides were used. Then, XCT90, a potentially anti-calcific drug (Science. 2021 Feb 12; 371(6530): eabd0724), was loaded onto these nanocarrier leading to reduced AV calcification/stenosis in vivo, collectively implying that the herein presented data supports the efficacy of a novel target drug delivery system for the prevention and potentially also treatment of CAVD. This is a very innovative and interesting study that certainly warrants publication. However, this reviewer identified some major and minor issues that need to be convincingly addressed by the authors to make the submitted article a candidate for publication in Nature Communications.

Dear reviewer:

Thank you very much for your time involved in reviewing the manuscript and your very encouraging comments on the merits. We appreciate your clear and detailed suggestions, which has significantly improved the presentation of our manuscript, and hope that the explanation has fully addressed all of your concerns. In the remainder of this letter, we discuss each of your comments individually along with our corresponding responses.

MAJOR:

1. It remains elusive why the authors solely focus on PAR2. Indeed, a variety of

transcriptional changes occur when calcification sets in (Circulation. 2018 Jul 24;138(4):377-393). Further experimental evidence coupled with convincing data on the molecular/biochemical properties of PAR2 should be provided.

Response:

Thank you for your great suggestion. We reanalyzed the data and conducted some experiments to address this issue.

First, the RNA-seq data of hVICs was analyzed, and more than 3,000 genes were found to be upregulated after osteogenic differentiation. Gene Ontology cellular component analysis was conducted and genes whose corresponding proteins are localized to the cell membrane were screened out. After excluding those cell membrane proteins without clear ligands, the 10 proteins with the most significantly increased expression were selected as candidate markers (**Supplementary Fig. 2**). Quantitative polymerase chain reaction verified that the expression of *F2RL1*, *F2RL2*, *NPR1*, *FPR3*, and *CSF2R* was prominently unregulated after osteogenic induction, while *NPR1*, *FPR3*, and *CSF2R* was expressed at very low levels in VICs (**Fig. 2a, b**). However, PAR3 (i.e., *F2RL2* encoding protein) does not respond to synthetic peptides that mimic the putative tethered ligand. (Peach, Edgington-Mitchell et al. Protease-activated receptors in health and disease. *Physiol Rev*, 2023, 103: 717-785.)

In addition, here, PAR2 is well suited for being selected as a target for drug delivery system targeting CAVD due to the following molecular properties. First, PAR2 is selectively and highly expressed in the plasma membrane of VICs located in the pericalcified area and fibrotic area of calcified leaflets (**Fig. 2g, h**). Secondly, PAR2 will not be shed outside of cells, like soluble CD30, (Nagata, Ise et al. Cell membrane-specific epitopes on CD30: Potentially superior targets for immunotherapy. *Proc Natl Acad Sci U S A*, 2005, 102: 7946-7951.) which may competitively inhibit the binding of targeting particles to diseased cells (**Supplementary Fig. 22**). Third, PAR2 will be internalized after binding its ligands and routed to lysosomes, which could promote the internalization and drug release of PAR2-based TDDS. (Jung, Jiang et al. beta-arrestin-dependent PI(4,5)P2 synthesis boosts GPCR endocytosis. *Proc Natl Acad Sci U S A*, 2021, 118.) Finally, although PAR2 is degraded in lysosomes after being endocytosed, PAR2 stored in the Golgi apparatus as well as de novo synthesized is rapidly mobilized to the plasma

membrane for repopulation. (Zhao, Pattison et al. Protein kinase D and Gbetagamma mediate sustained nociceptive signaling by biased agonists of protease-activated receptor-2. *J Biol Chem*, 2019, 294: 10649-10662.)

Following are our revisions to the manuscript:

Page 12 of Revised manuscript:

VICs membrane proteins up-regulated after osteogenic differentiation were screened as targets of the CAVD TDDS. To this end, we analyzed the gene expression profiles of quiescent hVICs and osteogenically differentiated hVICs (cellular characterization shown in **Supplementary Fig. 1**) and identified 3468 up-regulated genes (filtered by $|\text{fold change (FC)}| \geq 2$ and $p < 0.01$; **Supplementary Fig. 2a, b**). Gene Ontology cellular component analysis was conducted, and genes with corresponding proteins localized to the cell membrane were screened. After excluding those proteins without specific ligands, the top 20 differentially expressed genes were selected as candidate markers (**Fig. 2a**). Quantitative polymerase chain reaction (qPCR) verified that the expression of F2RL1, F2RL2, NPR1, FPR3, and CSF2R were prominently up-regulated after osteogenic induction. In contrast, NPR1, FPR3, and CSF2R were expressed at very low levels in hVICs (**Fig. 2b, Supplementary Fig. 2c**). Notably, PAR3 (i.e., F2RL2-encoded protein) does not respond to synthetic peptides that mimic the putative tethered ligand.³¹ Western blot further confirmed that with prolonged osteogenic induction time, the protein expression of PAR2 (i.e., F2RL1-encoded protein) increased in a time-dependent manner and was positively correlated with osteogenic differentiation markers (**Fig. 2c**).

Page 14 of Revised manuscript:

Immunohistochemical staining was performed to assess the distribution of PAR2 in the tri-layer structure of leaflets, including the collagen-rich fibrosa on the aortic side, the proteoglycan-rich spongiosa, and the elastin-rich ventricularis. Compared with those in non-calcified leaflets, PAR2-positive cells were more abundant in all three layers of calcified leaflets, particularly in the peri-calcified and fibrotic areas of the fibrosa layer, as evidenced by Alizarin Red staining and Masson's trichrome staining in serial slices (**Fig. 2g, h**). Immunofluorescence further revealed that PAR2 was not colocalized with CD31—an endothelial cell marker—but colocalized with vimentin (i.e., a marker of VICs).³⁴ The number of PAR2-positive cells in the calcified aortic valve markedly increased compared with normal leaflets (**Fig. 2i, j**).

Page 43 of Revised Manuscript (Discussion section):

To design a suitable TDDS for the treatment of CAVD, we determined that PAR2 is highly expressed in osteogenically differentiated VICs, and synthesized a PAR2-ligand functionalized magnetic response nanocarrier to deliver XCT790 to the calcified aortic valves. Targets for TDDS must be selectively and highly expressed at intercellular and subcellular levels.⁵ PAR2 is highly expressed in hVICs within the fibrosa layer of calcified leaflets, and is expressed at low levels by VECs. As a GPCR, PAR2 is primarily located within the plasma membrane, with a small amount distributed in the Golgi apparatus during the synthesis and sorting process and endosomes during endocytosis, showing extremely high subcellular localization specificity.³¹ Moreover, targets shed from the cell membrane may competitively inhibit the binding of TDDS

to diseased target cells (e.g., soluble CD30).⁴⁷ However, PAR2 is not shed from cells and, thus, will not impact the targeting ability of the TDDS (**Supplementary Fig. 22**). Furthermore, PAR2 is internalized after binding its ligands and routed to lysosomes, which may promote the internalization and drug release of PAR2-based TDDS.²⁰ Finally, once PAR2 is degraded in lysosomes after being endocytosed, it is stored in the Golgi apparatus; following de novo synthesis, it is rapidly mobilized to the plasma membrane for repopulation.⁴⁸

Page 2 of Revised Supplementary Information:

Supplementary Fig. 2. RNA-seq results of quiescent hVICs and osteogenically differentiated hVICs and validation by quantitative polymerase chain reaction (qPCR).

(a) Clustered heatmap shows differential gene expression between quiescent hVICs and osteogenically differentiated hVICs. (b) Volcano plot of differential gene expression. Red: up-regulated genes, blue: down-regulated genes, gray: not significant. Adjusted p value < 0.01, log₂ Fold Change < -1 or > 1, n = 3/each group. (c) QPCR validation of top 20 up-regulated plasma membrane proteins in osteogenically differentiated hVICs compared with quiescent hVICs. Values are presented as mean ± SD.

Page 16-17 of Revised Manuscript:

Fig. 2. PAR2 expression is increased in osteogenically differentiated hVICs and calcified aortic valves. (a) Heatmap of the top 20 differentially expressed genes with corresponding proteins that bind to specific ligands and are localized to the cell membrane. (b) Quantitative analysis of F2RL1, F2RL2, NPR1, FPR3, and CSF2R mRNA levels in hVICs after osteogenic induction; $n = 3/\text{group}$. (c) Western blot and quantification of PAR2, RUNX2, and ALP in hVICs cultured with osteogenic induction medium for different times (0, 3, 7, 14 days); $n = 3/\text{group}$. (d) Protein expression of PAR2 in hVICs undergoing myofibroblastic induction for 48 h; $n = 6/\text{group}$. (e) Non-calcified and calcified human aortic valve leaflet specimens collected from surgery. (f) Protein expression of PAR2 in human aortic valves. Non-calcified group: $n = 12$; calcified group: $n = 15$. (g) Representative images of Alizarin Red staining, Masson's Trichrome staining and immunohistochemical staining of PAR2 in normal and calcified aortic valve tissue in full size (left; scale bar = 500 μm) and in enlarged images (right; scale bar = 50 μm). (h) Quantitative analysis of immunohistochemical staining of

PAR2 in the ventricularis, spongiosa, and fibrosa layers of calcified leaflets; n = 8/group. (i) Immunofluorescence staining of CD31 (green), PAR2 (red) and DAPI (blue) in normal and calcified aortic valve tissue. Scale bar = 100 μ m. (j) Immunofluorescence photographs and quantitative analysis of vimentin (green), PAR2 (red), and DAPI (blue) in non-calcified and calcified aortic valve tissue (left; scale bar = 50 μ m) and in enlarged images (right; scale bar = 10 μ m). Yellow indicates colocalization of PAR2 in hVICs (merge; white arrow); n = 10/group. (k) PAR2 protein levels in hVICs isolated from non-calcified and calcified aortic valves; n = 9/group. (l) Immunofluorescence staining and quantification of PAR2 in aortic valves of *Ldlr*^{-/-} mice fed a normal or high-fat diet. Scale bar = 200 μ m, n = 8/group. Values are presented as mean \pm SD.

Page 16 of Revised Supplementary Information:

Supplementary Fig. 22

Immunoblot of PAR2 in whole cell lysate and supernatant of cultured hVICs.

[Ref]

- [5] Mitchell, M. J. *et al.* Engineering precision nanoparticles for drug delivery. *Nat Rev Drug Discov* **20**, 101-124, doi:10.1038/s41573-020-0090-8 (2021).
- [20] Jung, S. R. *et al.* beta-arrestin-dependent PI(4,5)P2 synthesis boosts GPCR endocytosis. *Proceedings of the National Academy of Sciences of the United States of America* **118**, doi:10.1073/pnas.2011023118 (2021).
- [31] Peach, C. J., Edgington-Mitchell, L. E., Bunnett, N. W. & Schmidt, B. L. Protease-activated receptors in health and disease. *Physiol Rev* **103**, 717-785, doi:10.1152/physrev.00044.2021 (2023).
- [34] Matilla, L. *et al.* A Role for MMP-10 (Matrix Metalloproteinase-10) in Calcific Aortic Valve Stenosis. *Arteriosclerosis, thrombosis, and vascular biology* **40**, 1370-1382, doi:10.1161/ATVBAHA.120.314143 (2020).
- [47] Nagata, S. *et al.* Cell membrane-specific epitopes on CD30: Potentially superior targets for immunotherapy. *Proceedings of the National Academy of Sciences of the United States of America* **102**, 7946-7951, doi:10.1073/pnas.0502975102 (2005).
- [48] Zhao, P. *et al.* Protein kinase D and Gbetagamma mediate sustained nociceptive signaling by biased agonists of protease-activated receptor-2. *J Biol Chem* **294**, 10649-10662, doi:10.1074/jbc.RA118.006935 (2019).

2. The avascular (!) AV is composed of three layers, with the endothelium lining the AV being mainly exposed to intravenously injected drugs. While the authors provide some evidence that F2RL1 expression is mainly confined to VICs rather than valvular

endothelial cells (VECs), data on the main mechanism of action of the herein presented drug delivery system (does it migrate across the transendothelial barrier?) should be provided.

Response:

Thank you for the suggestion. Indeed, both the ventricular side and aortic sides of the aortic valve leaflet are covered with an endothelial layer, which is composed of VECs. However, it has been confirmed in microfluidic devices and animal experiments that, due to the different hemodynamics on both sides (oscillatory blood flow on the aortic side and laminar blood flow patterns on the ventricular sides), VECs on the aortic side are more prone to apoptosis and lead to endothelial denudation. (Tanaka, Sata et al. Age-associated aortic stenosis in apolipoprotein E-deficient mice. *J Am Coll Cardiol*, 2005, 46: 134-141; Lee, Estlack et al. A microfluidic cardiac flow profile generator for studying the effect of shear stress on valvular endothelial cells. *Lab Chip*, 2018, 18: 2946-2954.) In addition, a human-based study has revealed that the progress of CAVD is accompanied by a decrease in the proliferation capacity of VECs and a reduction in the number and function of endothelial progenitor cells (EPCs). (Gossl, Khosla et al. Role of circulating osteogenic progenitor cells in calcific aortic stenosis. *J Am Coll Cardiol*, 2012, 60: 1945-1953.)

Follow your suggestion, we provide more data for the inference that the drug delivery system passes through the endothelial layer. In the aortic valve leaflets of HFD-fed *Ldlr*^{-/-} mice injected with SR@PFeCy5, we found that isolectin B4 fluorescence were not continuous in the aggregation region of Cy5 fluorescence, which may indicate that nanoparticles can enter the subendothelial layer through gaps between endothelium. (**Fig. 6g**)

Following are our revisions to the manuscript:

Page 32 of Revised Manuscript:

Additionally, considering that endothelial denudation occurs with valve calcification, we speculate that SR@PFe may enter the subendothelial layer through gaps between the endothelium. Immunofluorescence confirmed that isolectin B4 was not continuous in the calcified leaflets, and SR@PFeCy5 aggregated in the areas of endothelial defects (**Fig. 6g**).

Page 33 of Revised Manuscript:

Fig. 6
(g) Co-labelled isolectin B4 and Cy5.

3. Along a similar line of thinking, previous work has shown that gene/protein expression is completely distinct depending on the AV layer studied. CAVD occurs preferentially on the aortic side, but the herein presented evidence does not differentiate between the different AV layers. Hence, the authors should experimentally address this issues, with a particular focus on layer-specific PAR2 expression and how drug uptake differs depending on the layer studied.

Response:

It is really a great suggestion you pointed out. The aortic valve is composed of multiple layers, including the covered endothelium, the collagen-rich fibrosa on the aortic side, the proteoglycans-rich spongiosa, and the elastin-rich ventricularis. (Aikawa and Libby. A Rock and a Hard Place: Chiseling Away at the Multiple Mechanisms of Aortic Stenosis. *Circulation*, 2017, 135: 1951-1955.) Firstly, we mapped PAR2 protein expression across the layer of human aortic valve leaflets by immunohistochemistry. As shown in revised **Fig. 2g, h**, the PAR2-positive cells were enriched in fibrosa layer. Immunofluorescence staining confirmed that, PAR2-positive cells were hardly detectable on the endothelium (**Fig. 2i**). Then, the PAR2 expression pattern was examined in aortic valve leaflets of mouse. Taking the midline of the leaflets as the boundary, the PAR2 fluorescence intensity on the aortic side was

significantly higher than that on the ventricular side (**supplementary Fig. 4**).

Following are our revisions to the manuscript:

Page 14 of Revised Manuscript:

Immunohistochemical staining was performed to assess the distribution of PAR2 in the tri-layer structure of leaflets, including the collagen-rich fibrosa on the aortic side, the proteoglycan-rich spongiosa, and the elastin-rich ventricularis. Compared with those in non-calcified leaflets, PAR2-positive cells were more abundant in all three layers of calcified leaflets, particularly in the pericardified and fibrotic areas of the fibrosa layer, as evidenced by Alizarin Red staining and Masson's trichrome staining in serial slices (**Fig. 2g, h**). Immunofluorescence further revealed that PAR2 was not colocalized with CD31—an endothelial cell marker—but colocalized with vimentin (i.e., a marker of VICs) (**Fig. 2i, j**). The number of PAR2-positive cells in the calcified aortic valve markedly increased compared with normal leaflets (**Fig. 2j**).

Page 14 of Revised Manuscript:

Immunofluorescence staining revealed that PAR2 expression was significantly elevated in the aortic valves of *Ldlr*^{-/-} mice fed an HFD compared with those fed a normal diet (ND; Fig. 2l). Moreover, PAR2 expression was higher on the aortic side of the leaflet than on the ventricular side (**Supplementary Fig. 4**).

Page 13-14 of Revised Manuscript:

Fig. 2

(g) Representative images of Alizarin Red staining, Masson's Trichrome staining and immunohistochemical staining of PAR2 in normal and calcified aortic valve tissue in full size (left; scale bar = 500 µm) and in enlarged images (right; scale bar = 50 µm). (h) Quantitative analysis of immunohistochemical staining of PAR2 in the ventricularis, spongiosa and fibrosa

layers of non-calcified and calcified leaflets, n =8/group. (i) Immunofluorescence staining of CD31 (green), PAR2 (red) and DAPI (blue) in normal and calcified aortic valve tissue. Scale bar = 100 μ m.

Page 4 of Revised Supplementary Information:

Supplementary Fig. 4. Expression distribution of PAR2 on the mouse aortic valves. PAR2 fluorescence intensity distribution on the aortic valves of ND-fed and HFD- *Ldlr*^{-/-} mice; n = 8/group. Values are presented as mean \pm SD.

4. While each human control group has its individual pitfalls (eg, different patient baseline characteristics depending on the pathology studied, tissue preservation techniques may differ), this reviewer would strongly suggest to (i) show the baseline characteristics of the patients the AV tissues were retrieved from, (ii) apply tactics to mitigate any selection bias and/or residual confounding (eg, propensity-score matching), and (iii) provide more details on the tissue preservation protocol.

Response:

Thank you for your suggestions.

- The clinical characteristics of the tissue donors has been reorganized and provided in the **Supplementary Table 1**.

- Indeed, patient's baseline characteristics affects the statistical results. As suggested, propensity-score matching (PSM) was conducted, and parameters except echocardiographic results (i.e., age, sex, BMI, smoking, hypertension, coronary heart disease, lipid profiles) were selected as confounding factors. Six pairs of specimens were matched with a matching tolerance of 0.03, and a two-tailed t-test was performed. There was no difference in the baseline parameters, while the expression of PAR2 showed a significant difference between the two groups (**Supplementary Table 2**). Therefore, it is reasonable to assume that PAR2 expression is elevated in calcified

aortic valve tissues specimens.

- Details of tissue specimen acquisition, preservation, and use are provided.

Following are our revisions to the manuscript:

Page 13-14 of Revised Manuscript:

To further confirm that PAR2 expression is upregulated in osteogenically differentiated VICs as well as in calcified valve tissue *in vivo*, the protein expression pattern of PAR2 was examined in non-calcified and calcified aortic valves (leaflet morphology shown in **Fig. 2e**, patient information shown in **Supplementary table 1**). Western blot analysis revealed that the protein expression of PAR2 was significantly higher in the calcified valve tissue than in the control valve tissue (**Fig. 2f**). When male and female valve specimens were analyzed separately, we found that increased PAR2 expression in calcified valves was not affected by sex differences (**Supplementary Fig. 3c**). To minimize bias due to the baseline characteristics of different patients, propensity-score matching (PSM) was conducted. After matching, no difference was observed in the baseline parameters, while the expression of PAR2 was significantly up-regulated in the CAVD group (**Supplementary Table 2**).

Page 48-51 of Revised Manuscript (Methods):

Human valve leaflets collection

All experiments involving humans were conducted in accordance with the Declaration of Helsinki and were approved by The Second Affiliated Hospital, Zhejiang University (No. IRB-2022-0085). Written informed consent was provided by each participant. Calcified aortic valve leaflets were obtained from severe aortic stenosis patients who underwent aortic valve replacement surgery, while non-calcified aortic valves were obtained from heart transplant recipients with Stanford type A acute aortic dissection or aortic valve regurgitation. Valves from patients with rheumatic disease, infective endocarditis, congenital valvular disease, or diabetes were excluded (**Supplementary Fig. 24**). Leaflets were divided into four pieces immediately after acquisition: (1) digested with collagenase type II for valvular interstitial cell isolation and culture. (2) homogenized in radioimmunoprecipitation assay (RIPA) buffer for protein extraction and immunoblot; (3) embedded in paraffin for immunohistochemistry staining, Masson staining, and Alizarin Red staining; (4) embedded in optimal cutting temperature compound (OCT, Sakura Finetek, Japan) for immunofluorescence staining, H&E staining, and Alizarin Red staining.

Isolation, culture, osteogenic differentiation of hVICs

Isolation and culture of hVICs were performed as previously described. Briefly, non-calcified aortic valve leaflets were washed with PBS twice, minced into 1 mm³ pieces, and digested with collagenase type II (2 mg mL⁻¹, in DMEM, Gibco) for 4 h at 37 °C. The tissue solution was further dissociated by pipetting, and precipitated by centrifugation at 200 ×g for 5 min. Isolated hVICs were resuspended and cultured in high-glucose DMEM supplemented with 10% FBS and 1% penicillin/streptomycin. For the isolation of hVICs derived from calcified leaflets, calcium nodules in the leaflets were removed with forceps, and valves were washed twice with penicillin/streptomycin-containing PBS, minced, digested, used for subsequent cultures. All

hVICs used in in vitro experiments were from passages 2–5.

Western blot analysis

Aortic valve leaflets were collected and stored at -80°C . For protein extraction, after removing large calcific nodules, leaflets were minced and homogenized in cold RIPA buffer containing PMSF with using a tissue grinder (Tissuelyser, Jingxin, Shanghai). Protein concentration was determined with BCA protein assay (Thermo Fisher). For western blotting, equal quantities of proteins were separated using 10% sodium dodecyl sulfate polyacrylamide gel electrophoresis (SDS-PAGE) and transferred to PVDF membranes. After blocking with skim milk, proteins were detected with specific primary antibodies followed by HRP-linked secondary antibodies. Finally, the blots were developed using the enhanced chemiluminescence reagent (Vazyme, Nanjing, China) with ImageQuant 800 (Amersham, GE healthcare), and the bands were quantified by ImageQuant TL (Version 8.2, Amersham).

Immunohistochemistry staining

Immunohistochemistry staining was applied to detect PAR2 expression in aortic valve leaflets. In brief, tissues were fixed in paraformaldehyde, dehydrated in graded ethanol, embedded in paraffin, and sectioned at a $3.5\ \mu\text{m}$ thickness. After dewaxing, rehydration, antigen retrieval, and blocking, sections were incubated with a primary antibody against PAR2 overnight at 4°C . They were then incubated with an HRP-conjugated secondary antibody and visualized with 3,3-diaminobenzidine (DAB, zsbio, Beijing, China). Images were captured by a DM3000 microscope (Leica, Germany) with LAS X software (Version 3.0, Leica),

Page 19 of Revised Supplementary Information:

Supplementary Table 2 PAR2 expression analysis before and after Propensity-score matching

Parameters	Unmatched			Matched		
	Control (n = 12)	CAVD (n = 15)	p Value	Control (n = 6)	CAVD (n = 6)	p Value
Age	59.5833±17.3648	61.8±7.966	0.663	59.5±10.5972	62.8333±7.4677	0.544
BMI	23.0675±3.903	22.3013±2.7452	0.572	21.655±3.4219	22.9983±3.2718	0.503
Sex, (Female, %)	4 (33.33)	7 (46.7)	0.502	0 (0)	3 (50)	0.076
Smoking, n (%)	5 (41.67)	3 (20)	0.236	1 (16.67)	3 (50)	0.262
BAV, n (%)	0	0	N/A	0	0	N/A
DM, n (%)	0	0	N/A	0	0	N/A
Hypertension, n (%)	6(0.5)	6(0.4)	0.621	3(0.5)	1(0.1667)	0.262
CHD	0.1667±0.3892	0.2±0.414	0.832	0.1667±0.4082	0.1667±0.4082	1.000
LDL	2.1708±0.609	2.1507±0.8288	0.943	2.2033±0.4859	2.2083±0.8343	0.990
HDL	1.2975±0.3377	1.2393±0.3426	0.663	1.4533±0.3154	1.35±0.2188	0.526
TG	1.0733±0.3246	2.072±2.4369	0.173	1.1217±0.3773	1.1767±0.3018	0.786
T-CHO	4.1358±0.8583	4.3233±0.9298	0.592	4.3883±0.8134	4.2817±0.9082	0.835
Statin, n (%)	1(8.33)	2(13.33)	0.695	1(16.67)	1(16.67)	1.000
ACEi/ARB, n (%)	2(16.67)	2(13.33)	0.817	1(16.67)	1(16.67)	1.000
PAR2 expression	0.9033±0.7833	5.178±3.2122	1.41e-04	1.2667±0.9082	5.8833±3.0995	0.006

5. Given that CAVD evolves over years/decades with TAVI procedures now representing the therapeutic cornerstone of end-stage disease, medical therapy may find its place to prevent and/or delay the disease. Thus, this reviewer would strongly encourage the authors to test the herein presented drug delivery system at different disease stages.

Response:

Thank you for the great comments. Aortic valve stenosis is a disease with a long course induced by multiple factors. Generally, the pathophysiology of aortic stenosis can be divided into sclerosis phase and stenosis phase, which are continuous. The main features of the sclerosis stage are endothelial damage, lipid infiltration and inflammation, and the main features of the stenosis stage are valve calcification and fibrosis. In fact, these two phases may be active to varying degrees at the same time in different parts of the leaflet and at different stages of the disease process. (Moncla, Briand et al. Calcific aortic valve disease: mechanisms, prevention and treatment. *Nat Rev Cardiol*, 2023, 20: 546-559.) It manifests as leaflet sclerosis and fibrosis in the early stage and forms calcification in the later stage, and the myofibroblastic differentiation and osteogenic differentiation of VICs are usually considered to be the key causes of valve dysfunction and stenosis. In this study, we verified the targeting ability of nanoparticles to myofibroblastically differentiated and osteogenically differentiated cells (**Fig. 2c-d, and Fig. 3e-h**).

In addition, follow your suggestion, we examined the targeting ability of SR@PFeCy5 in mice fed with HFD at different time points, and found that once initial atherosclerosis was formed after 3 months of HFD feeding, a small amount of SR@PFeCy5 were targeted to the atherosclerotic valves (**Fig 6d, e**). As the modeling time was extended to 5 months, the SR@PFeCy5 were more effectively enriched on the diseased leaflets (**Fig 6d, e**). Meanwhile, the enrichment of nanoparticles in the atherosclerotic aortic sinus at different stages were examined. After 12 weeks of HFD feeding, atherosclerosis plaques were formed, and the SR@PFeCy5 were enriched on plaques. More enrichment domains appeared at 5 months modeling (when plaques are thicker and more severe) (**supplementary Fig. 13**).

Following are our revisions to the manuscript:

Page 31-32 of Revised Manuscript:

To specifically examine the ability of the SR@PFe nanoplatform in targeting lesioned aortic valves at different disease stages, *Ldlr*^{-/-} mice fed an HFD for various time durations were administered with Cy5-labeled SR@PFe (SR@PFeCy5); hearts were subsequently collected and sectioned for analysis. In the aortic valve of ND-fed mice, regardless of the type of nanoparticles injected, relatively no fluorescence was observed, indicating that SR@PFe were not enriched in non-diseased aortic valves (Figure 6d, e). Under EMF navigation, few SR@PFe were enriched in the lesion leaflets of mice fed with HFD for 3 months, while the mice fed with HFD for 5 months had more nanoparticles on the lesion leaflets (Figure 6d, e). The fluorescence intensity of SR@PFe in the EMF group was higher than that of PAR2-targeting alone or EMF alone, indicating that the dual-targeting worked synergistically (Figure 6d, e). Notably, most of the SR@PFeCy5 was accumulated in the aortic side of the aortic valve leaflets, while negligible amount was observed on the ventricular side (Fig. 6f), possibly due to a vortex area with lower flow velocity on the aortic side and the hierarchical expression of PAR2. Additionally, considering that endothelial denudation occurs with valve calcification, we speculate that SR@PFe may enter the subendothelial layer through gaps between the endothelium. Immunofluorescence confirmed that isolectin B4 was not continuous in the calcified leaflets, and SR@PFeCy5 aggregated in the areas of endothelial defects (Fig. 6g). Moreover, fluorescence was detected in the atherosclerotic aortic sinus, which agreed with the higher PAR2 expression in this region. As expected, copious amounts of fluorescence were detected in the aortic sinus of the HFD-fed *Ldlr*^{-/-} mice, and more SR@PFeCy5 nanoparticles were detected in the aortic root under magnetic navigation when compared with the single-targeting control groups (Supplementary Fig. 13).

Page 33 of Revised Manuscript:

Fig. 6.

(d) Fluorescence images of aortic valves of HFD fed *Ldlr*^{-/-} mice 4 h after injected with nanoparticles. (e) Quantification of Cy5 mean fluorescence in the aortic valves, *n* = 7/group.

Supplementary Fig. 13. SR@PFeCy5 is enriched in atherosclerotic aortic sinus.

Fluorescence images of aortic sinus of HFD fed *Ldlr*^{-/-} mice 4 h post-nanoparticle injection; n = 7/group.

6. The fibrotic burden is much higher in female CAVD patients relative to their male counterparts. Hence, any evolving drug delivery system should be tested separately in women and men. This must find consideration in any revision.

Response:

Thank you for your valuable concern. We agree that sex difference could be an important factor affecting CAVD progression.

First, the immunoblot result of PAR2 expression in valve tissues donated by females and males were analyzed separately (patient information shown in **Supplementary table S1**). Compared with non-calcified leaflets, PAR2 protein expression was elevated in calcified leaflets, and not affected by sex differences (**Supplementary Fig. 3c**). Then, myofibrogenic induction and osteogenic induction were conducted on VICs derived from female patients, and elevated protein expression of PAR2 was observed, which was consistent with male patient derived hVICs (**Supplementary Fig. 3a, b**). Subsequently, further verification of targeting ability showed that, similar to male-derived VICs, female patient-derived VICs after lesion induction phagocytosed a large amount of SK@PFeCy5 compared with quiescent cells, indicating the herein

presented drug delivery system was also effective in female patients (**Supplementary Fig. 3d, e**). However, we were unable to verify the targeting ability in female mice because estrogen may affect the formation of aortic valve calcification and stenosis in mice, causing unpredictable bias. In addition, the drug delivery system has not been performed on humans due to ethical constraints.

Following are our revisions to the manuscript:

Page 13 of Revised manuscript:

Considering the differences in the pathological process of CAVD between female and male patients, including a higher fibrotic burden in female patients relative to their male counterparts, PAR2 protein expression were examined in female patient-derived hVICs.³³ Immunoblot confirmed the elevation of PAR2 expression in response to osteogenic differentiation and myofibroblastic differentiation in hVICs derived from female patients (**Supplementary Fig. 3a, b**).

Page 13 of Revised manuscript:

To further confirm that PAR2 expression is upregulated in osteogenically differentiated VICs as well as in calcified valve tissue in vivo, the protein expression pattern of PAR2 was examined in non-calcified and calcified aortic valves (leaflet morphology shown in **Fig. 2e**, patient information shown in **Supplementary table 1**). Western blot analysis revealed that the protein expression of PAR2 was significantly higher in the calcified valve tissue than in the control valve tissue (**Fig. 2f**). When male and female valve specimens were analyzed separately, we found that increased PAR2 expression in calcified valves was not affected by sex differences (**Supplementary Fig. 3c**).

Page 22 of Revised manuscript:

Additionally, the targeting ability of SK@PFeCy5 was further confirmed in female-derived osteogenically differentiated and myofibroblastically differentiated hVICs (**Supplementary Fig. 3d, e**).

Page 3 of Revised Supplementary Information

Supplementary Fig. 3.

(a) Immunoblot and quantification of PAR2, RUNX2, and ALP in hVICs derived from female patients cultured with osteogenic induction medium for 4 days; $n = 6/\text{group}$. (b) Protein expression of PAR2 in female patient-derived hVICs undergoing myofibrogenic induction for 48 h; $n = 6/\text{group}$. (c) Immunoblot quantification of PAR2 in human aortic valves from female and male patients; female: $n = 4$ for NC, $n = 7$ for CAVD; male: $n = 8$ for NC, $n = 8$ for CAVD.

Supplementary Fig. 3.

(d) Flow cytometry histogram and quantification of the PFeCy5 or SK@PFeCy5 uptake by osteogenically differentiated hVICs; $n = 3/\text{group}$. (e) Flow cytometry histogram and quantification of SK@PFeCy5 uptake by myofibrogenically differentiated hVICs; $n = 3/\text{group}$. Values are presented as mean \pm SD.

[Ref]

[33] Aguado, B. A. et al. Genes That Escape X Chromosome Inactivation Modulate Sex Differences in Valve Myofibroblasts. *Circulation* 145, 513-530, doi:10.1161/CIRCULATIONAHA.121.054108 (2022).

7. In the discussion section, network-guided tactics to interpret CAVD-derived omics

data should be more comprehensively discussed.

Response:

Thank you for your valuable suggestion. The application of a network-based integrated approach in the analysis of CAVD-derived multi-omics has become an important tool to better understand the molecular regulatory network of CAVD initiation, pathogenesis, and treatment, so we discuss the application of this strategy in CAVD research in recent years.

Following are our revisions to the manuscript:

Page 45-46 of Revised Manuscript:

Network-based integration methods are applied in the analysis of CAVD-derived multi-omics and have become an important tool for better understanding the molecular regulatory networks of CAVD initiation, pathogenesis, and treatment.^{53,54} By analyzing the gene regulatory networks via RNA-seq, NOTCH1 haploinsufficiency was identified as a potential risk factor for BAV and CAVD.⁵⁵ Moreover, network analysis of the sexually dimorphic transcriptome of porcine VICs identified two X-chromosome inactivation escape genes (BMX nonreceptor tyrosine kinase (BMX) and steroid sulfatase (STS)), which may explain the effect of sex on myofibroblast activation in VICs.³³ Furthermore, combining spatiotemporal transcriptomics with protein-protein interaction networks revealed that fibronectin-1 and protease inhibitor alpha-2-macroglobulin are associated with CAVD progression.⁵⁶ Meanwhile, the development of single-cell RNA sequencing (scRNA-seq) has enabled the study of VICs heterogeneity and the identification of three VICs subtypes.³⁵ Combining scRNA-seq and network analysis led to the isolation of a disease-driver VICs population with procalcific potential from human CAVD tissue, while temporal proteomic profiling identified monoamine oxidase-A (MAOA) and collagen triple helix repeat containing-1 (CTHRC1) as potential therapeutic targets.⁵⁷ Extracellular vesicles serve as major messengers in intercellular communication, greatly influencing disease progression.⁵⁸ The miRNA-mRNA target network facilitates the delivery of miRNAs by extracellular vesicles in valve tissue to targets in recipient cells. Integrated analysis has been employed to predict the cumulative effects of extracellular vesicle cargo. Ultimately, 62 altered miRNAs were screened and 1813 target genes were predicted. KEGG pathway enrichment and siRNA-mediated knockdown experiments verified that WNT5A (Wnt family member 5A), APP (amyloid beta precursor protein), and APC (adenomatous polyposis coli WNT signaling regulator) are regulators that promote CAVD progression.⁵⁹ Machine learning strategies combined with targeted RNA-seq have been used to map gene networks disrupted in human NOTCH1 haploinsufficient iPSC-derived endothelial cells; XCT790 was screened to correct the impaired gene network signature and prevent CAVD initiation and progression in *Notch1/mTR^G2* mice.²⁵ With the emergence of novel sequencing technologies, such as single-molecule protein sequencing and single-cell metabolomics, network-guide tactics will become a more powerful tool to accelerate the study of CAVD pathogenesis and the discovery of novel drugs.⁶⁰

[25] Theodoris, C. V. *et al.* Network-based screen in iPSC-derived cells reveals therapeutic candidate for heart valve disease. *Science* **371**, doi:10.1126/science.abd0724 (2021).

[33] Aguado, B. A. *et al.* Genes That Escape X Chromosome Inactivation Modulate Sex

- Differences in Valve Myofibroblasts. *Circulation* **145**, 513-530, doi:10.1161/CIRCULATIONAHA.121.054108 (2022).
- [35] Xu, K. *et al.* Cell-Type Transcriptome Atlas of Human Aortic Valves Reveal Cell Heterogeneity and Endothelial to Mesenchymal Transition Involved in Calcific Aortic Valve Disease. *Arteriosclerosis, thrombosis, and vascular biology* **40**, 2910-2921, doi:10.1161/ATVBAHA.120.314789 (2020).
- [53] Blaser, M. C., Kraler, S., Luscher, T. F. & Aikawa, E. Network-Guided Multiomic Mapping of Aortic Valve Calcification. *Arteriosclerosis, thrombosis, and vascular biology* **43**, 417-426, doi:10.1161/ATVBAHA.122.318334 (2023).
- [54] Blaser, M. C., Kraler, S., Luscher, T. F. & Aikawa, E. Multi-Omics Approaches to Define Calcific Aortic Valve Disease Pathogenesis. *Circulation research* **128**, 1371-1397, doi:10.1161/CIRCRESAHA.120.317979 (2021).
- [55] Garg, V. *et al.* Mutations in NOTCH1 cause aortic valve disease. *Nature* **437**, 270-274, doi:10.1038/nature03940 (2005).
- [56] Schlotter, F. *et al.* Spatiotemporal Multi-Omics Mapping Generates a Molecular Atlas of the Aortic Valve and Reveals Networks Driving Disease. *Circulation* **138**, 377-393, doi:10.1161/CIRCULATIONAHA.117.032291 (2018).
- [57] Decano, J. L. *et al.* A disease-driver population within interstitial cells of human calcific aortic valves identified via single-cell and proteomic profiling. *Cell Rep* **39**, 110685, doi:10.1016/j.celrep.2022.110685 (2022).
- [58] Chen, X. *et al.* Hepatic steatosis aggravates atherosclerosis via small extracellular vesicle-mediated inhibition of cellular cholesterol efflux. *J Hepatol*, doi:10.1016/j.jhep.2023.08.023 (2023).
- [59] Blaser, M. C. *et al.* Multiomics of Tissue Extracellular Vesicles Identifies Unique Modulators of Atherosclerosis and Calcific Aortic Valve Stenosis. *Circulation* **148**, 661-678, doi:10.1161/CIRCULATIONAHA.122.063402 (2023).
- [60] Eisenstein, M. Seven technologies to watch in 2023. *Nature* **613**, 794-797, doi:10.1038/d41586-023-00178-y (2023).

8. The graphical abstract should be redrawn to concisely convey the main message(s) of the submitted article.

Response:

We were very sorry for the unsuccinct graphical abstract. As suggest, we have provided a new graphic abstract that conveys the main messages to replace the old one.

Following are our revisions to the manuscript:

Page 5 of Revised Manuscript:

Dual-active targeting strategy for CAVD

Dual-active targeting drug delivery strategy for the treatment of calcific aortic valve disease (CAVD). Under the magnetic field navigation, the magnetic nanoparticles functionalized with protease-activated receptor 2 (PAR2) ligands deliver an anti-calcification drug XCT790 into the diseased cells within the calcified aortic valves, inhibits the osteogenic differentiation of valvular interstitial cells, and alleviates CAVD progression.

9. What was the rationale of using two distinct strategies (Tukey vs Dunn's) to mitigate any inflate in type-I errors due to multiple testing?

Response:

Thank you for your valuable suggestions for our statistical strategy. We agree that there are differences between Tukey and Dunn's in the way and rigor of mitigating type-I errors. In fact, Tukey's honest significant difference (HSD) assumes that the dependent variable is normally distributed and so that it is not appropriate as a post-hoc test for non-parametric omnibus test (i.e., Kruskal–Wallis test). The only real non-parametric post-hoc test for unpaired data is the Dunn's test. Refer to previous research, for non-parametric data, Mann–Whitney U test was performed for the comparisons of two groups, and Kruskal–Wallis test followed by Dunn's multiple comparisons test were applied for the comparisons of multiple groups. (Iqbal, Schlotter et al. Sortilin enhances fibrosis and

MINOR:

1. A variety of minor issues in the abstract have been identified that should be revised:

- please avoid general statements such as "serious threat to the health of older adults"
- based on current evidence, we cannot say whether so far tested medical therapies (statins, RANKL inhibition, among others) were ineffective due to inappropriate drug delivery systems or the fact that these targets do not significantly contribute to CAVD pathogenesis in vivo. Thus, the statement 'A proper drug delivery system can be a potential solution to CAVD drug therapy' should be rewritten.
- the statement 'osteogenically differentiated VICs ... after valve calcification' is a pleonasm. Please rephrase.
- the specific mouse models used should be stated
- the sentence 'in induced animal model' should be rewritten

Response:

Thank you for your careful checks. We have revised the Abstract section and hope it meets your requirements.

- The general statements have been replaced by "Calcific aortic valve disease (CAVD) is a prevalent cardiovascular disease with no available drugs capable of effectively preventing its progression."

- The statements have been amended to "Hence, an efficient drug delivery system could serve as a valuable tool in drug screening and potentially enhance therapeutic efficacy."

- The statements have been amended to "Herein, protease-activated-receptor 2 (PAR2) expression was up-regulated on the plasma membrane of osteogenically differentiated valvular interstitial cells (VICs)."

- The specific mouse model has been stated as "a high-fat diet-fed low-density lipoprotein receptor-deficient (*Ldlr*^{-/-}) mouse model."

- The sentence has been corrected to "To our knowledge, this work presents the first effective targeted drug delivery system for treating CAVD in a murine model."

Following are our revisions to the manuscript:

Page 4 of Revised Manuscript:

Calcific aortic valve disease (CAVD) is a prevalent cardiovascular disease with no available drugs capable of effectively preventing its progression. Hence, an efficient drug delivery system could serve as a valuable tool in drug screening and potentially enhance therapeutic efficacy. However, due to the rapid blood flow rate associated with aortic valve stenosis and the lack of specific markers, achieving targeted drug delivery for CAVD has proved to be challenging. Herein, protease-activated-receptor 2 (PAR2) expression was up-regulated on the plasma membrane of osteogenically differentiated valvular interstitial cells (VICs). Accordingly, a magnetic nanocarrier functionalized with PAR2-targeting hexapeptide for dual-active targeting drug delivery was developed. The nanocarriers effectively delivered XCT790—an anti-calcification drug—to the calcified aortic valve under extra magnetic field navigation. Consequently, VICs osteogenic differentiation was inhibited, and aortic valve calcification and stenosis were alleviated in a high-fat diet-fed low-density lipoprotein receptor-deficient (*Ldlr*^{-/-}) mouse model. To our knowledge, this work presents the first effective targeted drug delivery system for treating CAVD in a murine model. Hence, the composite drug delivery platform combining PAR2- and magnetic-targeting may represent a promising therapeutic strategy for CAVD.

2. It is Tukey and not Turkey's multiple comparison test. Please revise.

Response:

We were very sorry for the spelling mistake. According to your suggestion, "Turkey's multiple comparison test " has been corrected as " Tukey's multiple comparison test ".

Following are our revisions to the manuscript:

Page 66 of Revised Manuscript:

The two-tailed, unpaired Student's t-test was applied for comparisons involving two groups, one-way ANOVA followed by Tukey's test was used for comparisons involving ≥ 3 groups.

3. A flow-chart of the patients included in the present study should be provided (could be Supplement).

Response:

Thank you for your comment. A detailed tissue selection flowchart was drawn up

(Supplementary Fig. 23)

Following are our revisions to the manuscript:

Page 48-49 of Revised Manuscript:

All experiments involving humans were conducted in accordance with the Declaration of Helsinki and were approved by The Second Affiliated Hospital, Zhejiang University (No. IRB-2022-0085). Written informed consent was provided by each participant. Calcified aortic valve leaflets were obtained from severe aortic stenosis patients who underwent aortic valve replacement surgery, while non-calcified aortic valves were obtained from heart transplant recipients with Stanford type A acute aortic dissection or aortic valve regurgitation. Valves from patients with rheumatic disease, infective endocarditis, congenital valvular disease, or diabetes were excluded (Supplementary Fig. 24). Leaflets were divided into four pieces immediately after acquisition: (1) digested with collagenase type II for valvular interstitial cell isolation and culture. (2) homogenized in radioimmunoprecipitation assay (RIPA) buffer for protein extraction and immunoblot; (3) embedded in paraffin for immunohistochemistry staining, Masson staining, and Alizarin Red staining; (4) embedded in optimal cutting temperature compound (OCT, Sakura Finetek, Japan) for immunofluorescence staining, H&E staining, and Alizarin Red staining.

Page 17 of Revised Supplementary Information

Supplementary Fig. 24 Flow-chart of the patients' tissues included and excluded in this study.

4. Is the expression of the reference protein (GAPDH) influenced by pro-calcific/pro-fibrotic conditions?

Response:

Thank you for your rigorous consideration. Two other housekeeping proteins β -actin and β -tubulin are used as reference genes and we found that the expression of GAPDH was not affected by the induction of osteogenic/myofibrogenic differentiation (Figure for Review #1).

Figure for Review #1

Immunoblot and quantification of osteogenic differentiation markers, myofibrogenic differentiation markers and three housekeeping proteins in hVICs.

5. Individual data points should be shown throughout (eg, in Figure 2B they are missing).

Response:

We are very sorry for our careless omissions. All individual data points are shown in the graph (revised Fig. 2c).

Following are our revisions to the manuscript:

Page 23 of Revised Manuscript:

Fig. 2

Zeta-potential of FeXCT, PFeXCT, and SK@PFeXCT; n = 3/group.

6. In the wire injury mouse model, was CAVD induced without the presence of hyperlipidaemia?

Response:

According to previous studies,^{49, 52} the wire injury model was conducted in wild-type C57BL/6J mice fed with standard chow diet.

[Ref]

- [49] Artiach, G. *et al.* Omega-3 Polyunsaturated Fatty Acids Decrease Aortic Valve Disease Through the Resolvin E1 and ChemR23 Axis. *Circulation* **142**, 776-789, doi:10.1161/CIRCULATIONAHA.119.041868 (2020).
- [52] Honda, S. *et al.* A novel mouse model of aortic valve stenosis induced by direct wire injury. *Arterioscler Thromb Vasc Biol* **34**, 270-278, doi:10.1161/ATVBAHA.113.302610 (2014).

7. The advantages/disadvantages of the two mouse models used should be more comprehensively discussed.

Response:

Thank you for pointing this important issue out. We agree that it is important to choose the right modeling method for different disease backgrounds, and we discuss the advantages and disadvantages of diet-induced hypercholesterolemic murine model and wire injury (WI) model in CAVD, hoping to meet your requirements.

Following are our revisions to the manuscript:

Page 44-45 of Revised Manuscript:

Various animal models have been used to study the occurrence of CAVD, including a transgenic aging model, hyperlipidemia model, guidewire injury model, and the recently published hyperlipidemia-based wire injury model, etc.^{40,49,50} *Ldlr*^{-/-} mouse fed with HFD for 6 months is a classic mouse model of aortic valve calcification, which simulates the process of valve calcification caused by non-genetic factors, including lipid deposition, inflammatory cell invasion, plaque formation, and thickening and calcification.⁵¹ The wire injury (WI) model was generated by inserting and rotation a guidewire to cause mechanical damages on the endothelium. The damages further induced infiltration of blood cells, inflammation, and valve thickening.⁵² The WI model was applied on wild type mice, which have normal blood lipid level, so the pathological process was different from the *Ldlr*^{-/-} mouse fed with HFD. Moreover, the damage degree can be controlled by operation during wire injury, which lead to higher repeatability of the modeling. In this study, we used two mouse models with different pathological mechanisms to study the expression of PAR2. The results from both of the models

[Ref]

- [40] Bouchareb, R. *et al.* Activated platelets promote an osteogenic programme and the progression of calcific aortic valve stenosis. *Eur Heart J*, doi:10.1093/eurheartj/ehy696 (2018).
- [41] Iqbal, F. *et al.* Sortilin enhances fibrosis and calcification in aortic valve disease by inducing interstitial cell heterogeneity. *Eur Heart J*, doi:10.1093/eurheartj/ehac818 (2023).
- [49] Artiach, G. *et al.* Omega-3 Polyunsaturated Fatty Acids Decrease Aortic Valve Disease Through the Resolvin E1 and ChemR23 Axis. *Circulation* **142**, 776-789, doi:10.1161/CIRCULATIONAHA.119.041868 (2020).
- [50] Dutta, P. *et al.* KPT-330 Prevents Aortic Valve Calcification via a Novel C/EBPbeta Signaling Pathway. *Circulation research* **128**, 1300-1316, doi:10.1161/CIRCRESAHA.120.318503 (2021).
- [51] Aikawa, E. *et al.* Multimodality molecular imaging identifies proteolytic and osteogenic activities in early aortic valve disease. *Circulation* **115**, 377-386, doi:10.1161/CIRCULATIONAHA.106.654913 (2007).
- [52] Honda, S. *et al.* A novel mouse model of aortic valve stenosis induced by direct wire injury. *Arterioscler Thromb Vasc Biol* **34**, 270-278, doi:10.1161/ATVBAHA.113.302610 (2014).

We would like to take this opportunity to thank you for all your time involved and this great opportunity for us to improve the manuscript. We hope you will find this revised version satisfactory.

Sincerely,

The Authors

-----End of Reply to Reviewer #1-----

Response to reviewer #2

Reviewer #2 (Remarks to the Author):

This is an interesting study on a drug delivery system for calcific aortic valve disease.

There are a number of issues to be addressed:

Dear reviewer:

We feel great thanks for your professional review work on our article. As you are concerned, there are several problems that need to be addressed. According to your nice suggestions, we have made extensive corrections to our previous draft, the detailed corrections are listed below.

1. The choice of delivery system should be better rationalized, since the platform itself (i.e. drug loaded PLGA NP) has been well known for many years.

Response:

Thank you for the suggestion. Indeed, the drug delivery system based on PLGA has been developed for many years, where it was chosen as a carrier herein for the following reasons: First, while the novelties of this study are mainly the concept that using a targeted delivery system to increase drug treatment effect to CAVD and the identification of PAR2 as a binding site of calcified valve, we hope the carrying system to be with more clinical translational potential, thus we chose materials from the FDA approved products. PLGA is a promising biodegradable and biocompatible polymer, which have been approved by FDA for clinical practice. (Nkanga, Fisch et al. Clinically established biodegradable long acting injectables: An industry perspective. *Adv Drug Deliv Rev*, 2020, 167: 19-46.) Second, PLGA has shown great capacity in loading hydrophobic drugs and nanoparticles. (Bao, Tian et al. Exosome-loaded degradable polymeric microcapsules for the treatment of vitreoretinal diseases. *Nat Biomed Eng*, 2023,) As a fat-soluble drug, XCT790 can be well loaded and sustained released by PLGA, which make it possible to reduce dosing frequency and drug side effects for the treatment of chronic diseases like CAVD. Third, a growing number of studies have revealed the surface modification ability of PLGA, which made it a suitable candidate polymer for constructing a PAR2-targeting magneto-responsive drug delivery systems. (Luo, Lu et al.

Neutrophil hitchhiking for drug delivery to the bone marrow. *Nat Nanotechnol*, 2023, 18: 647-656.) **Related statements** have been rephrased accordingly and marked in red in the revised manuscript.

Following are our revisions to the manuscript:

Page 19 of Revised Manuscript:

To verify the feasibility of PAR2 as a binding site of calcified valve, FDA-approved materials with strong clinical translational potential are more appropriate. PLGA is a promising biodegradable and biocompatible polymer with surface modification potential, and can well load hydrophobic drugs and nanoparticles.^{37,38} More specifically, an oil-in-water (O/W) emulsion solvent evaporation method was used to synthesize the nanoparticle core comprising PLGA, XCT790, and 10-nm MNPs, with bovine serum albumin (BSA, an amphiphile) dissolved in water as a stabilizer. The BSA molecules on the particle surface provided amino groups that became covalently conjugated to OHC-PEG-CHO via condensation of aldehyde and amino groups. OHC-PEG-CHO also functioned as the crosslinker of PAR2-targeting peptides (SLIGKV-NH₂ or SLIGRL-NH₂) (**Fig. 3a**).

[ref]

[37] Bao, H. *et al.* Exosome-loaded degradable polymeric microcapsules for the treatment of vitreoretinal diseases. *Nat Biomed Eng*, doi:10.1038/s41551-023-01112-3 (2023).

[38] Luo, Z. *et al.* Neutrophil hitchhiking for drug delivery to the bone marrow. *Nat Nanotechnol* **18**, 647-656, doi:10.1038/s41565-023-01374-7 (2023).

2. Explain the regulatory status of XCT790.

Response:

Thank you for pointing out this important information. Actually, to our knowledge, XCT790 is currently in preclinical studies. As there are no clinical drug for CAVD has been reported, we chose a promising compound, XCT790, as a proof of concept. We also hope our targeting delivery system can provide an option to accelerate the exploration of CAVD medication, because the drug administration strategy can also significantly affect the efficacies and risks of drugs.

Following are our revisions to the manuscript:

Page 46-47 of Revised Manuscript (Limitations in the Discussion section):

This study has certain limitations. First, other cell surface proteins may also be appropriate

markers for nanoparticle targeting. However, as an exploratory study, our results sufficiently demonstrate that PAR2 is elevated in aortic valve calcification and represents an effective anchor for targeting drug delivery. Second, the optimal application doses of the synthesized nanoparticles were not confirmed via rigorous pharmacological and metabolic testing. Although no significant systemic toxicity was observed in vitro or in vivo (**Supplementary Fig. 19-21**), further studies are essential to optimize drug dosage. Third, a preclinical drug was used in this study as the model drug for CAVD therapy as no clinical drugs have been reported. Hence, the systemic toxicology and adverse effects to other organs require further assessment.

3. Need to demonstrate avoidance of the XCT790 side effects in a model where the authors show that they do occur with the free drug.

Response:

Thank you for your valuable concern. As suggested, we tested the occurrence of liver steatosis and fibrosis in mice fed a high-fat diet, as it has been previously reported that liver-specific ERR α -deficient affects the development of liver steatosis and fibrosis and XCT790 is an ERR α inhibitor.

Following are our revisions to the manuscript:

Page 46-47 of Revised Manuscript: (Limitations in the Discussion section):

This study has certain limitations. First, other cell surface proteins may also be appropriate markers for nanoparticle targeting. However, as an exploratory study, our results sufficiently demonstrate that PAR2 is elevated in aortic valve calcification and represents an effective anchor for targeting drug delivery. Second, the optimal application doses of the synthesized nanoparticles were not confirmed via rigorous pharmacological and metabolic testing. Although no significant systemic toxicity was observed in vitro or in vivo (**Supplementary Fig. 19-21**), further studies are essential to optimize drug dosage. Third, a preclinical drug was used in this study as the model drug for CAVD therapy as no clinical drugs have been reported. Hence, the systemic toxicology and adverse effects to other organs require further assessment. Here, we preliminarily assessed the side effects of XCT790 in HFD-induced liver lesion, as it has been previously reported that liver-specific ERR α -deficient affects the development of liver steatosis and fibrosis, which may lead to non-alcoholic fatty liver disease (NAFLD).²⁶ Oil red O staining showed that, when compared to the HFD group, steatosis is slightly increased in response to XCT790, while there was no significant increase in the SR@PF α XCT group (**Supplementary Fig. 23a, b**). qPCR confirmed that the adipogenic-related genes, ATP-citrate lyase (ACLY) and fatty acid synthase (FASN), were up-regulated in the XCT790 group (**Supplementary Fig. 23c**). However, no difference was found in collagen deposition, as indicated by Picro-Sirius Red staining, or profibrotic gene transcription, as evidenced by qPCR analysis (**Supplementary Fig. 23d, e**). Nevertheless, more in-depth studies on XCT790 metabolism and side effects are needed before XCT790 enters clinical trials. With the assistance

of our CAVD targeting drug delivery system, the dosage and frequency of drug administration can be significantly reduced, which may decrease the potential systemic toxicity without impacting the curative effect.

Page 16-17 of Revised Supplementary Information:

[Ref]

- [26] Yang, M. *et al.* Dysfunction of estrogen-related receptor alpha-dependent hepatic VLDL secretion contributes to sex disparity in NAFLD/NASH development. *Theranostics* **10**, 10874-10891, doi:10.7150/thno.47037 (2020).

4. The ratio of ligand to nanoparticle should be determined experimentally.

Response:

Thank you for the great comment. To optimize the ratio of ligand and nanoparticles, PLGA cores were synthesized with Cy7-labelled BSA, and pipetted into Rhodamine B-labelled SLIGKV peptides at different concentration. After centrifuge and washed with water thrice, nanoparticles were collected for relative quantification. The fluorescence intensity at excitation/emission = 545/570 was determined as the concentration of SLIGKV, and 750/780 as the concentration of nanoparticles. With the concentration of ligand increased from 3.91 μg mL⁻¹ to 0.125 mg mL⁻¹, the fluorescence intensity ratio gradually increased, indicated the increase of peptide on the surface of the

nanoparticle. While the concentration of ligand exceeds 0.125 mg mL^{-1} , the increase in fluorescence intensity slows down significantly, indicated that the peptides on the nanoparticle surface are gradually saturated (**Supplementary Fig. 7**). In this study, we used a concentration of 0.2 mg/mL to ensure a high peptide grafting ratio.

Following are our revisions to the manuscript:

Page 20 of Revised Manuscript (Results section):

First, to optimize the ratio of ligands and nanoparticles, BSA was labeled with Cy7 and SLIGKV peptides were labeled with Rhodamine B; relative quantification was conducted according to the fluorescence intensity ratio (Rhodamine B/Cy7). When the ligand concentration increased from $3.91 \text{ } \mu\text{g mL}^{-1}$ to 0.125 mg mL^{-1} , the fluorescence intensity ratio gradually increased, indicating an increase in the number of peptides on the nanoparticle surface. However, when the ligand concentration exceeded 0.125 mg mL^{-1} , the increase in fluorescence intensity slowed significantly, indicating that the nanoparticle surface had become saturated with peptides (**Supplementary Fig. 7**). In this study, we used a concentration of 0.2 mg/mL to ensure a high peptide grafting ratio.

Page 56-57 of Revised Manuscript (Methods section):

The relative content of ligands on the nanoparticles was determined by fluorescence detection. In brief, different concentration of Rhodamine B-labelled SLIGKV-NH₂ peptides was used to modify PEGylated PLGA core which was synthesized with Cy7-labelled BSA, and the relative quantification was conducted according to the ratio of fluorescence intensity (Rhodamine B / Cy7) with Spark® multimode microplate reader (Tecan, Switzerland).

Page 6 of Revised Supplementary Information:

Supplementary Fig. 7. Efficiency of different SLIGKV-NH₂ concentrations in conjugation with nanoparticles.

Fluorescence intensity of the Cy7 (BSA, indicate nanoparticles concentration) and Rhodamine B (indicate SLIGKV-NH₂ concentration) in nanoparticles synthesized with different concentrations of SLIGKV-NH₂; $n = 3/\text{group}$. Data are shown as the mean \pm SD.

5. Controls of nanoparticle with mismatch peptide are missing. The differences in results could simply be that the 'targeted' NP is different. This should be added throughout.

Response:

Thank you for your constructive suggestion. Nanoparticles were conjugated with mismatch peptides SLKGIV-NH₂ (correct sequence is SLIGKV-NH₂), and incubated with hVICs. The fluorescence intensity of hVICs was detected with flow cytometry. As shown in **Supplementary Fig. 9**, the mean fluorescence of hVICs cultured with nanoparticles conjugated with mismatch peptide was significantly decreased when compared with those cultured with SK@PFe, demonstrated that it is SLIGKV-NH₂ peptide-PAR2 interactions which mediated the targeting ability of the drug delivery system developed herein.

Following are our revisions to the manuscript:

Page 21-22 of Revised Manuscript:

To further verify the specific targeting ability (i.e., the specific binding of SLIGKV-NH₂⁻ with PAR2), nanoparticle cores were conjugated with a mismatch peptide (SLKGIV-NH₂⁻) and incubated with hVICs. The mean fluorescence of hVICs cultured with nanoparticles conjugated with the mismatch peptide was significantly decreased compared with those cultured with SK@PFe (**Supplementary Fig. 9**).

Page 7 of Revised Supplementary Information:

Supplementary Fig. 9. Targeting ability of mismatched PAR2-ligand functionalized nanoparticles toward hVICs.

Flow cytometry histogram and quantification of SLIGKV-NH₂ or SLKGIV-NH₂ functionalized nanoparticle uptake by hVICs; *n* = 3/group

6. Control of drug delivery without the magnetic field should also be added.

Response:

Thank you for the insightful comment. As suggested, the targeting efficiency of SR@PFe without magnetic navigation or without PAR2 targeting was further detected in HFD fed *Ldlr*^{-/-} mice, respectively. IVIS revealed that, in the aortic root isolated from the ND-fed *Ldlr*^{-/-} mice, a slightly enhanced fluorescence signal was observed in the SR@PFeIR EMF group compared with the PFeIR without EMF group. In the HFD-fed mice, both the grafting of PAR2 ligand and the addition of EMF can enhance the targeting ability of bare PLGA, and this two functionalization can synergistically enhance the targeting ability. (**Fig. 6b, c**).

Fluorescence microscopy results showed that, in the aortic valve of ND-fed mice, regardless of the type of nanoparticles injected, almost no fluorescence can be observed, indicated that SR@PFe did not enrich in the non-diseased aortic valves (**Figure 6d, e**). Under the EMF navigation, a small number of SR@PFe were enriched onto the lesion leaflets of mice fed with HFD for 3 months, while the mice fed with HFD for 5 months had a larger number of nanoparticles on the lesion leaflets (**Figure 6d, e**). The fluorescence intensity in SR@PFe with magneto group was higher than that of PAR2-targeting alone and magnetic targeting alone, indicating that the dual-targeting effects worked synergistically (**Figure 6d, e**). Additionally, enrichment of nanoparticles in the atherosclerotic aortic sinus was examined. The fluorescence intensity in the aortic sinus was much higher in the SR@PFe with magneto group when compared with single targeting control groups. (**Supplementary Fig. 13**).

Following are our revisions to the manuscript:

Page 30-32 of Revised Manuscript:

To detect the targeting capabilities and systematically evaluate the in vivo distribution of the nanoplatfrom, IR780-labeled nanoparticles (PFeIR and SR@PFeIR) were synthesized, and a mouse CAVD model was established in HFD-fed *Ldlr*^{-/-} mice. PFeIR and SR@PFeIR were injected intravenously, and a magnet with a diameter of 5 mm was placed on the mouse's chest above the aortic root for 15 min immediately after injection. Mice were euthanized 6 h after injection, and the major organs were harvested for fluorescence intensity detection (**Fig. 6a**). The distribution of the nanoplatfrom in the heart, aorta, liver, kidneys, lungs, and spleen was

observed (**Supplementary Fig. 12**), with a particular focus on the heart and aorta (**Fig. 6b**). In the aortic root isolated from the ND-fed *Ldlr*^{-/-} mice, a slightly enhanced fluorescence signal was observed in the SR@PFeIR EMF group compared with the PFeIR without EMF group. In the HFD-fed mice, grafting of PAR2 ligand and adding EMF enhanced the targeting ability of bare PLGA; this dual functionalization synergistically enhanced the targeting ability. (**Fig. 6b, c**). These results demonstrated the targeting capability of SR@PFeIR under EMF in a CAVD mouse model.

To specifically examine the ability of the SR@PFe nanoplatform to target lesioned aortic valves in different disease stages, *Ldlr*^{-/-} mice fed an HFD for various times were administered Cy5-labeled SR@PFe (SR@PFeCy5); hearts were subsequently collected and sectioned for analysis. In the aortic valve of ND-fed mice, regardless of the type of nanoparticles injected, relatively no fluorescence was observed, indicating that SR@PFe were not enriched in non-diseased leaflets (**Figure 6d, e**). Under EMF navigation, few SR@PFe were enriched in the lesion leaflets of mice fed with HFD for 3 months, while the mice fed with HFD for 5 months had more nanoparticles on the lesion leaflets (**Figure 6d, e**). The fluorescence intensity of SR@PFe in the EMF group was higher than that of PAR2-targeting alone or EMF alone, indicating that the dual-targeting worked synergistically (**Figure 6d, e**). Notably, most of the SR@PFeCy5 was accumulated in the aortic side of the aortic valve leaflets, while negligible amounts were observed on the ventricular side (**Fig. 6f**), possibly due to a vortex area with low flow velocity on the aortic side and the hierarchical expression of PAR2. Additionally, considering that endothelial denudation occurs with valve calcification, we speculate that SR@PFe may enter the subendothelial layer through gaps between the endothelium. Immunofluorescence confirmed that isolectin B4 was not continuous in the calcified leaflets, and SR@PFeCy5 aggregated in the areas of endothelial defects (**Fig. 6g**). Moreover, fluorescence was detected in the atherosclerotic aortic sinus, which agrees with the higher PAR2 expression in this region.²³ As expected, copious amounts of fluorescence were detected in the aortic sinus of the HFD-fed *Ldlr*^{-/-} mice, and more SR@PFeCy5 nanoparticles were detected in the aortic root under magnetic navigation when compared with the single-targeting control groups (**Supplementary Fig. 13**).

Fig. 6. Targeting ability of SR@ PFe in CAVD mice.

(a) Experimental outline. (b) Ex vivo IVIS images of *Ldlr*^{-/-} mouse heart and aorta 4 h after intravenous injection of nanoparticles. Green circles indicate aortic roots. (c) Quantification of fluorescence intensity in the aortic root area (green circles) relative to the left ventricle apex of the heart; $n = 7/\text{group}$. (d) Fluorescence images of aortic valves of HFD fed *Ldlr*^{-/-} mice 4 h after injected with nanoparticles. (e) Quantification of Cy5 mean fluorescence in the aortic valves; $n = 7/\text{group}$. (f) Fluorescence intensity distribution in the aortic valve (white arrow in the image on the left). (g) Co-labelled isolectin B4 and Cy5. Values are presented as mean \pm SD.

Page 9 of Revised Supplementary Information:

Supplementary Fig. 13. SR@PFeCy5 is enriched in atherosclerotic aortic sinus. Fluorescence images of aortic sinus of HFD fed *Ldlr*^{-/-} mice 4 h post-nanoparticle injection; *n* = 7/group.

7. What is the nature of the IONPs used? Are they oleic acid coated? This detail is missing.

Response:

We are very sorry for our carelessness. Fe₃O₄ nanoparticles (diameter: 8-10 nm, purchased from XFNANO Materials, Nanjing, China) modified with 3-(Trimethoxysilyl)propyl methacrylate (TPM) were used. The hydrophobic methacrylate groups modified on the Fe₃O₄ nanoparticles can enhance their binding ability to PLGA. The particle information is listed below and added to Methods.

Page 54 of Revised Manuscript (Methods section):

Materials for nano-carrier synthesis

Carboxyl-terminated Poly(lactic-co-glycolic acid) (PLGA, 75:25, MW: 40,000–60,000) was purchased from Daigang Biomaterials (Jinan, China); α,ω -Diformyl poly(ethylene glycol) (OHC-PEG-CHO, MW: 2,000) was purchased from Yusi Pharma (Chongqing, China); Fe₃O₄ nanoparticle (diameter: 8–10 nm) was purchased from XFNANO Materials (Nanjing, China); 3-(Trimethoxysilyl)propyl methacrylate (TPM) was provided by Aladdin Biochemical Tech (Shanghai, China). BSA and IR-780 iodide were purchased from Merck, while Cyanine5 carboxylic acid (Cy5-COOH) was provided by Duofluor (Wuhan, China). XCT790 was obtained from GlpBio (CA, USA). The PAR2 ligand peptide (SLIGKV-NH₂ and SLIGRL-NH₂) was synthesized by Synpeptides (Nanjing, China) while Rhodamine B labeled SLIGKV-NH₂ were synthesized by Allpeptide (Hangzhou, China), and Cy7-labeled BSA was synthesized by Qiyuebio (Xi'an, China).

8. The results in Fig 5B/C are slightly curious since there is higher signal everywhere in the targeted NP image. The quantification analysis should be redone normalizing to the signal in the heart or other bystander organ.

Response:

Thank you for pointing this important issue out. We re-quantitated and reanalyzed the IVIS data, the fluorescence intensity at all aortic roots was normalized to the signal in the left ventricle apex, and the images were replaced. The data quality was much

improved as shown in revised (Figure 6b).

Following are our revisions to the manuscript:

Page 33 of Revised Manuscript:

Figure 6

(b) Ex vivo IVIS images of *Ldlr*^{-/-} mouse heart and aorta 4 h after intravenous injection of nanoparticles. Green circles indicate aortic roots. (c) Quantification of fluorescence intensity in the aortic root area (green circles) relative to the left ventricle apex of the heart; $n = 7$ /group.

We would like to take this opportunity to thank you for all your time involved and this great opportunity for us to improve the manuscript. We hope you will find this revised version satisfactory.

Sincerely,

The Authors

-----End of Reply to Reviewer #2-----

Reviewer #3 (Remarks to the Author):

In this study, the authors utilized magnetic nanoparticles loaded with XCT790, designed to target PAR2 receptor for the treatment of CAVD. The notable results of the manuscript are:

- PAR2 expression is elevated in calcific in both calcified aortic valves and osteogenic differentiated VICs
- the construction of magnetic nanoparticles functionalized with PAR2 ligands and carrying XCT790 that accumulate within calcified aortic valves and osteodifferentiated VICs
- magnetic nanoparticles
- this internalization leads to the inhibition of metabolic reprogramming in VICs by regulating the expression of PDK4.

This research stands out as one of the few studies employing magnetic nanoparticles for the purpose of targeting particular molecules involved in CAVD, with the majority of existing literature primarily concentrating on targeting cancer-related aspects. Lately, magnetic nanoparticles gathered attention due to their potential for targeting specific tissues thorough magnetically guided delivery. However, this technique requires further refinement to achieve optimal precision in delivering to specific tissues.

The authors used suitable and vast methodologies, presented in detail. The statistical analysis was clearly explained and the quality of the data is adequate. The specific results are clearly stated and some implications in the field are explained. However, some improvements should be made. My comments to the authors are:

Dear reviewer:

We would like to thank you for your careful reading, helpful comments, and constructive suggestions, which has significantly improved the presentation of our manuscript.

We have carefully considered all your comments and revised our manuscript accordingly. The manuscript has also been double-checked, and the typos and grammar errors we found have been corrected. In the following section, we summarize

our responses to each comment. We believe that our responses have well addressed all your concerns.

1. The Introduction section should explain in an additional paragraph how SPIO NPs were used previously in targeting other diseases.

Response:

Thank you for the kind suggestion. Based on your suggestions, the application of magnetic nanoparticles (MNPs) in other diseases has been explained in Introduction. We found that both the term of “SPIO” and “MNPs” were used in our manuscript, we now used MNPs throughout the manuscript to be consistent.

Following are our revisions to the manuscript:

Page 7-8 of Revised Manuscript:

Magnetic targeting is a promising strategy for improving drug delivery efficiency. Due to the responsiveness to exogenous magnetic field (EMF), which is non-invasive and tissue-penetrating, magnetic nanoparticles (MNPs)-based drug delivery systems are highly anticipated, especially for targeting deep tissues. To date, MNPs have primarily been applied for the targeted treatment of tumors. Guided by EMF, MNPs increase the localization of antineoplastic drugs and serve as the source of magneto-thermodynamic therapy.¹³ Moreover, the combination of magnetic targeting and hyperthermia has been applied in anti-infection therapy within deep tissues (e.g., bacterial osteomyelitis in bone marrow), which are not readily accessible via systemic drug administration.¹⁴ Meanwhile, MNPs have been increasingly used to transport therapeutic agents to other pathological tissues, including the ischemic brain and heart.^{15,16} Additionally, mesenchymal stem cell-derived exosomes incorporated with MNPs have been directed to brain ischemic lesions, ultimately reducing the infarct volume and promoting motor function recovery.¹⁷ MNPs have also been conjugated with anti-CD63 and anti-myosin light chain (MLC) antibodies to increase heart function after myocardial infarction. More specifically, the MNPs capture endogenous circulating exosomes via the anti-CD63 antibody, causing accumulation in the ischemic heart under EMF navigation; subsequently, the anti-MLC antibodies bind to the damaged cardiomyocytes, releasing exosomes and increasing heart function.¹⁸ Although MNPs can be enriched around the diseased tissue under EMF guidance, the drug anchoring capacity must be enhanced to counteract the violent blood flow of the stenotic aortic valves. Moreover, targeting accuracy must be improved to enable nanoparticles to act on disease-causing valvular interstitial cells (VICs).

[Ref]

- [13] Chan, M.-H., Hsieh, M.-R., Liu, R.-S., Wei, D.-H. & Hsiao, M. Magnetically Guided Theranostics: Optimizing Magnetic Resonance Imaging with Sandwich-Like Kaolinite-Based

- Iron/Platinum Nanoparticles for Magnetic Fluid Hyperthermia and Chemotherapy. *Chemistry of Materials* **32**, 697-708, doi:10.1021/acs.chemmater.9b03552 (2020).
- [14] Qiao, Y. *et al.* Treatment of MRSA-infected osteomyelitis using bacterial capturing, magnetically targeted composites with microwave-assisted bacterial killing. *Nat Commun* **11**, 4446, doi:10.1038/s41467-020-18268-0 (2020).
- [15] Qiao, R. *et al.* Magnetic iron oxide nanoparticles for brain imaging and drug delivery. *Adv Drug Deliv Rev* **197**, 114822, doi:10.1016/j.addr.2023.114822 (2023).
- [16] Dadfar, S. M. *et al.* Iron oxide nanoparticles: Diagnostic, therapeutic and theranostic applications. *Adv Drug Deliv Rev* **138**, 302-325, doi:10.1016/j.addr.2019.01.005 (2019).
- [17] Kim, H. Y. *et al.* Mesenchymal stem cell-derived magnetic extracellular nanovesicles for targeting and treatment of ischemic stroke. *Biomaterials* **243**, 119942, doi:10.1016/j.biomaterials.2020.119942 (2020).
- [18] Liu, S. *et al.* Treatment of infarcted heart tissue via the capture and local delivery of circulating exosomes through antibody-conjugated magnetic nanoparticles. *Nat Biomed Eng* **4**, 1063-1075, doi:10.1038/s41551-020-00637-1 (2020).

2. Diabetic patients (1 in control and 2 in CAVD) should be excluded, as diabetes is known to accelerate CAVD and some results from these patients may interfere with overall statistics.

Response:

Thank you for your rigorous consideration. We agree that diabetes accelerates the development of CAVD, so we have removed these specimens and reanalyzed the results (**Fig. 2f**, **Figure for Reviewer#3**, and **Supplementary Table 1**).

Following are our revisions to the manuscript:

Page 16 of Revised Manuscript:

Fig. 2

(f) Protein expression of PAR2 in human aortic valves. Non-calcified group: $n = 12$; calcified group: $n = 15$.

Figure for Reviewer#3

Immunoblot of PAR2 protein expression in human aortic valves. #N9, #C12, and #C14 were excluded because those tissues were donated by patients with diabetes mellitus.

3. How was the aortic valve area (in cm², presented in Table S1) calculated? Is it the aortic valve opening? Why was this preferred instead of calcific area?

Response:

Thank you for your valuable concern. The aortic valve area (AVA) was obtained using the Philips EPIQ color ultrasound diagnostic system, and calculated via the well-validated continuity equation concept that the stroke volume ejected through the left ventricular outflow tract (LVOT) all passes through the stenotic aortic valve. The AVA calculated by continuity equation is effective valve area (the flow area when flowing through the valve), not the anatomical valve area.

Indeed, computed tomography (CT) is widely used to image vascular calcification, and CT calcium scoring is a well-established method for quantifying coronary calcium volume. However, CT aortic valve calcium scoring (CT-AVC) may ignore non-calcified leaflets thickening, such as fibrosis, which may be the main cause of aortic valve stenosis in some patients. Therefore, in clinical practice, echocardiography remains the first-line gold-standard diagnostic evaluation for aortic valve stenosis, and CT-AVC is used as one of the bases for selecting treatment strategies, which serves as a

complementary marker of stenosis severity.(Otto, Nishimura et al. 2020 ACC/AHA Guideline for the Management of Patients With Valvular Heart Disease: Executive Summary: A Report of the American College of Cardiology/American Heart Association Joint Committee on Clinical Practice Guidelines. *Circulation*, 2021, 143: e35-e71.)

4. Throughout the article, it must be made clear what results are from human patients, isolated VICs or mouse model. This is hard to follow sometimes. When appropriate, the title of the result and the title of the figure should also specify the experimental model.

Response:

Thank you for pointing this important issue out. We have examined all the subheadings of the results and figures, and highlighted the experimental model used in each title. And here we did not list the changes but marked in red in the revised manuscript.

5. In Section 2.4, the phrase stating that targeted NPs suspended in the cell culture media are not internalized into the cells because the distance is too high should be replaced. It is known from literature that based on ligand-receptor affinity, other NPs do not need magnetic field to bind to cellular membrane. This is also confirmed by the authors in 2.5, where they showed SK@PFeXCT binds to PAR2 receptor.

Response:

Thank you for the great suggestion. We have rephrased the statement to clearly convey the messages, which is that external magnetic fields enhanced the contact and binding of SK@PFeXCT with hVICs. The related statement has been explained in Results.

Following are our revisions to the manuscript:

Page 24-25 of Revised Manuscript:

To evaluate the magnetic field-guided cell-nanoparticle interactions, osteogenically differentiated hVICs were incubated with SK@PFeCy5 while magnets were placed below the Petri dish without covering the whole bottom. After 4 h of incubation, SK@PFeCy5 was

colocalized with hVICs in a magnetic field-dependent manner; that is, the regions exposed to the magnet exhibited higher fluorescence from the SK@PFeCy5 (**Fig. 4c, d**). Herein, the addition of EMF promoted the sedimentation and accumulation of the nano-cargoes, facilitating the contact between the cargo and hVICs seeded at the bottom of the Petri dish and promoting their internalization.

6. The results in Section 2.6 and Figure 5 were compared to NPs not under magnetic field control?

Response:

Thank you for your rigorous consideration. The curative effect of nanoparticles was conducted without EMF navigation, whether it was PFeXCT or SK@PFeXCT. The only variable between these two sets is the presence or absence of PAR2 functionalization in the surface of the nanoparticles.

7. The data in Section 2.9 does not present cytotoxicity of nanoparticles in magnetic field.

Response:

Thank you for pointing out this important question. We have presented the cytotoxicity of SK@PFe in magnetic field in **Supplementary Fig. 19d** and described related texts in the results in the revised manuscript. The PI-positive cells in the area with or without magnetic fields was calculated, respectively. This result indicated that enriched SK@PFe induced by magnetic fields did not cause VICs death.

Following are our revisions to the manuscript:

Page 41-42 of Revised Manuscript:

Additionally, the cytotoxicity of SK@PFe in the magnetic field was assessed; no difference was observed in the number of PI-positive cells in the area with or without magnetic fields (**Supplementary Fig. 19d**). These results confirm the absence of cytotoxicity induced by the nano-cargoes.

Page 14 of Revised Supplementary Information:

Supplementary Fig. 19.

(d) Representative calcein-AM/PI staining of hVICs incubation with SK@PFe nanoparticles under EMF. Scale bar = 200 µm

8. Knowing that iron-based nanoparticles induce oxidative stress, do the authors have any data on this matter?

Response:

Thank you for your constructive suggestion. Indeed, long-term high-dose iron stimulation leads to the production of intracellular oxidative stress and can even lead to ferroptosis.

Therefore, following your suggestion, the intracellular reactive oxygen species (ROS) production in VICs treated with SK@PFe or not were detected with dichloro-dihydro-fluorescein diacetate (DCFH-DA) probe. Flow cytometry assay demonstrated that SK@PFe did not alter the ROS generation in VICs (**Supplementary Fig. 20a**). Furthermore, the DCFH staining confirmed that the intracellular ROS production caused by magnetic field-guided nanoparticles enrichment is almost unobservable (**Supplementary Fig. 20b**) In addition, ferroptosis-related markers were further examined. The protein expression of glutathione peroxidase 4 (GPX4), which is an important regulator of ferroptosis, was not down-regulated by SK@PFe treatment (**Supplementary Fig. 20c**). Similarly, the content of malondialdehyde (MDA), an indicator of lipid peroxidation and an end product of ferroptosis, is not affected by SK@PFe stimulation (**Supplementary Fig. 20d**). Therefore, we suggested that the nano-cargo developed herein did not cause the intracellular ROS accumulation and ferroptosis.

Following are our revisions to the manuscript:

Page 42 of Revised Manuscript:

Long-term high-dose iron stimulation produces intracellular oxidative stress, which can cause ferroptosis and facilitate the osteogenic differentiation of VICs.⁴⁶ Thus, intracellular reactive oxygen species (ROS) production and ferroptosis were evaluated in SK@PFe-treated hVICs. Flow cytometry results demonstrated that SK@PFe treatment did not induce ROS generation in hVICs (**Supplementary Fig. 20a**). DCFH staining confirmed that the intracellular ROS production caused by magnetic field-guided nanoparticle enrichment was negligible (**Supplementary Fig. 20b**). Additionally, the abundance of glutathione peroxidase 4 (GPX4)—an important regulator of ferroptosis—was not down-regulated by SK@PFe treatment (**Supplementary Fig. 20c**). Similarly, the content of malondialdehyde (MDA)—an indicator of lipid peroxidation and an end product of ferroptosis—was not affected by SK@PFe stimulation (**Supplementary Fig. 20d**). Therefore, the nano-cargo did not elicit apparent intracellular ROS accumulation or ferroptosis.

Page 15 of Revised Supplementary Information:

Supplementary Fig. 20. ROS production and ferroptosis of hVICs incubated with SK@PFe nanoparticles.

(a) Flow cytometry histogram and quantification of DCFH fluorescence intensity of hVICs incubated with or without SK@PFe nanoparticles; $n = 6$ /group. (b) Representative DCFH staining of hVICs incubated with SK@PFe nanoparticles under EMF. Scale bar = 200 μm. (c) Western blot and quantification of GPX4 in hVICs cultured with or without SK@PFe nanoparticles; $n = 6$ /group. (d) Malondialdehyde (MDA) content in hVICs cultured with or

without SK@PFe nanoparticles; $n = 6/\text{group}$.

[Ref]

[46] Xu, R., Huang, Y., Zhu, D. & Guo, J. Iron promotes Slc7a11-deficient valvular interstitial cell osteogenic differentiation: A possible mechanism by which ferroptosis participates in intraleaflet hemorrhage-induced calcification. *Free Radic Biol Med* **184**, 158-169, doi:10.1016/j.freeradbiomed.2022.03.013 (2022).

9. Taking into account that this is an exploratory study, as the authors claim themselves, the conclusion that the designed NPs could solve the clinical problem of CAVD is overestimated. This study and other using magnetic NPs need further validation in order to interpret the results in such a manner.

Response:

Thank you for your rigorous consideration. We acknowledge that this is an exploratory experiment that is still a long way from it can be implemented in clinical practice. The statement has been amended to “This study proposes the design of a drug delivery system with dual targeting capabilities, which may provide some insight for the design and translation of precision therapies for CAVD.”

Following are our revisions to the manuscript:

Page 47-48 of Revised Manuscript:

In this study, we identified a cell surface anchoring site for calcified VICs (i.e., PAR2) and devised a targeting nanocarrier with a PAR2-binding peptide shell as the anchor and a MNP core in PLGA as the navigator. This system achieved enhanced intracellular drug delivery within calcified aortic valves. Hence, this study proposes the design of a drug delivery system with dual targeting capabilities, which may provide insights for the design and translation of precision therapies for CAVD.

10. The English writing should be revised throughout the article, regarding grammar or phrase construction. For example, the authors should use “targeted nanoparticles” instead of “targeting NPs” or “protein expression” vs “abundance”.

Response:

We are sorry for the low-quality English writing. We have corrected the improper use of spelling/grammar mistakes and revised our manuscript accordingly. In addition, the manuscript has been polished with the help of a professional editing service. And all the changes were marked in red in the revised manuscript.

11. Some paragraphs in the Introduction Section are not clear. Some examples are:

- “Magnetic navigation is a promising strategy to increase the drug accumulation. With the help of magnetic field, anti-CD63 antibodies-conjugated magnetic nanoparticles dragged captured circulating exosomes to accumulated around the infarcted myocardium.”

- “A disease specific cell membrane marker that exposure during proctological process can be a potential bollard for drug anchoring.”

Response:

We apologize for the lack of clarity. We've made corrections to the sentences, and hope these sentences are now clearer.

- We have rephrased the introduction of magnetic targeting.

- The statements “A disease specific cell membrane marker that exposure during proctological process can be a potential bollard for drug anchoring” had been corrected into “A cell membrane marker that specifically upregulated and exposed in diseased tissues can be a potential target for drug anchoring.”

Following are our revisions to the manuscript:

Page 7-8 of Revised Manuscript:

Magnetic targeting is a promising strategy for improving drug delivery efficiency. Due to the responsiveness to exogenous magnetic field (EMF), which is non-invasive and tissue-penetrating, magnetic nanoparticle (MNP)-based drug delivery systems are highly anticipated, especially for targeting deep tissues. To date, MNPs have primarily been applied for the targeted treatment of tumors. Guided by EMF, MNPs increase the localization of antineoplastic drugs and serve as the source of magneto-thermodynamic therapy.¹³ Moreover, the combination of magnetic targeting and hyperthermia has been applied in anti-infection therapy within deep tissues (e.g., bacterial osteomyelitis in bone marrow), which are not readily accessible via systemic drug administration.¹⁴ Meanwhile, MNPs have been increasingly used to transport

therapeutic agents to other pathological tissues, including the ischemic brain and heart.^{15,16} Additionally, mesenchymal stem cell-derived exosomes incorporated with MNPs have been directed to brain ischemic lesions, ultimately reducing the infarct volume and promoting motor function recovery.¹⁷ MNPs have also been conjugated with anti-CD63 and anti-myosin light chain (MLC) antibodies to increase heart function after myocardial infarction. More specifically, the MNPs capture endogenous circulating exosomes via the anti-CD63 antibody, causing accumulation in the ischemic heart under EMF navigation; subsequently, the anti-MLC antibodies bind to the damaged cardiomyocytes, releasing exosomes and increasing heart function.¹⁸ Although MNPs can be enriched around the diseased tissue under EMF guidance, the drug anchoring capacity must be enhanced to counteract the violent blood flow of the stenotic aortic valves. Moreover, targeting accuracy must be improved to enable nanoparticles to act on disease-causing valvular interstitial cells (VICs).

Page 8 of Revised Manuscript:

A cell membrane marker specifically upregulated and exposed in diseased tissues can be a target for drug anchoring. During CAVD progression, the lesion site features endothelium damage and the osteogenic differentiation of VICs. Hence, cell membrane markers highly expressed in osteogenically differentiated VICs could represent a potential TDDS targeting site for treating CAVD. One such example is protease-activated receptor-2 (PAR2), a membrane-bound G-protein coupled receptor; its hexapeptide ligand has high affinity and selectivity in binding to the extracellular segment of PAR2. Once bound to its ligand, PAR2 is internalized and routed to lysosomes. Moreover, PAR2 has been implicated in atherosclerosis, which shares similar initial pathological features with CAVD. Hence, if the protein expression of PAR2 is upregulated in CAVD, it may represent a potential marker of CAVD for a TDDS.

We would like to take this opportunity to thank you for all your time involved and this great opportunity for us to improve the manuscript. We hope you will find this revised version satisfactory.

Sincerely,

The Authors

-----End of Reply to Reviewer #3-----

REVIEWER COMMENTS

Reviewer #1 (Remarks to the Author):

In this extensive revision, the authors have addressed the majority of issues raised by this reviewer during initial review. However, some minor issues remain:

1. The authors now describe their target discovery pipeline in a more detailed way. However, a visualisation (eg, flow-chart; Supplementary figure) would be very helpful coupled with more details on each step of the discovery pipeline (eg, how where proteins screened for cellular distribution and/or whether they have "specific" ligands or not?). Also, limitations of their approach must be convincingly discussed.

2. Baseline characteristics of tissue donors are now noted in Suppl. Table 1 (of which only roughly 50% are shown in the submission documents as the right half is cut off), but the n-numbers do not match those shown in Suppl. Table 2. Secondly, data in Suppl. Table 1 should be shown quantitatively and not qualitatively (ie, there is no need to list each donor individually), and should be re-organized (columns = control/CAVD; rows = clinical variables). Third, Suppl. Table 1 and 2 appear to be redundant and could be combined in one table showing quantitative data. Finally, this reviewer assumes that 1:1 nearest neighbouring PSM was used and the "confounding variables" refer to the variables the PS was derived from. Please specify and use the correct terminology throughout.

Reviewer #2 (Remarks to the Author):

The authors have done a good job responding to the prior critiques.

Reviewer #3 (Remarks to the Author):

I would like to convey my appreciation for the comprehensive revision of the manuscript. I am pleased to report that your revisions were meticulous and effectively resolved the issues raised during the review process.

Response to reviewer #1

Reviewer #1 (Remarks to the Author):

In this extensive revision, the authors have addressed the majority of issues raised by this reviewer during initial review. However, some minor issues remain:

Dear reviewer:

Thank you very much for your careful reading, helpful comments, and constructive suggestions. According to your nice suggestions, we have made extensive corrections to our previous draft, the detailed corrections are listed below.

1. The authors now describe their target discovery pipeline in a more detailed way. However, a visualisation (eg, flow-chart; Supplementary figure) would be very helpful coupled with more details on each step of the discovery pipeline (eg, how where proteins screened for cellular distribution and/or whether they have "specific" ligands or not?). Also, limitations of their approach must be convincingly discussed.

Response:

Thank you for your great suggestion. We developed a flowchart in which we elaborated on the selection criteria and illustrated the number of candidate targets for each step (**Supplementary Fig. 2c**). First, up-regulated genes whose corresponding proteins localized to the cell membrane were screened by Gene Ontology (GO) annotation with term plasma membrane (GO:0005886). Then, those genes whose corresponding proteins can be shed or secreted outside the cell were excluded by the term extracellular space (GO:0005615). Subsequently, whether there are specific ligands for these proteins was identified by literature. Finally, those candidate genes were validated by qPCR. *NPR1*, *FPR3*, and *CSF2R* were excluded due to their low expression in hVICs, and *F2RL2* was excluded because PAR3 (i.e., *F2RL2*-encoded protein) does not respond to synthetic peptides that mimic the putative tethered ligand. The limitations of our approach were discussed.

Following are our revisions to the manuscript:

Page 2 of Revised Supplementary Information:

Supplementary Fig. 2.

(a) Flowchart for identifying PAR2 as a target of calcified aortic valve disease (CAVD).

Page 46-47 of Revised manuscript (limitations in Discussion sections):

This study has certain limitations. First, other cell surface proteins may also be appropriate markers for nanoparticle targeting. The current findings are inferences and validation based on transcriptome results from osteogenic differentiated VICs and then identified in pathologic valve tissues. If novel omics methods are applied to analyze the proteome or transcriptome of cells directly in valve tissues, some other membrane proteins may become candidate targets. Furthermore, in addition to membrane proteins, extracellular matrix proteins, as the main protein components of calcified thickened valves, may also be an effective target for CAVD TDDSs.⁶¹ By combining multi-omics analysis with and network-guide tactics, more targeting anchors may be able to identified and compared with higher throughput and efficiency. However, as an exploratory study, our results sufficiently demonstrate that PAR2 is elevated in aortic valve calcification and represents as an effective anchor for targeting drug delivery.

[ref]

[61] Voicu G, *et al.* Nanocarriers of shRNA-Runx2 directed to collagen IV as a nanotherapeutic system to target calcific aortic valve disease. *Mater Today Bio.* doi: 10.1016/j.mtbio.2023.100620. (2023)

2. Baseline characteristics of tissue donors are now noted in Suppl. Table 1 (of which only roughly 50% are shown in the submission documents as the right half is cut off), but the n-numbers do not match those shown in Suppl. Table 2. Secondly, data in Suppl. Table 1 should be shown quantitatively and not qualitatively (ie, there is no need to list each donor individually), and should be re-organized (columns = control/CAVD; rows = clinical variables). Third, Suppl. Table 1 and 2 appear to be redundant and could be combined in one table showing quantitative data. Finally, this reviewer assumes that 1:1 nearest neighbouring PSM was used and the "confounding variables" refer to the

variables the PS was derived from. Please specify and use the correct terminology throughout.

Response:

Thank you for your valuable suggestions.

- In fact, the patients listed in Supplementary Table 1 are donors of all the valve tissues we used throughout our experiments (e.g., western blot, immunohistochemistry, immunofluorescence, and isolation and culture of valvular interstitial cells), while the information listed in Supplementary Table 2 is the baseline information of the donors of valve tissues which were used for western blotting analysis of PAR2 expression in valve tissues (Fig. 2e). Thus, the n-numbers in Supplementary Table 1 do not match those shown in Supplementary Table 2. To make it more straightforward, we now added this information in table description.

- Additionally, as you suggested, we analyzed the baseline information and re-organized **Supplementary Table 1**.

- Thirdly, we agree that some of the information in Supplementary Table 2 and Supplementary Table 1 is duplicative and may be misleading. Therefore, we removed the unmatched columns in **Supplementary Table 2** to convey the information clearly.

- Finally, we changed the statements in the manuscript, and listed below.

Following are our revisions to the manuscript:

Page 13-14 of Revised manuscript:

To assess the robustness of the findings, propensity-score matching (PSM) was employed to minimize the influence of confounding variables. Baseline parameters, such as age, BMI, sex, smoking, comorbidity, and serum lipid profiles, were utilized as matching criteria. Following the matching process, no disparities were observed in these baseline characteristics, while differences in PAR2 protein expression remained (**Supplementary Table 2**).

Page 18 of Revised Supplementary Information (Supplementary Table 1):

Supplementary Table 1 Baseline clinical and echocardiographic parameters of the donors of tissues used for immunoblot, immunohistochemistry, immunofluorescence and cell isolation

parameters	Control (n=14)	CAVD (n=18)	p Value
Age	59.93 ± 16.07	61.44 ± 7.310	0.7238

Sex, (Female, %)	4 (28.57%)	7 (38.89%)	N/A
BMI (kg/m ²)	22.99 ± 3.787	22.27 ± 2.679	0.5357
Diabetes, n (%)	0	0	N/A
Hypertension, n (%)	7 (50%)	6 (33.33%)	N/A
Smoking, n (%)	5 (35.71%)	4 (22.22%)	N/A
Coronary heart disease, n (%)	2 (14.29%)	4 (22.22%)	N/A
Bicuspid aortic valves, n (%)	0	0	N/A
LVEF (%)	55.51 ± 9.498	61.37 ± 8.745	0.0860
Aortic valve area (cm ²)	2.826 ± 0.6929	0.8379 ± 0.5897	<0.0001
Aortic valve peak velocity (m/s)	1.927 ± 0.5202	4.435 ± 0.6985	<0.0001
Peak transvalvular pressure gradient (mmHg)	15.79 ± 8.911	80.56 ± 24.52	<0.0001
Mean transvalvular pressure gradient (mmHg)	7.846 ± 4.506	45.89 ± 16.05	<0.0001
LDL (mmol/L)	2.200 ± 0.5713	2.213 ± 0.8200	0.9608
HDL (mmol/L)	1.281 ± 0.3336	1.250 ± 0.3166	0.7921
Triglycerides (mmol/L)	1.079 ± 0.3184	1.872 ± 2.259	0.2035
Total cholesterol (mmol/L)	4.135 ± 0.7899	4.331 ± 0.9416	0.5372
Statins, n (%)	1 (13.3%)	3 (20%)	N/A
ACEi/ARB, n (%)	2 (20%)	2 (10%)	N/A

Values are mean ± standard deviation (SD) or %. CAVD, calcified aortic valve disease; BMI, body mass index; LVEF, left ventricular ejection fraction; LDL, low-density lipoprotein cholesterol; HDL, high-density lipoprotein cholesterol; ACEi, angiotensin-converting enzyme inhibitor; ARB, angiotensin receptor blocker.

Page 19 of Revised Supplementary Information (Supplementary Table 2):

Supplementary Table 2 PAR2 expression analysis in immunoblot after Propensity-score matching

Parameters	Control (n = 6)	CAVD (n = 6)	p Value
Age	59.5±10.5972	62.8333±7.4677	0.544
BMI (kg/m ²)	21.655±3.4219	22.9983±3.2718	0.503
Sex, (Female, %)	0 (0)	3 (50)	0.076
Smoking, n (%)	1 (16.67)	3 (50)	0.262
Bicuspid aortic valves, n (%)	0	0	N/A
Diabetes, n (%)	0	0	N/A

Hypertension, n (%)	3(50)	1(16.67)	0.262
Coronary heart disease, n (%)	1(16.67)	1(16.67)	1.000
LDL (mmol/L)	2.2033±0.4859	2.2083±0.8343	0.990
HDL (mmol/L)	1.4533±0.3154	1.35±0.2188	0.526
Triglycerides (mmol/L)	1.1217±0.3773	1.1767±0.3018	0.786
Total cholesterol (mmol/L)	4.3883±0.8134	4.2817±0.9082	0.835
Statin, n (%)	1(16.67)	1(16.67)	1.000
ACEi/ARB, n (%)	1(16.67)	1(16.67)	1.000
PAR2 expression	1.2667±0.9082	5.8833±3.0995	0.006

We would like to take this opportunity to thank you for all your time involved and this great opportunity for us to improve the manuscript. We hope you will find this revised version satisfactory.

Sincerely,

The Authors

-----End of Reply to Reviewer #1-----

Response to reviewer #2

Reviewer #2 (Remarks to the Author):

The authors have done a good job responding to the prior critiques.

Response:

Thank you very much for the positive comment.

Reviewer #3 (Remarks to the Author):

I would like to convey my appreciation for the comprehensive revision of the manuscript. I am pleased to report that your revisions were meticulous and effectively resolved the issues raised during the review process.

Response:

Thank you very much for your appreciation of our work.